

# Anomaly inflow for CSS and fractonic lattice models and dualities via cluster state measurement

Takuya Okuda,[1] Aswin Parayil Mana[2*] and Hiroki Sukeno[2]

**1** Graduate School of Arts and Sciences, University of Tokyo,
Komaba, Meguro-ku, Tokyo 153-8902, Japan
**2** C. N. Yang Institute for Theoretical Physics & Department of Physics and Astronomy,
State University of New York at Stony Brook, Stony Brook, NY 11794-3840, USA

* aswin.parayilmana@stonybrook.edu

## Abstract

Calderbank-Shor-Steane (CSS) codes are a class of quantum error correction codes that contains the toric code and fracton models. A procedure called foliation defines a cluster state for a given CSS code. We use the CSS chain complex and its tensor product with other chain complexes to describe the topological structure in the foliated cluster state, and argue that it has a symmetry-protected topological order protected by generalized global symmetries supported on cycles in the foliated CSS chain complex. We demonstrate the so-called anomaly inflow between CSS codes and corresponding foliated cluster states by explicitly showing the equality of the gauge transformations of the bulk and boundary partition functions defined as functionals of defect world-volumes. We show that the bulk and boundary defects are related via measurement of the bulk system. Further, we provide a procedure to obtain statistical models associated with general CSS codes via the foliated cluster state, and derive a generalization of the Kramers-Wannier-Wegner duality for such statistical models with insertion of twist defects. We also study the measurement-assisted gauging method with cluster-state entanglers for CSS/fracton models based on recent proposals in the literature, and demonstrate a non-invertible fusion of duality operators. Using the cluster-state entanglers, we construct the so-called strange correlator for general CSS/fracton models. Finally, we introduce a new family of subsystem-symmetric quantum models each of which is self-dual under the generalized Kramers-Wannier-Wegner duality transformation, which becomes a non-invertible symmetry.

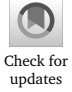
# 1  Introduction

The gapped phases of quantum matter at zero temperature showcase a variety of exotic properties at low energy. Topologically ordered states [1], exemplified by the toric code ground state, possess a non-trivial ground state degeneracy, and their excitations exhibit non-trivial braiding statistics, which are of practical use for the fault-tolerant quantum computation [2]. Preparation of topologically ordered states has been thus long sought after. However, their stability supported by the long-range entanglement is a hurdle that prohibits us from obtaining them with a finite-depth local unitary circuit, at the same time [3]. This 'no-go' was recently resolved for a large class of topologically ordered states using methods with short-range entanglers assisted by local measurements and adaptive, or feedforwarded correction procedures [4–11]. Remarkably, this measurement-assisted procedure was formulated as a physical incarnation of the celebrated Kramers-Wannier (KW) duality transformation [12–14] — a mathematical map relating two spin systems, where the global symmetries in the original theory are gauged. In the case of subsystem global symmetries, gauging them makes the transformation even more exotic, and the resulting states possess a so-called fracton order [15]. This procedure was also generalized in [7] to arbitrary Calderbank-Shor-Steane (CSS) codes [16, 17]. In constructing measurement-based KW transformations, the so-called cluster-state entangler has proved to be useful.

In the context of Measurement-Based Quantum Computation (MBQC) [18–22], one can implement a CSS code by a procedure called foliation [23], which constructs a cluster state extended along an extra direction. The 3d Raussendorf-Bravyi-Harrington (RBH) state [24], which is a 3d cluster state with qubits on a 3d cubic lattice, is a prominent example of the foliation construction and implements the 2d toric code. The quantum error correction is performed by single-qubit measurements on the foliated cluster state.

Our aim in this paper is to uncover the physics of a large class of quantum error correcting codes from the holographic perspective via measurement on a higher-dimensional cluster state. Our constructions and results apply to arbitrary CSS codes, which encompass many fracton models and low-density parity check codes [25] that attract significant attention recently.

We study an interplay between the bulk and boundary systems by analyzing their anomalies. We further exhibit dualities for both quantum models and partition functions that derive from CSS codes. When the model is self-dual, it exhibits a non-invertible symmetry [26, 27], an emerging concept in condensed matter physics, high energy physics, and quantum information science. Our results unify a number of prominent models in the literature and will be useful in studying quantum phases of matter with generalized global symmetries including non-invertible and subsystem symmetries. Let us give an overview of our main results below.

## 1.1 Summary of results

First, we reformulate the construction of the foliated cluster state introduced in [23] in terms of the so-called CSS chain complex (see e.g. Ref. [28]). We show that the tensor product of chain complexes can be used to obtain a new chain complex describing the foliated cluster state as well as the one that describes their spacetime lattice. Using it as an efficient tool, we describe symmetries in the foliated cluster state and show that it has symmetry-protected topological (SPT) order [29–37] with respect to symmetries extended from those of the boundary CSS code. When the boundary CSS code is subsystem symmetric, then the bulk symmetry can be also subsystem symmetric.[1]

We further establish the anomaly inflow mechanism [43] between an arbitray CSS code on the boundary and the foliated cluster state in the bulk. We write the bulk and the boundary partition functions with insertions of defects, which describe the spacetime motion of excitations, and show that, upon gauge transformations and when put on a manifold with boundaries, the bulk partition function produces a phase which is identical to an anomalous phase from the boundary partition function. We show that an operator configuration introduced to define the boundary partition function with defects is also produced from measuring the bulk state with defects. This gives a CSS generalization of the relation between the bulk SPT response and the boundary BF theory [44, 45]. In particular, our discussion covers the fracton models that have the structure of a CSS code such as the X-cube model and the checkerboard model [15].

Then, we utilize the (foliated) cluster state to construct a variety of statistical models. Using an appropriate product state $|\Omega(J,K)\rangle$, we get a relation of the form

$$\mathcal{Z}(J,K) = \mathcal{N} \times \langle \Omega(J,K)|\psi_{\mathcal{C}}\rangle \,, \tag{1}$$

where $|\psi_{\mathcal{C}}\rangle$ is the wave function of the foliated cluster state, $\mathcal{Z}(J,K)$ is the resulting partition function of a statistical model with parameters $J$ and $K$, and $\mathcal{N}$ is a normalization constant. For instance, setting the 2d quantum plaquette Ising model as the CSS code, the partition function of the 3d classical anisotropic plaquette Ising model, which is studied in the context of superconductors [46], is obtained. Studying the interplay between the foliated cluster state and measurements, we further obtain a generalization of the Kramers-Wannier-Wegner duality [13] for classical partition functions, where in the precise version, dual partition functions with defects are summed over, i.e., in the dual theory a symmetry is gauged. This gives a CSS generalization of such exact dualities mentioned e.g., in [47].

Then, we consider the measurement-assisted implementation of the KW operator following Refs. [5,7] for general CSS codes. We will demonstrate that the fusion of the duality operators is non-invertible,

$$\mathsf{KW}^{\dagger} \circ \mathsf{KW} = \sum (\text{symmetry generators}) \,, \tag{2}$$

where $\mathsf{KW}^{\dagger}$ is the adjoint of the Kramers-Wannier duality operator $\mathsf{KW}$. This generalizes a result for the 2d plaquette Ising model provided in [48], where the right hand side becomes the

---

[1]See e.g. Ref. [38, 39] for discussion on lattice models with SPT orders with respect to higher-form global symmetries and Refs. [40–42] with respect to subsystem global symmetries.

sum of rigid-line symmetry generators.[2] Our miscellaneous results along these lines include a mathematical proof that the protocol presented in [7] can be implemented deterministically for general CSS codes by explicitly proving that randomness of measurements can be converted into an operator that is homologically trivial. We also provide an extension of the procedure to the Chamon model [50], which is a non-CSS code.

Further, with appropriate product states $|\omega(K)\rangle$ and $|+\rangle^n$, we will obtain relations of the form

$$\mathbf{Z}(K) = \mathcal{N}' \times \langle \omega(K)|\mathsf{KW}|+\rangle^n, \tag{3}$$

where $\mathbf{Z}$ is a statistical partition function with parameter $K$ and $\mathcal{N}'$ is a normalization constant. The equation (3) is a generalization of the strange correlator [51, 52] in the sense that $\mathsf{KW}|+\rangle^n$ is a gapped quantum state described by the CSS code, which includes topologically ordered or fracton states, and the overlap gives us a classical statistical model.[3] As examples, we discuss the 2d classical Ising model, the 2d classical plaquette Ising model, and the 3d tetrahedral Ising model, whose partition functions we will derive from ground states of the toric code, the 2d quantum plaquette Ising model, and the checkerboard model, respectively. The general construction covers a variety of examples including other subsystem symmetric spin models, which we summarize and review in Table 1 and Appendix A. In the literature, the strange correlator for a topologically ordered state was constructed with its string-net representation [52]. Our result provides an alternative and straightforward route to obtaining a strange correlator for a broad class of gapped quantum systems represented by CSS codes.

Finally, we apply our results to a family of self-dual models with subsystem symmetries. The family contains the plaquette Ising model as an example and some of the models are new to the best of our knowledge. The Kramers-Wannier duality becomes a non-invertible symmetry (see e.g. [27, 53–57]) at a special coupling constant.

## 1.2 Example: The toric code and the RBH model

Let us present some of our main results using a particular example, before presenting our results in full generality in the main text. Here we avoid using the chain complex machinery as much as possible for the sake of accessibility. We take the 2d toric code on the 2d square lattice with the periodic boundary condition as an example of the CSS code.[4]

The toric code has stabilizers $A_v$ ($v \in V$: the set of vertices) and $B_f$ ($f \in F$: the set of faces), i.e.,

$$A_v|\Psi\rangle = B_f|\Psi\rangle = |\Psi\rangle, \tag{4}$$

with

$$A_v = \prod_{e \supset v} X_e = \begin{array}{c} X \\ X{-}\!\!|\!\!{-}X \\ X \end{array}, \tag{5}$$

$$B_f = \prod_{e \subset f} Z_e = \begin{array}{c} Z \\ Z \phantom{--} Z \\ Z \end{array}. \tag{6}$$

---

[2] Such a sum of symmetry generators can be identified with the so-called condensation defect [49], which arises from gauging the relevant symmetry along a submanifold with non-zero codimension.

[3] The relation (3) is different from the relation (1). For example, with the Kramers-Wannier transformation KW for the (2+1)d quantum plaquette Ising model, the relation (3) gives the 2d classical plaquette Ising model, while Eq. (1) gives the 3d classical anisotropic plaquette Ising model.

[4] This is a special case of the family of models studied in our previous paper [45], where the chain complex notation was used but foliation was not introduced.

To implement parity checks in the toric code within the MBQC framework, one can use the foliation construction, which we explain in the following section, and it leads to the 3d RBH state as shown in [23]. It is a stabilizer state defined on the 3d cubic lattice and qubits are placed on its faces and edges. We denote cells in the 3d lattice using bold fonts ($\boldsymbol{v}, \boldsymbol{e}, \boldsymbol{f}$, etc.) to distinguish them from those in 2d lattices at its boundaries. The state is stabilized by the following commuting operators:

$$K_f = X_f \prod_{e \subset f} Z_e = \quad \boxed{\begin{array}{ccc} & Z & \\ Z & X & Z \\ & Z & \end{array}} \quad , \tag{7}$$

$$K_e = X_e \prod_{f \supset e} Z_f = \tag{8}$$

The state can be seen as a ground state of the Hamiltonian $H_{\text{RBH}} = -\sum_{\boldsymbol{f} \in F} K_f - \sum_{\boldsymbol{e} \in E} K_e$. The state can be explicitly written as

$$|\psi_{\text{RBH}}\rangle = \prod_{f \in F} \prod_{e \subset f} CZ_{e,f} |+\rangle^{\otimes E} |+\rangle^{\otimes F}, \tag{9}$$

where $|+\rangle$ is the $+1$ eigenstate of the Pauli $X$ operator and $CZ_{j,k}|a\rangle_j \otimes |b\rangle_k = (-1)^{ab}|a\rangle_j \otimes |b\rangle_k$ ($a, b = 0, 1$) is the controlled-$Z$ gate.

### 1.2.1 SPT and anomaly inflow

The SPT order of the RBH model is demonstrated in Ref. [39], and the argument has been generalized to a family of cluster states in general dimensions in Ref. [58]. Namely, it has been shown that $H_{\text{RBH}}$ has the RBH state as the unique ground state when the model is placed on the periodic 3d lattice, while it has ground state degeneracies when it is put on lattices with boundaries — a hallmark of the bulk SPT order. The SPT order of the RBH model is protected by a product of 1-form and dual 1-form symmetries. In the bulk, it is generated by membrane operators,

$$U(\boldsymbol{M}) = \prod_{f \in M} X_f, \tag{10}$$

$$V(\boldsymbol{N}) = \prod_{e \in N} X_e, \tag{11}$$

where $\boldsymbol{M}$ is a closed surface formed by faces and $\boldsymbol{N}$ is a set of edges which forms a closed surface in the dual lattice. Later in this manuscript, in terms of the homology algebra, they correspond to chains termed cycles and dual cycles. They can also wrap along the periodic direction, in which case they are said to be in a non-trivial homology class. The surfaces that support the symmetry generators can be deformed "smoothly" in this example, but for general CSS chain complex — such as fractons — this topological deformation may not exit nor is it essential in our discussion.

To exhibit the ground state degeneracy on boundaries, we can for example consider smooth boundaries consisting of edges $E$, apply the symmetry generators on the bulk RBH state, and investigate how they act on the boundary degrees of freedom (i.e., the edge mode). They

become the pair of 1-form and dual 1-form symmetries in the boundary toric code:

$$U(\boldsymbol{M}) \longrightarrow W(M) = \prod_{e \in M} Z_e \,, \tag{12}$$

$$V(\boldsymbol{N}) \longrightarrow \widetilde{W}(N) = \prod_{e \in N} X_e \,, \tag{13}$$

where $M$ is a closed loop along edges in the 2d square lattice, and $N$ is a set of edges which forms a closed loop in the dual lattice. The cycle $N$ is simply the restriction of $\boldsymbol{N}$ to the boundary, while the cycle $M$ is related to $\boldsymbol{M}$ in a slightly more subtle manner. They anticommute $\{W(M), \widetilde{W}(N)\} = 0$ when $M$ and $N$ are in non-trivial homology classes with a nonzero intersection number in the 2d periodic lattice, which may occur when the parent bulk cycles are both in non-trivial homology classes in the 3d lattice, showing a projective symmetry action on the boundary system. In the main text, we generalize this argument to arbitrary CSS codes and their foliated cluster states.

We will give a path integral picture of the SPT/anomaly physics above and demonstrate the anomaly inflow. To define the path integral for the bulk theory, we identify excitations and operators that move them in the foliated cluster state. In the case with the RBH model, excitations are Pauli $Z$ operators supported on closed loops consisting of edges and a set of faces that forms a closed loop in the dual lattice. The former is moved by $X$ operators acting on face qubits, the latter by those acting on edge qubits. Indeed, we show that we can combine the excitations and the $X$ operator in spacetime by considering a four-dimensional (Euclidean) lattice and placing surface defects as the world-volume of the loop excitations. The two type of surface defects are cycles $\mathbf{z}_2$ (surfaces that consist of faces without boundaries) and dual cycles $\mathbf{z}_2^*$ (surfaces in the dual lattice that consist of dual faces without boundaries) in the four-dimensional spacetime lattice. By defining the path integral as a trace of the density matrix of the "initial" excited state and the time-ordered operator insertions describing the time evolution, we obtain the bulk partition function

$$Z_{\text{bulk}}[\mathbf{z}_2, \mathbf{z}_2^*] = (-1)^{\#(\mathbf{z}_2 \cap \mathbf{z}_2^*)} \,, \tag{14}$$

where $\#(\mathbf{z}_2 \cap \mathbf{z}_2^*)$ is the intersection number between $\mathbf{z}_2$ and $\mathbf{z}_2^*$, counting the number of faces (which can be identified with dual faces in four dimensions) incident with both $\mathbf{z}_2$ and $\mathbf{z}_2^*$.

Similarly, we can write down the partition function for the boundary toric code theory by evaluating a trace of the density matrix of the initial state with electric and magnetic excitations and insertions of time-ordered string operators that move them. This becomes a functional of spacetime loop-like defects denoted by $z_1$ (closed loops that consist of edges) and $z_1^*$ (closed loops that consist of dual edges), which we write $Z_{\text{bdry}}^{\text{toric code}}[z_1, z_1^*]$. The functional is in fact the path integral of the so-called BF theory.[5] This is as expected because the toric code was introduced as a topological $\mathbb{Z}_2$ gauge theory in the original paper [2], and the latter admits a BF theory formulation [60].

The background gauge fields for the 1-form symmetries in the toric code are Poincaré dual to the symmetry defects $(z_1, z_1^*)$, and their gauge transformation is a local deformation of defects. We show that the partition function $Z_{\text{bdry}}^{\text{toric code}}[z_1, z_1^*]$ picks up a sign upon such local deformations, which signals the 't Hooft anomaly of 1-form symmetries. We also show that the corresponding deformation in the bulk partition function similarly defined in the presence of boundaries precisely cancels the additional signs, exhibiting the anomaly inflow mechanism.

In the main text, we will be more explicit and fully general, and demonstrate the anomaly inflow between the bulk and boundary partition functions for CSS codes. The tensor product of chain complexes proves to be a powerful tool to describe physics of SPT and anomaly inflow.

---

[5]We obtain a discrete sum over two 1-chains. The summation is equivalent to the functional integration over two continuous gauge fields $a$ and $b$ with the action proportional to $\int b \wedge da$, which defines the BF theory. See, for example, Section 3.2 of [59] for an explanation.

### 1.2.2 Strange correlator and dualities associated with toric code

Now let us explain our results of strange correlators [51,52] and duality for CSS codes using the toric code as an example. Let us consider the $(2+1)$-dimensional $\mathbb{Z}_2$ lattice gauge theory, a model closely related to the toric code. The Hamiltonian is given by

$$H_{\text{LGT}} = -\sum_{e \in E} X_e - \lambda \sum_{f \in F} B_f \,, \tag{15}$$

and the model is symmetric under the transformations generated by $A_v$. The Kramers-Wannier dual of this model is the $(2+1)$d transverse-field Ising model, which lives on vertices in the dual lattice. We denote the cells in the dual lattice using asterisk. The Hamiltonian is given by

$$H_{\text{Ising}} = -\sum_{e^* \in E^*} \prod_{v^* \subset e^*} Z_{v^*} - \lambda \sum_{v^* \in V^*} X_{v^*} \,. \tag{16}$$

The Hamiltonian is invariant under the transformation generated by $U_0 = \prod_{v^* \in V^*} X_{v^*}$.

The Kramers-Wannier transformation between the two models can be implemented by the operator [5,8]

$$\text{KW} = \langle + |^{\otimes E} \prod_{f \in F} \prod_{e \subset f} CZ_{e,f} |+\rangle^{\otimes F} \,. \tag{17}$$

One can indeed verify that

$$\text{KW} \cdot H_{\text{LGT}} = H_{\text{Ising}} \cdot \text{KW} \,. \tag{18}$$

We obtain the fusion of symmetry generators and duality operators:

$$\text{KW} \cdot A_v = \text{KW} \,, \tag{19}$$

$$U_0 \cdot \text{KW} = \text{KW} \,, \tag{20}$$

$$\text{KW} \cdot \text{KW}^\dagger = \frac{1}{2^{|F|}} (1 + U_0) \,, \tag{21}$$

$$\text{KW}^\dagger \cdot \text{KW} = \frac{1}{2^{|E|}} \sum_{C^*} \prod_{e^* \in C^*} X_{e^*} \,, \tag{22}$$

where in the last equation, the sum is over all the possible loops $C^*$ in the dual lattice, including but not limited to the product of $A_v$'s. The duality operator and its adjoint do not become identity, but they lead to the sum of symmetry generators; namely, a projector to a symmetric subspace. The duality maps KW and $\text{KW}^\dagger$ are hence non-invertible. We give an implementation of the operator KW with CZ gates, local measurements, and feedforward operations, generalizing e.g. Refs. [4,5,7] to general CSS codes in the main text.

The state $\text{KW}^\dagger |+\rangle^{\otimes F}$ is a ground state of the toric code. The overlap with a product state

$$\langle \omega(K) | = \bigotimes_{e \in E} \langle 0|_e e^{KX_e} \,, \tag{23}$$

can be easily computed using this expression, and it is proportional to the partition function of the classical 2d Ising model,

$$\mathbf{Z}_{\text{Ising}}(K) = \sum_{\{s_{v^*} = \pm 1\}_{v^* \in V^*}} \exp \left[ K \sum_{e^* \in E^*} \prod_{v^* \subset e^*} s_{v^*} \right] \,. \tag{24}$$

We will also show a general recipe based on the CSS chain complex to obtain the Kramers-Wannier-Wegner dual of the partition function obtained as a strange correlator. The derivation is done in Appendix D and makes use of the exact solvability in stablizer states. In the current example, we get the dual 2d classical Ising model with twists:

$$\mathbf{Z}_{\text{Ising}}(K) = \frac{2^{|F|}(\sinh 2K)^{|E|/2}}{2^{|V|}|H_1(T^2, \mathbb{Z}_2)|} \sum_{[z_1] \in H_1(T^2, \mathbb{Z}_2)} \mathbf{Z}_{\text{dual Ising}}^{\text{twisted}}(K^*, z_1). \tag{25}$$

Here, $H_1(T^2, \mathbb{Z}_2)$ denotes the first homology group of the 2-dimensional torus with $\mathbb{Z}_2$ coefficients, and $[z_1]$ is the homology class represented by the cycle $z_1$ (closed loops that consist of edges). There are four distinct classes. We used $K^* = -\frac{1}{2}\log\tanh K$. We also introduced the twisted partition function

$$\mathbf{Z}_{\text{dual Ising}}^{\text{twisted}}(K^*, z_1) = \sum_{\{s_v = \pm 1\}_{v \in V}} \exp\left[K^* \sum_{e \in E} (-1)^{\#(z_1 \cap e^*)} \prod_{v \in e} s_v\right], \tag{26}$$

where $\#(z_1 \cap e^*)$ is the intersection number between $z_1$ and $e^*$ and we identify $e^*$ with $e$.

### 1.2.3 Strange correlator and dualities associated with the RBH model

Finally, we explain our results on strange correlators and duality for foliated cluster states using the RBH model $|\psi_{\text{RBH}}\rangle$ as an example. In foliated cluster states for most of CSS codes, the foliated direction and its orthogonal direction becomes anisotropic. In such cases, we will introduce $J$ for the coupling constant in the direction orthogonal to the foliation, while we use $K$ for the coupling in the foliation direction. For the RBH model (and cluster states in arbitrary dimensions considered in Ref. [58]), the foliated cluster state happens to be isotropic, and hence we simply use $K$ in this subsection. We denote the cells in the dual lattice using asterisk.

We take a wave function overlap between $|\psi_{\text{RBH}}\rangle$ and the product state

$$\langle\Omega(K)| = \bigotimes_{e \in E} \langle+|_e \bigotimes_{f \in F} \langle 0|_f e^{KX_f}, \tag{27}$$

and we find that it is proportional to the partition function, or the Euclidean path integral of the 3d $\mathbb{Z}_2$ lattice gauge theory (see also Ref. [58]):

$$\mathscr{Z}_{\text{gauge}}(K) = \sum_{\{s_e = \pm 1\}_{e \in E}} \exp\left[K \sum_{f \in F} \prod_{e \in f} s_e\right]. \tag{28}$$

Again, by making use of the solvability in the stabilizer formalism, we construct a generalization of the Kramers-Wannier-Wegner duality for CSS codes. In the current example with the RBH model, we obtain a canonical example (see also Ref. [45]):

$$\mathscr{Z}_{\text{gauge}}(K) = \frac{2^{|V|}(\sinh 2K)^{|F|}}{2^{|C|}|H_2(T^3, \mathbb{Z}_2)|} \sum_{[z_2] \in H_2(T^3, \mathbb{Z}_2)} \mathscr{Z}_{\text{Ising}}^{\text{twisted}}(K^*, z_2), \tag{29}$$

where $H_2(T^3, \mathbb{Z}_2)$ denotes the second homology group of the 3-dimensional torus with $\mathbb{Z}_2$ coefficients, and $[z_2]$ is the homology class represented by the cycle $z_2$ (closed surfaces that consist of faces). The symbol $|C|$ is the number of cubes in the 3d lattice. The summand in the right hand side is the twisted Ising partition function in three dimensions and it is given by

$$\mathscr{Z}_{\text{Ising}}^{\text{twisted}}(K^*, z_2) = \sum_{\{s_{v^*} = \pm 1\}_{v^* \in V^*}} \exp\left[K^* \sum_{e^* \in E^*} (-1)^{\#(z_2 \cap e^*)} \prod_{v^* \in e^*} s_{v^*}\right], \tag{30}$$

where $\#(z_2 \cap e^*)$ is the intersection number between $z_2$ and $e^*$.

Compared to (25), the foliated cluster state gives rise to a duality of partition functions in one higher dimension. It is interesting to notice a hierarchy: the quantum Hamiltonian $H_{\text{LGT}}$ in (15) can be obtained from $\mathcal{Z}_{\text{gauge}}(K)$ via the classical-quantum correspondence (see e.g. [14]) and similarly the quantum Ising model in (16) comes from 3d classical Ising model.

In generalizing the above examples to general CSS codes, we will obtain statistical models whose Boltzmann weight reflects the structure of the foliated cluster state. In the generalized Kramers-Wannier-Wegner duality, the sum over standard homology classes in (29) will be replaced by those of the foliated CSS chain complex, whose efficacy we will appreciate throughout this work.

## 1.3 Organization of the paper

The rest of the paper is organized as follows. In Section 2, we review the construction of the foliated cluster state [23] and reformulate it in terms of the CSS complex. We then argue that, when the CSS code defines a lattice model, the foliated cluster state is in the SPT phase with respect to the symmetries dictated by the CSS code. We illustrate the symmetries and the excitations of the cluster state with examples that possess subsystem symmetries. In Section 3, we demonstrate the anomaly inflow mechanism for a general CSS code and its foliated cluster state. Readers who are interested in dualities may skip this section. Section 4 is devoted to the study of the interplay between the foliated cluster state and the Kramers-Wannier dualities. In Section 5, we study some models that are invariant under the Kramers-Wannier duality. Appendix A studies the Kramers-Wannier duality for the CSS codes that underlie some fracton models. In Appendix B, we study the measurement-based gauging (Kramers-Wannier duality) of a non-CSS code, namely, the Chamon model. Appendix C provides evidence that the foliated cluster state possesses an SPT order using an argument based on gauging. In Appendix D, we provide a duality between classical statistical models obatined as strange correlators of CSS code ground states. In Appendix E, we prove the non-degeneracy of the intersection pairing between homology classes, a fact we used in proving the possibility of a correction procedure for random measurement outcomes when preparing general CSS codes. Appendix F reviews the algebraic formalism for translationally invariant lattice models and applies it to study the Kramers-Wannier duality, the correctability of measurement outcomes, and the ground state degeneracies for various higher dimensional models that we consider in Section 5.

## 2 Symmetry protected topological states for CSS codes

### 2.1 Foliation of CSS codes

To discuss the anomaly inflow and SPT states, we will make use of the foliation construction of a cluster state for a given CSS code [23].[6] Here we give a short summary. A stabilizer code is the subspace stabilized by a set of stabilizer operators, each of which is a product of Pauli $X$ or $Z$ operators multiplied by a phase. A CSS code by definition has stabilizers consisting of purely $X$ or purely $Z$ operators. We let $\mathcal{S}_X = \{A_\alpha\}$ and $\mathcal{S}_Z = \{B_\beta\}$ be the sets of stabilizer generators made of $X$ and $Z$ operators and labeled by $\alpha$ and $\beta$, respectively. We label the physical qubits of the code by $i = 1, \ldots, n$. For a given CSS code, we define a graph $G_X$ from the $X$-stabilizers $\mathcal{S}_X$ as follows [23].

- Introduce $n + |\mathcal{S}_X|$ vertices, labeled by either $i$ or $\alpha$.

- Introduce a single edge between vertices $i$ and $\alpha$ if and only if the stabilizer $A_\alpha$ contains $X$ for qubit $i$ so that all the edges in $G_X$ are of this type.

---

[6]The possibility to construct SPT states via foliation was mentioned in [61]. See [28] for the construction of an SPT state in one *lower* dimension for an arbitrary CSS code.

From the $Z$-stabilizers $\mathcal{S}_Z$ we construct a graph $G_Z$ in a similar way. We then construct a bigger graph $\mathcal{G}$. Let $j$ be integers in some range $J = \{j = 0, 1, \dots, L_w - 1\}$, which is periodic ($j \sim j + L_w$) or open. The graph $\mathcal{G}$ is constructed by the following procedure.

- For each integer $j \in J$, let $G_Z^{(j)}$ be a copy of the graph $G_Z$.

- For a half-integer $j + 1/2$, let $G_X^{(j+1/2)}$ be a copy of the graph $G_X$.

- Between $j$ and $j + 1/2$, introduce an edge connecting the qubit $i$ in graph $G_Z^{(j)}$ and the qubit $i$ in graph $G_X^{(j+1/2)}$, for all $i = 1, \dots, n$.

- Between $j - 1/2$ and $j$, introduce an edge connecting the qubit $i$ in graph $G_X^{(j-1/2)}$ and the qubit $i$ in graph $G_Z^{(j)}$ for $i = 1, \dots, n$.

The foliated cluster state $|\psi_{\mathcal{C}}^{(\mathrm{CSS})}\rangle$ for a given CSS code is obtained from the graph $\mathcal{G}$ as the simultaneous $+1$ eigenstate of the stabilizers

$$K_v = X_v \prod_{\langle v, v' \rangle \in E} Z_{v'} \quad (v \in V), \tag{31}$$

where $V$ and $E$ are the sets of vertices and edges in $\mathcal{G}$ and the product is over the vertices $v'$ such that there is an edge $\langle v, v' \rangle$ connecting $v$ and $v'$. The state can be also written as[7]

$$|\psi_{\mathcal{C}}^{(\mathrm{CSS})}\rangle = \prod_{\langle v, v' \rangle \in E} CZ_{v,v'} |+\rangle^V, \tag{32}$$

where $CZ_{1,2}$ is the controlled-$Z$ gate, $CZ_{1,2} = |0\rangle_1 \langle 0|_1 \otimes I_2 + |1\rangle_1 \langle 1|_1 \otimes Z_2$. See Figure 2 for an illustration by a relevant model.

## 2.2 Chain complex for foliation

We assign abstract symbols $\sigma_i$ to the qubits $i$ above and denote their set by $\Delta_{\mathrm{q}} = \{\sigma_i\}_{i=1,\dots,n}$. Likewise, we let $\Delta_X = \{\sigma_\alpha\}_{\alpha=1,\dots,|\mathcal{S}_X|}$ ($\Delta_Z = \{\sigma_\beta\}_{\beta=1,\dots,|\mathcal{S}_Z|}$) be the set of abstract symbols assigned to the stabilizers $A_\alpha$ ($B_\beta$). When the CSS code is the toric code discussed in Section 1.2, $\sigma_\alpha \in \Delta_X$ is a vertex, $\sigma_i \in \Delta_{\mathrm{q}}$ is an edge, and $\sigma_\beta \in \Delta_Z$ is a face, all in a square lattice. We write $C_k$ ($k = \mathrm{q}, X, Z$) for the group of chains $c_k$ with $\mathbb{Z}_2$ coefficients — i.e., the formal linear combinations

$$c_k = \sum_{\sigma \in \Delta_k} a(c_k; \sigma) \sigma, \tag{33}$$

with $a(c_k; \sigma) = \{0, 1 \bmod 2\}$. For any CSS code, one can introduce a CSS chain complex [2, 62, 63] (see also [25])

$$0 \xrightarrow{\delta} C_Z \xrightarrow{\delta_Z} C_{\mathrm{q}} \xrightarrow{\delta_X} C_X \xrightarrow{\delta} 0. \tag{34}$$

Here $\delta_Z(\sigma_\beta)$ is the sum of $\sigma_i$'s such that the vertices $\beta$ and $i$ are connected in $G_Z$, and $\delta_X(\sigma_i)$ is the sum of $\sigma_\alpha$'s such that the vertices $i$ and $\alpha$ are connected in $G_X$. The nilpotency condition $\delta_X \circ \delta_Z = 0$ is equivalent to the commutativity of the stabilizers. For each lattice model, we will give explicit expressions for the differentials. The groups[8] $\mathrm{Im}\,\delta_Z$ and $\mathcal{L} := \mathrm{Ker}\,\delta_X / \mathrm{Im}\,\delta_Z$

---

[7]In general, we write $|\phi\rangle^S$ for the tensor product of copies of state $|\phi\rangle$ for qubits labeled by a set $S$.

[8]Throughout the paper, $\mathrm{Im}\,\varphi := \{\varphi(g) \,|\, g \in G\} \subset H$ denotes the image of $\varphi$ and $\mathrm{Ker}\,\varphi = \{g \in G \,|\, \varphi(g) = 0\} \subset G$ denotes the kernel of $\varphi$, where $\varphi : G \to H$ is a homomorphism between Abelian groups $G$ and $H$. We also denote by $\mathrm{Hom}(G, H)$ the group consisting of all the homomorphisms from $G$ to $H$.

label the stabilizers and the logical operators made of $Z$, respectively. We can also consider the dual chain (cochain) complex

$$0 \xleftarrow{\delta^*} C_Z \xleftarrow{\delta_Z^*} C_{\mathrm{q}} \xleftarrow{\delta_X^*} C_X \xleftarrow{\delta^*} 0, \tag{35}$$

where we identify the dual group $C_k^* = \mathrm{Hom}(C_k, \mathbb{Z}_2)$ of the group $C_k$ with $C_k$ itself using the bases $\Delta_k$ and the intersection pairing.[9] The groups $\mathrm{Im}\,\delta_X^*$ and $\mathcal{L}^* := \mathrm{Ker}\,\delta_Z^* / \mathrm{Im}\,\delta_X^*$ label the stabilizers and the logical operators made of $X$, respectively.

As a useful notation, for a chain $c_k \in C_k$, we write

$$P(c_k) = \prod_{\sigma \in \Delta_k} P(\sigma)^{a(c_k; \sigma)}, \tag{36}$$

with $P(\sigma)$ a single-qubit Pauli or Hadamard operator acting on the qubit at $\sigma \in \Delta_k$. Then, the CSS stabilizers can be written as

$$A_\alpha = X(\delta_X^* \sigma_\alpha), \qquad B_\beta = Z(\delta_Z \sigma_\beta), \tag{37}$$

with $\sigma_\alpha \in \Delta_X$ and $\sigma_\beta \in \Delta_Z$. The models we will consider are defined either by the Hamiltonian[10]

$$H^{\mathrm{CSS}} = -\sum_{\sigma_\alpha \in \Delta_X} A_\alpha - \sum_{\sigma_\beta \in \Delta_Z} B_\beta, \tag{38}$$

or by the Hamiltonian

$$\widetilde{H}^{\mathrm{CSS}} = -\lambda \sum_{\sigma_i \in \Delta_{\mathrm{q}}} X_i - \sum_{\sigma_\beta \in \Delta_Z} B_\beta, \tag{39}$$

with parameter $\lambda$ and the Gauss law constraint $A_\alpha = 1$ for $\sigma_\alpha \in \Delta_X$ imposed on the physical states. We ensure the locality of the models by demanding that each model is defined on a lattice such that its fundamental region contains a finite number of stabilizer generators, each of which only contains Pauli operators from nearby cells.

To reformulate the foliation procedure, we generalize the sets $\Delta_k$ ($k = \mathrm{q}, X, Z$) by introducing a coordinate $w$, which is periodic $w \sim w + L_w$ if the periodic boundary condition is imposed. We define $\mathbf{\Delta}_k$ ($k = \mathrm{q}, X, Z$) as the set equipped with a point $\{w = j\}$ on the line parametrized by $w$:

$$\mathbf{\Delta}_k = \bigcup_j \mathbf{\Delta}_k^{(j)}, \qquad \mathbf{\Delta}_k^{(j)} = \{\sigma \times \{w = j\} \,|\, \sigma \in \Delta_k\}, \tag{40}$$

where the range for $j$ is $0 \le j \le L_w$ ($0 \le j \le L_w - 1$) for the open (periodic) boundary condition. On the other hand, we define $\mathbf{\Delta}_{k,w}$ ($k = \mathrm{q}, X, Z$) as the set equipped with an interval $[j, j+1] = \{j \le w \le j+1\}$:

$$\mathbf{\Delta}_{k,w} = \bigcup_{j=0}^{L_w - 1} \mathbf{\Delta}_{k,w}^{(j)}, \qquad \mathbf{\Delta}_{k,w}^{(j)} = \{\sigma \times [j, j+1] \,|\, \sigma \in \Delta_k\}. \tag{41}$$

---

[9]Explicitly, we identify $c^* \in \mathrm{Hom}(C_k, \mathbb{Z}_2)$ with $c \in C_k$ so that $c^*(c') = \#(c \cap c')$ for any $c' \in C_k$, where $\#(c \cap c') = \sum_{\sigma \in \Delta_k} a(c; \sigma) a(c'; \sigma)$ is the intersection number (pairing) between $c$ and $c'$. Since $c^*(\delta c') = (\delta^* c)(c')$ by definition, we have $\#(c \cap \delta c') = \#(\delta^* c \cap c')$.

[10]In Kitaev's toric code discussed in Section 1.2, the differential $\delta_Z \sigma_2$ ($\sigma_2 \in \Delta_2$) is the sum of four edges around a face, and $\delta_X^* \sigma_0$ ($\sigma_0 \in \Delta_0$) is the sum of four edges around a vertex. The stabilizers $A_\alpha$ and $B_\beta$ are explicitly given in (5) and (6).

We define chains $c_k$ and $c_{k,w}$ ($k = q, Z, X$) in the same manner as above and define the tensor product $\otimes$ as a multi-linear map satisfying $\sigma \otimes \tau = \sigma \times \tau$ for cells $\sigma$ and $\tau$, where $\times$ denotes the direct product of sets. The foliation construction of the graph $\mathcal{G}$ can be described with the following *foliated chain complex*:

$$0 \xrightarrow{\delta} C_{Z,w} \xrightarrow{\delta} C_Z \oplus C_{q,w} \xrightarrow{\delta} C_q \oplus C_{X,w} \xrightarrow{\delta} C_X \xrightarrow{\delta} 0 \,. \tag{42}$$

The differentials $\delta$ in the foliated chain complex are defined as follows.[11]

- For $\sigma_{Z,w} = \sigma_Z \times [w, w+1]$ with $\sigma_Z \in \Delta_Z$, we define

$$\delta \sigma_{Z,w} = \delta_Z \sigma_Z \otimes [w, w+1] + \sigma_Z \otimes \big(\{w\} + \{w+1\}\big)\,. \tag{43}$$

- For $\sigma_Z = \sigma_Z \times \{w\}$ with $\sigma_Z \in \Delta_Z$, we define

$$\delta \sigma_Z = \delta_Z \sigma_Z \otimes \{w\}\,. \tag{44}$$

- For $\sigma_{q,w} = \sigma_q \times [w, w+1]$ with $\sigma_q \in \Delta_q$, we define

$$\delta \sigma_{q,w} = \delta_X \sigma_q \otimes [w, w+1] + \sigma_q \otimes \big(\{w\} + \{w+1\}\big)\,. \tag{45}$$

- For $\sigma_q = \sigma_q \times \{w\}$ with $\sigma_q \in \Delta_q$, we define

$$\delta \sigma_q = \delta_X \sigma_q \otimes \{w\}\,. \tag{46}$$

- For $\sigma_{X,w} = \sigma_X \times [w, w+1]$ with $\sigma_X \in \Delta_X$, we define

$$\delta \sigma_{X,w} = \sigma_X \otimes \big(\{w\} + \{w+1\}\big)\,. \tag{47}$$

- For $\sigma_X = \sigma_X \times \{w\}$ with $\sigma_X \in \Delta_X$, we define

$$\delta \sigma_X = 0\,. \tag{48}$$

The nilpotency of the differentials can be shown by explicit calculations. For example, take $\sigma_{Z,w} \in C_{Z,w}$. We have $\delta \sigma_{Z,w} = \delta_Z \sigma_Z \otimes [w, w+1] + \sigma_Z \otimes \big(\{w\} + \{w+1\}\big)$. The first term is in $C_{q,w}$ and the second term is in $C_Z$. The differential of the first term is then $\delta_Z \sigma_Z \otimes \big(\{w\} + \{w+1\}\big)$ (note that we used $\delta^2 = 0$ to reduce terms), and the differential of the second term is also $\delta_Z \sigma_Z \otimes \big(\{w\} + \{w+1\}\big)$. Thus $\delta^2 \sigma_{Z,w} = 0$. One can repeat this calculation for every generator of every chain group to see that $\delta$ is nilpotent. The complex (42) is the tensor product[12] of the CSS complex (34) and the cell complex of a 1-dimensional lattice. We obtain the dual differential $\delta^*$ as the dual (i.e., transpose if formulated in terms of matrices) of $\delta$.

We identify "vertices" in the graph in the foliation process [23] with elements of $\Delta_k$ as follows:

- $G_Z^{(j)}$ consists of vertices assigned to $\Delta_Z \cup \Delta_q$. After the foliation, vertices are placed at $\Delta_Z \cup \Delta_q$ at the layer $w = j$.

- $G_X^{(j+1/2)}$ consists of vertices assigned to $\Delta_X \cup \Delta_q$. After the foliation, vertices are placed at $\Delta_{X,w} \cup \Delta_{q,w}$ with the interval $w = [j, j+1]$.

---

[11]Here we omit subscripts from differentials $\delta$ to simplify notations. We make them distinguishable from the context.

[12]Given two complexes $\ldots \xrightarrow{\delta} C_{i+1} \xrightarrow{\delta} C_i \xrightarrow{\delta} \ldots$ and $\ldots \xrightarrow{\delta'} C'_{i+1} \xrightarrow{\delta'} C'_i \xrightarrow{\delta'} \ldots$ with $\mathbb{Z}_2$-coefficients, their tensor product is the complex defined by $(C \otimes C')_i = \bigoplus_j C_j \otimes C_{i-j}$ with differentials given by $c \otimes c' \mapsto \delta c \otimes c' + c \otimes \delta' c$.

Let us write $\boldsymbol{\Delta}_{Q_1} = \boldsymbol{\Delta}_Z \cup \boldsymbol{\Delta}_{q,w}$ and $\boldsymbol{\Delta}_{Q_2} = \boldsymbol{\Delta}_q \cup \boldsymbol{\Delta}_{X,w}$.[13] Qubits in the foliated cluster state are placed at $\boldsymbol{\Delta}_{Q_1} \cup \boldsymbol{\Delta}_{Q_2}$. Let us write $\boldsymbol{C}_{Q_1} = \boldsymbol{C}_Z \oplus \boldsymbol{C}_{q,w}$ and $\boldsymbol{C}_{Q_2} = \boldsymbol{C}_q \oplus \boldsymbol{C}_{X,w}$. Then, the foliated chain complex can be written as

$$0 \xrightarrow{\delta} C_{Z,w} \xrightarrow{\delta} C_{Q_1} \xrightarrow{\delta} C_{Q_2} \xrightarrow{\delta} C_X \xrightarrow{\delta} 0, \tag{49}$$

and its dual chain complex as

$$0 \xleftarrow{\delta^*} C_{Z,w} \xleftarrow{\delta^*} C_{Q_1} \xleftarrow{\delta^*} C_{Q_2} \xleftarrow{\delta^*} C_X \xleftarrow{\delta^*} 0. \tag{50}$$

The stabilizers (31) of the foliated cluster state can be written using a notation similar to (36) as

$$K(\boldsymbol{\sigma}) = X(\boldsymbol{\sigma})Z(\boldsymbol{\delta\sigma}) \qquad (\boldsymbol{\sigma} \in \boldsymbol{\Delta}_{Q_1}), \tag{51}$$

$$K(\boldsymbol{\tau}) = X(\boldsymbol{\tau})Z(\boldsymbol{\delta^*\tau}) \qquad (\boldsymbol{\tau} \in \boldsymbol{\Delta}_{Q_2}). \tag{52}$$

The foliated cluster state $|\psi_C^{(\mathrm{CSS})}\rangle$ can be written as

$$|\psi_C^{(\mathrm{CSS})}\rangle = \mathcal{U}_{CZ}|+\rangle^{\boldsymbol{\Delta}_{Q_1} \cup \boldsymbol{\Delta}_{Q_2}}, \tag{53}$$

$$\mathcal{U}_{CZ} = \prod_{\substack{\boldsymbol{\sigma} \in \boldsymbol{\Delta}_{Q_1} \\ \boldsymbol{\tau} \in \boldsymbol{\Delta}_{Q_2}}} CZ_{\boldsymbol{\sigma},\boldsymbol{\tau}}^{a(\boldsymbol{\delta\sigma};\boldsymbol{\tau})}. \tag{54}$$

The operators $X(\boldsymbol{\delta\tau})$ and $X(\boldsymbol{\delta^*\sigma})$ ($\boldsymbol{\tau} \in \boldsymbol{\Delta}_{Z,w}$, $\boldsymbol{\sigma} \in \boldsymbol{\Delta}_X$) play the role of parity check operators in the topological measurement-based quantum computation [23, 24, 64] based on the foliated cluster state.

## 2.3 SPT order of the foliated cluster state

Below, we analyze the foliated cluster state (53) for a general CSS code that defines a quantum many-body system. The general discussion here will be followed by the analysis of concrete examples. Since the state is the (unique) eigenstate of $K(\boldsymbol{\sigma})$ ($\forall \boldsymbol{\sigma} \in \boldsymbol{\Delta}_{Q_1}$), it is also a $+1$ eigenstate of $K(\boldsymbol{c}) := X(\boldsymbol{c})Z(\boldsymbol{\delta c})$ ($\boldsymbol{c} \in \boldsymbol{C}_{Q_1}$). It follows that

$$X(\boldsymbol{z})|\psi_C^{(\mathrm{CSS})}\rangle = K(\boldsymbol{z})|\psi_C^{(\mathrm{CSS})}\rangle = |\psi_C^{(\mathrm{CSS})}\rangle, \tag{55}$$

where $\boldsymbol{z} \in \boldsymbol{C}_{Q_1}$ is a cycle ($\boldsymbol{\delta z} = 0$) in the foliated chain complex. Similarly,

$$X(\boldsymbol{z}^*)|\psi_C^{(\mathrm{CSS})}\rangle = |\psi_C^{(\mathrm{CSS})}\rangle, \tag{56}$$

where $\boldsymbol{z}^* \in \boldsymbol{C}_{Q_2}$ is a dual cycle ($\boldsymbol{\delta^* z^*} = 0$) in the foliated chain complex. The operators $X(\boldsymbol{z})$ and $X(\boldsymbol{z}^*)$ can be seen as the generators of global symmetry transformations.[14] The global symmetry can be a higher-form or subsystem symmetry depending on the CSS code.

---

[13]In the foliation process for a specialized CSS code (generalized toric code) in Ref. [45], $\boldsymbol{\Delta}_{Q_1}$ corresponds to $n$-cells and $\boldsymbol{\Delta}_{Q_2}$ to $(n-1)$-cells in the $d$-dimensional hypercube lattice. For the RBH cluster state [24] discussed in Section 1.2, which is obtained as the foliation of the toric code, $\boldsymbol{\Delta}_{Q_1}$ and $\boldsymbol{\Delta}_{Q_2}$ are respectively the sets of faces and edges in a 3-dimensional cubic lattice.

[14]Taking the toric code as an example of a CSS code, the foliated cluster state is the 3d RBH cluster state, and it has a pair of 1-form symmetries, see Ref. [39].

### 2.3.1 Projective representation on the boundary

Let us consider the graph and complexes constructed as in Sections 2.1 and 2.2, with with the open boundary condition. We entangle general states $|\phi^{(0)}\rangle$ and $|\phi^{(L_w)}\rangle$ defined on qubits in $\Delta_q^{(0)}$ and $\Delta_q^{(L_w)}$ with a cluster state in the bulk:

$$|\psi_{\mathcal{C}}^{(\text{CSS})}; \phi^{(0)}, \phi^{(L_w)}\rangle := \mathcal{U}_{CZ}\big(|+\rangle_{\text{bulk}}|\phi\rangle_{\text{bdry}}\big), \tag{57}$$

where

$$|+\rangle_{\text{bulk}} := |+\rangle^{\Delta_Z \cup \Delta_{q,w} \cup \Delta_{X,w} \cup (\Delta_q \setminus (\Delta_q^{(0)} \cup \Delta_q^{(L_w)}))}, \qquad |\phi\rangle_{\text{bdry}} := |\phi^{(0)}\rangle|\phi^{(L_w)}\rangle, \tag{58}$$

and $\mathcal{U}_{CZ}$ is the product of $CZ$ gates defined as before. Note that we include $\Delta_Z^{(0)}$ and $\Delta_Z^{(L_w)}$ in the bulk part.

We are interested in two kinds of bulk operators. The first kind is $X(\boldsymbol{z}_2^*)$ with $\boldsymbol{z}_2^* \in \boldsymbol{C}_{Q_2}$ satisfying $\boldsymbol{\delta}^* \boldsymbol{z}_2^* = 0$ and $\boldsymbol{z}_2^* = z_q^{*(0)} \otimes \{w = 0\} + z_q^{*(L_w)} \otimes \{w = L_w\} + \ldots$, where the ellipses are supported on $\Delta_{Q_2} \setminus (\Delta_q^{(0)} \cup \Delta_q^{(L_w)})$. The second is $X(\boldsymbol{z}_1^{\text{rel}})$ on $|\psi_{\mathcal{C}}^{(\text{CSS})}, \phi\rangle$ with $\boldsymbol{z}_1^{\text{rel}} \in \boldsymbol{C}_{Q_1}$ satisfying $\boldsymbol{\delta} \boldsymbol{z}_1^{\text{rel}} = z_q^{(0)} \otimes \{w = 0\} + z_q^{(L_w)} \otimes \{w = L_w\}$ with $z_q^{(0)}, z_q^{(L_w)} \in \boldsymbol{C}_q$. The bulk operator $X(\boldsymbol{z}_2^*)$ induces logical operator $X(z_q^*)$ on states at $w = 0, L_w$:

$$X(\boldsymbol{z}_2^*)\mathcal{U}_{CZ}\big(|+\rangle_{\text{bulk}}|\phi\rangle_{\text{bdry}}\big) = \mathcal{U}_{CZ}X(\boldsymbol{z}_2^*)Z(\boldsymbol{\delta}^*\boldsymbol{z}_2^*)\big(|+\rangle_{\text{bulk}}|\phi\rangle_{\text{bdry}}\big)$$
$$= \mathcal{U}_{CZ}\big(|+\rangle_{\text{bulk}}X(z_q^{*(0)})X(z_q^{*(L_w)})|\phi\rangle_{\text{bdry}}\big). \tag{59}$$

The other bulk operator $X(\boldsymbol{z}_1^{\text{rel}})$ induces the logical operator $Z(z_q)$ on the boundary:

$$X(\boldsymbol{z}_1^{\text{rel}})\mathcal{U}_{CZ}\big(|+\rangle_{\text{bulk}}|\phi\rangle_{\text{bdry}}\big) = \mathcal{U}_{CZ}X(\boldsymbol{z}_1^{\text{rel}})Z(\boldsymbol{\delta}\boldsymbol{z}_1^{\text{rel}})\big(|+\rangle_{\text{bulk}}|\phi\rangle_{\text{bdry}}\big)$$
$$= \mathcal{U}_{CZ}\big(|+\rangle_{\text{bulk}}Z(z_q^{(0)})Z(z_q^{(L_w)})|\phi\rangle_{\text{bdry}}\big). \tag{60}$$

See Figure 1 for an illustration. In particular, on each boundary $w = 0$ or $w = L_w$, the logical operators furnish a non-trivial projective representation of (copies of) $\mathbb{Z}_2 \times \mathbb{Z}_2$, which guarantees the non-trivial degeneracy on the boundary and the existence of edge modes.

This is strong evidence for an SPT order. See [39,65] for a related argument. We note that the logical operators are all induced by bulk $X$-operators that commute with each other. This will be important for anomaly inflow to be discussed in Section 3.

In Appendix C, we provide another argument for an SPT order of the foliated cluster state by gauging global symmetries.[15]

### 2.3.2 Partition function

A useful quantity that characterizes an SPT order is the partition function that depends on the background gauge fields coupled to the global symmetries that protect the topological order. Each background gauge field is Poincaré dual to the world-volume of a defect that generates a global symmetry. Here we propose a formula for the partition function of the foliated cluster state on a torus as a functional of the world-volumes of defects.

Symmetry generators $X(\boldsymbol{z})$ and $X(\boldsymbol{z}^*)$ in (55) and (56) are defects extended purely in the spatial directions ($w \sim w + L_w$). We introduce defects whose world-volumes include not only spatial directions but also the time direction. Consider deforming a purely spatial world-volume supported on a cycle into an arbitrary shape in spacetime. On the intersection of the deformed world-volume and a constant time slice, there appears a cycle

$$\boldsymbol{z}_{Q_2} \in \boldsymbol{C}_{Q_2}, \quad \text{with} \quad \boldsymbol{\delta}\boldsymbol{z}_{Q_2} = 0, \qquad \text{or} \qquad \boldsymbol{z}_{Q_1}^* \in \boldsymbol{C}_{Q_1}, \quad \text{with} \quad \boldsymbol{\delta}^*\boldsymbol{z}_{Q_1}^* = 0. \tag{61}$$

---

[15]Another piece of evidence for an SPT order is that the global symmetries act projectively on the tensor network representation of the cluster state, as explicitly described in the case of Wegner models in [58].

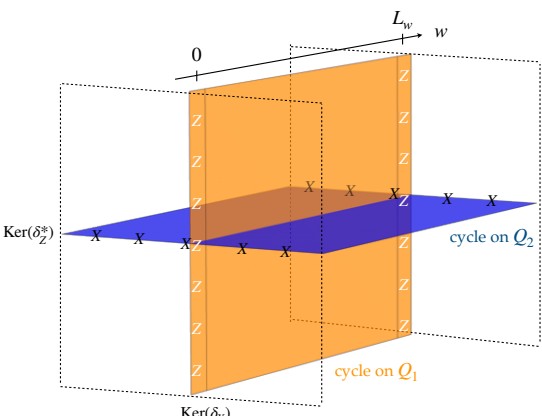

Figure 1: The foliated cluster state with edge modes satisfies the relations in (59) and (60). The logical operator in (59) is supported on $z_q^{*(0)}$ and $z_q^{*(L_w)}$, which are elements in $\mathrm{Ker}(\delta_Z^*)$. On the other hand, the logical operator in (60) is supported on $z_q^{(0)}$ and $z_q^{(L_w)}$, which are elements in $\mathrm{Ker}(\delta_X)$.

Such cycles represent the excitations (violation of stabilizer conditions) created by $Z(\mathbf{z}_{Q_2})$ and $Z(\mathbf{z}_{Q_1}^*)$. The location of a defect can be changed by acting with the $X$ operators supported on a chain. For example, acting with $X(\mathbf{c}_{Q_2}^*)$ such that $\boldsymbol{\delta}^* \mathbf{c}_{Q_2}^* = \mathbf{z}_{Q_1}^{*\prime} - \mathbf{z}_{Q_1}^*$ changes the location of an excitation from $\mathbf{z}_{Q_1}^*$ to $\mathbf{z}_{Q_1}^{*\prime}$:

$$X(\mathbf{c}_{Q_2}^*) \cdot Z(\mathbf{z}_{Q_1}^*)|\psi_{\mathcal{C}}^{(\mathrm{CSS})}\rangle = K(\mathbf{c}_{Q_2}^*) Z(\mathbf{z}_{Q_1}^{*\prime})|\psi_{\mathcal{C}}^{(\mathrm{CSS})}\rangle = Z(\mathbf{z}_{Q_1}^{*\prime})|\psi_{\mathcal{C}}^{(\mathrm{CSS})}\rangle, \tag{62}$$

where we used the equality $K(\mathbf{c}_{Q_2}^*) = X(\mathbf{c}_{Q_2}^*) Z(\boldsymbol{\delta}^* \mathbf{c}_{Q_2}^*)$. The combination of such excitations and $X$ operators form the world-volume of a defect in spacetime.

The world-volumes of defects can be described succinctly by a chain complex that models the spacetime. We model the Euclidean time by the circle $M_\tau$ parametrized with $\tau \sim \tau + L_\tau$ and consider 0-cells $\{j\}$ and 1-cells $[j, j+1]$ parametrized by integers $j$. Let us consider the cell complex $C_1^{(\tau)} \xrightarrow{\partial} C_0^{(\tau)}$, where $C_i^{(\tau)}$ is the Abelian group generated by $i$-cells. We then take the tensor product[16] of the complex in (42) and $C_\bullet(M_\tau)$. The product complex includes a portion

$$\ldots \xrightarrow{\mathbf{d} = \delta \otimes \mathrm{id} + \mathbf{id} \otimes \partial} \boldsymbol{C}_{Q_1} \otimes C_0(M_\tau) \oplus \boldsymbol{C}_{Q_2} \otimes C_1(M_\tau) \xrightarrow{\mathbf{d}' = \delta \otimes \mathrm{id} + \mathbf{id} \otimes \partial} \ldots \tag{63}$$

A general cycle $\mathbf{z} \in \mathrm{Ker}\, \mathbf{d}'$ is of the form

$$\mathbf{z} = \sum_j \mathbf{c}_j \otimes \{j\} + \sum_j \mathbf{z}_j \otimes [j, j+1], \tag{64}$$

where $\mathbf{c}_j \in \boldsymbol{C}_{Q_1}$, $\mathbf{z}_j \in \boldsymbol{C}_{Q_2}$, $\delta \mathbf{z}_j = 0$, and $j \in \mathbb{Z}$. To $\mathbf{z}$, we associate the following operator insertions and the excitations of states. At time $\tau = j$, we insert (act by) the operator $X(\mathbf{c}_j)$. During the period $[j, j+1]$, the system is in a state excited by $Z(\mathbf{z}_j)$. The condition $\mathbf{d}' \mathbf{z} = 0$, or equivalently

$$\mathbf{z}_j = \mathbf{z}_{j-1} + \delta \mathbf{c}_j, \tag{65}$$

implies that the excitations are created and destroyed on the boundary $\delta \mathbf{c}_j$ of the chain $\mathbf{c}_j$ on which $X$ operators are inserted.

---

[16] See footnote 12.

Similarly, a general dual cycle $\mathbf{z}^* \in \operatorname{Ker} \mathbf{d}^*$ takes the form

$$\mathbf{z}^* = \sum_r \mathbf{z}_r^* \otimes [r, r+1] + \sum_r \mathbf{c}_r^* \otimes \{r\}, \tag{66}$$

with $\mathbf{z}_r^* \in C_{Q_1}$, $\delta^* \mathbf{z}_r^* = 0$, $\mathbf{c}_r^* \in C_{Q_2}$, and $r \in \mathbb{Z} + 1/2$.[17] We associate to $\mathbf{z}^*$ the insertions of $X(\mathbf{c}_r^*)$ at times $\tau = r$ and the excitations by $Z(\mathbf{z}_r^*)$ during the period $[r, r+1]$.

With such associations, the partition function[18]

$$Z_{\text{foliated}}^{\text{CSS}}[\mathbf{z}, \mathbf{z}^*] := \operatorname{Tr}\left[ X(\mathbf{c}_{L_\tau}) \dots X(\mathbf{c}_1) X(\mathbf{c}_{L_\tau - 1/2}^*) \dots X(\mathbf{c}_{1/2}^*) \mathbb{P}(\mathbf{z}_0, \mathbf{z}_{-1/2}^*) \right], \tag{67}$$

where

$$\mathbb{P}(\mathbf{z}_0, \mathbf{z}_{-1/2}^*) = Z(\mathbf{z}_0) Z(\mathbf{z}_{-1/2}^*) |\psi_{\mathcal{C}}^{(\text{CSS})}\rangle \langle \psi_{\mathcal{C}}^{(\text{CSS})}| Z(\mathbf{z}_{-1/2}^*) Z(\mathbf{z}_0), \tag{68}$$

is given in terms of the intersection number $\#(\mathbf{z} \cap \mathbf{z}^*)$ as

$$Z_{\text{foliated}}^{\text{CSS}}[\mathbf{z}, \mathbf{z}^*] = (-1)^{\#(\mathbf{z} \cap \mathbf{z}^*)}. \tag{69}$$

To see this, note that the insertion of $X(\mathbf{c}_j) = K(\mathbf{c}_j) Z(\delta \mathbf{c}_j)$ or $X(\mathbf{c}_r^*) = K(\mathbf{c}_r^*) Z(\delta^* \mathbf{c}_r^*)$ picks up a sign when the position of a $K$ operator coincides with that of an excitation, and changes the locations of excitations by the $Z$ operators. See [45] for a more explicit computation in the case of Wegner's models.

## 2.4 Foliated cluster states for fracton models

In this subsection, we illustrate the construction of the foliated cluster state using examples with subsystem symmetries.

### 2.4.1 $(2+1)$d plaquette Ising model

We begin with the plaquette Ising model in 2+1 dimensions (2d-qPIM). An advantage of this model is that the symmetry generators for the foliated cluster state can be visualized by drawing three-dimensional figures. On the other hand, the model is atypical because one of the differentials, namely $\delta_X$, is zero.

The model defined by the Hamiltonian

$$H_{\text{2d-qPIM}} = -\sum_{x,y} Z_{x,y} Z_{x+1,y} Z_{x,y+1} Z_{x+1,y+1} = -\sum_f \prod_{v \subset f} Z_v. \tag{70}$$

Here $v = (x, y) \in \mathbb{Z}^2 / (L_x \mathbb{Z} \oplus L_y \mathbb{Z})$ are the vertices of a 2-dimensional square lattice of linear sizes $L_x$ and $L_y$. The model is solvable and has $2^{L_x + L_y - 1}$ ground states [66]. On each vertex $v$, we introduce a qubit. An elementary excitation, where the operators $\prod_{v \subset f} Z_v$ take values $+1$ for all faces $f$ except one, cannot be moved to another location by the action of Pauli operators without producing extra excitations. On the other hand, a pair of such adjacent excitations in the $x$ ($y$)-direction can be moved in the $y$ ($x$)-direction by the action of a string of $X$-operators.

The model can be understood in terms of a stabilizer code whose defining stabilizers are (up to minus signs) the terms in the Hamiltonians. This stabilizer code is of the CSS type because each stabilizer consists of entirely Pauli $Z$ operators. We identify $\Delta_Z = \Delta_{xy}$, $\Delta_q = \Delta_0$, $\Delta_X = 0$. Here and in what follows, subscripts $\{x, y, z, w\}$ in $\Delta_\bullet$ denote the directions in which

---

[17]We identify $[r, r+1]$ with $\{r + 1/2\} \in C_0(M_\tau)$, and $\{r\}$ with $[r - 1/2, r + 1/2] \in C_1(M_\tau)$.

[18]The $X$ insertions can be viewed as time evolution operators, and the projections as specifying the initial condition.

each cell is stretched. When the subscript is 0, then $\Delta_0$ denotes the set of vertices. The CSS chain complex for the 2d-qPIM is

$$0 \xrightarrow{\delta} C_{xy} \xrightarrow{\delta_Z} C_0 \xrightarrow{\delta_X} 0\,, \tag{71}$$

where differentials are defined as follows.

- For a plaquette $[x, x+1] \times [y, y+1] \in \Delta_{xy}$, we set

$$\delta_Z([x, x+1] \times [y, y+1]) = \{(x, y)\} + \{(x+1, y)\} + \{(x, y+1)\} + \{(x+1, y+1)\}\,, \tag{72}$$

where the R.H.S. is a formal sum of four vertices.

- For a vertex $\{(x, y)\} \in \Delta_0$, we set

$$\delta_X(\{(x, y)\}) = 0\,. \tag{73}$$

Then the stabilizer can be written as $\mathcal{S}_Z = \left\{ Z(\delta_Z f) \right\}_{f \in \Delta_{xy}}$. For this model, as can be inferred from (71), there is no $X$-type stabilizer: $\mathcal{S}_X = \emptyset$.

Global symmetries of the plaquette Ising model are generated by the operators $X(z^*)$ ($z^* \in C_0$ with $\delta_Z^* z^* = 0$) and the operators $Z_{x,y}$. Examples of $z^* \in C_0$ with $\delta_Z^* z^* = 0$ are the straight lines in the $x$- or $y$-direction, corresponding to the operators

$$W_{(1)}(x) = \prod_{y=1}^{L_y} X_{x,y}\,, \qquad W_{(2)}(y) = \prod_{x=1}^{L_x} X_{x,y}\,. \tag{74}$$

The operators obey the anti-commutation relations

$$Z_{x,y} W_{(1)}(x) = -W_{(1)}(x) Z_{x,y}\,, \qquad Z_{x,y} W_{(2)}(y) = -W_{(2)}(y) Z_{x,y}\,. \tag{75}$$

The operators $W_{(1,2)}$ and $Z_{x,y}$ are special cases of logical operators mentioned at the beginning of Section 2.2.

Following the general formalism we outlined in the previous subsections, we consider cells $\Delta_{Z,w} = \Delta_{xyw}$ (cubes), $\Delta_Z = \Delta_{xy}$ (horizontal faces), $\Delta_{q,w} = \Delta_w$ (vertical edges), $\Delta_q = \Delta_0$ (vertices), $\Delta_{X,w} = 0$, $\Delta_X = 0$. We make the $w$ direction periodic with $w \sim w + L_w$. We obtain the foliated chain complex for the 2d-qPIM:

$$0 \xrightarrow{\delta} C_{xyw} \xrightarrow{\delta} C_{xy} \oplus C_w \xrightarrow{\delta} C_0 \xrightarrow{\delta} 0\,. \tag{76}$$

Qubits in the foliated cluster state are placed at $\Delta_{Q_1} = \Delta_{xy} \cup \Delta_w$ and $\Delta_{Q_2} = \Delta_0$. We use the entangler

$$\mathcal{U}_{CZ} = \prod_{\substack{\sigma \in \Delta_{xy} \cup \Delta_w \\ \tau \in \Delta_0}} CZ_{\sigma,\tau}^{a(\delta\sigma;\tau)} = \Big( \prod_{e \in \Delta_w} \prod_{v \subset e} CZ_{e,v} \Big) \times \Big( \prod_{f \in \Delta_{xy}} \prod_{v \subset f} CZ_{f,v} \Big)\,, \tag{77}$$

to define the cluster state

$$|\psi_C^{(\text{2d-qPIM})}\rangle = \mathcal{U}_{CZ} |+\rangle^{\Delta_0 \cup \Delta_w \cup \Delta_{xy}}\,. \tag{78}$$

The stabilizers can be written as

$$K_f = X(f) Z(\delta f)\,, \qquad K_e = X(e) Z(\delta e)\,, \qquad K_v = X(v) Z(\delta^* v)\,. \tag{79}$$

By construction, the cluster state is the ground state of the Hamiltonian

$$H_C^{\text{2d-qPIM}} = -\sum_f K_f - \sum_e K_e - \sum_v K_v\,. \tag{80}$$

See Figure 2.

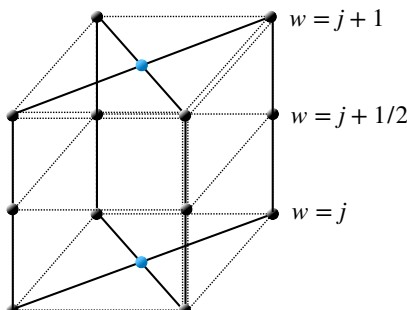

Figure 2: The unit cell of the cubic lattice on which the bulk cluster state for the $(2+1)$d plaquette Ising model is defined. The horizontal layer for $w = j$ ($j \in \mathbb{Z}$) corresponds to the face stabilizers $A_f = \prod_{v \subset f} Z_v$ and contains qubits corresponding to code qubits (on vertices) and $Z$-stabilizers (on horizontal faces). The horizontal layer for $w = j + 1/2$ contains only qubits corresponding to code qubits (on vertical edges).

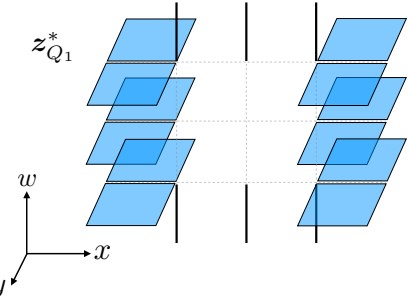

Figure 3: An example of excitations in $H_C^{\text{2d-qPIM}}$ described by a dual cycle $z_{Q_1}^*$.

A cycle $z_{Q_2}$ that supports excitations is an arbitrary linear combination of vertices. Examples of dual cycles $z_{Q_1}^*$ in (61) that support excitations are

- straight lines in the $w$-direction formed by faces in the $xy$-directions,

- straight lines in the $x$- or $y$-direction formed by edges in the $w$-direction, and more generally,

- curves in an $xw$- or $yw$-plane formed by faces in the $xy$-directions and edges in the $w$-direction. This is depicted in Figure 3.

A curve in the third item is of the form

$$z_{Q_1}^* = \sum_{j=0}^{L_w-1} z_{xy}^{*(j)} \otimes \{w = j\} + \sum_{j=0}^{L_w-1} \{(x_j, y_j)\} \times \{j \leq w \leq j+1\}, \tag{81}$$

with

$$z_{xy}^{*(j)} = z_{xy}^{*(0)} + \sum_{i=0}^{j-1} \delta_Z^* \{(x_i, y_i)\}, \qquad \sum_{i=0}^{L_w-1} \delta_Z^* \{(x_i, y_i)\} = 0. \tag{82}$$

Similar constructions will be used for many examples below.

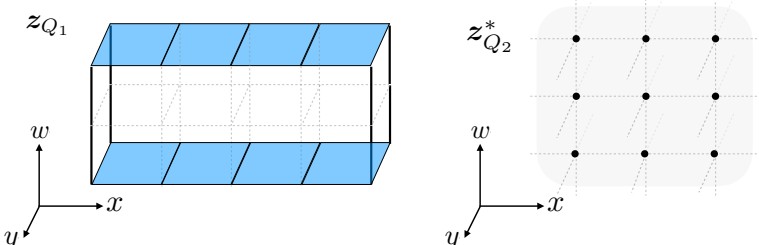

Figure 4: Symmetries in the cluster state $H_C^{\text{2d-qPIM}}$. (Left) An example of symmetry generators supported on $\boldsymbol{z}_{Q_1}$. (Right) An example of symmetry generators supported on $\boldsymbol{z}_{Q_2}^*$.

A symmetry generator of the foliated cluster state is supported on a cycle $\boldsymbol{z}_{Q_1}$ or a dual cycle $\boldsymbol{z}_{Q_2}^*$. Examples of a cycle $\boldsymbol{z}_{Q_1}$ are

- straight lines in the $w$-direction formed by vertical edges giving the symmetry generators

$$\prod_{w=0}^{L_w-1} X_{\{(x,y)\}\times[w,w+1]}\,. \tag{83}$$

- straight lines in the $x$- or $y$-direction formed by faces in the $xy$-directions giving

$$\prod_{x=0}^{L_x-1} X_{[x,x+1]\times[y,y+1]\times\{w\}}\,, \qquad \prod_{y=0}^{L_y-1} X_{[x,x+1]\times[y,y+1]\times\{w\}}\,, \tag{84}$$

and more generally

- curves in an $xw$- or $yw$-plane formed by vertical edges and faces in the $xy$-directions.

An example of a dual cycle $\boldsymbol{z}_{Q_2}^*$ is an $xw$- or $yw$-plane formed by vertices, giving

$$\prod_{y=1}^{L_y}\prod_{w=1}^{L_w} X_{(x,y,w)}\,, \qquad \prod_{x=1}^{L_x}\prod_{w=1}^{L_w} X_{(x,y,w)}\,. \tag{85}$$

The two types of symmetries are depicted in Figure 4.

### 2.4.2 X-cube model

The X-cube model is a quantum spin model on a 3d cubic lattice, which we take to be periodic in all three directions with periods $L_x$, $L_y$, and $L_z$. We identify $\Delta_Z = \Delta_{xyz}$ (cubes), $\Delta_q = \bigcup_{k=x,y,z} \Delta_k$ (edges), and $\Delta_X = \bigcup_{k=x,y,z} \Delta_0^{(k)}$ (three copies of vertices labeled by $k$ at each vertex). The chain complex for the X-cube model is given by

$$0 \xrightarrow{\delta} C_{xyz} \xrightarrow{\delta_Z} \bigoplus_{k=x,y,z} C_k \xrightarrow{\delta_X} \bigoplus_{k=x,y,z} C_0^{(k)} \xrightarrow{\delta} 0\,, \tag{86}$$

where differentials are defined as follows.

- For $[x,x+1]\times[y,y+1]\times[z,z+1]\in\Delta_{xyz}$, we set

$$\begin{aligned}
\delta_Z\big([x,x+1]\times[y,y+1]&\times[z,z+1]\big)\\
&= \sum_{s,t=0,1}\Big([x,x+1]\times\{(y+s,z+t)\}+\{x+s\}\times[y,y+1]\times\{z+t\}\\
&\qquad + \{(x+s,y+t)\}\times[z,z+1]\Big)\,. \tag{87}
\end{aligned}$$

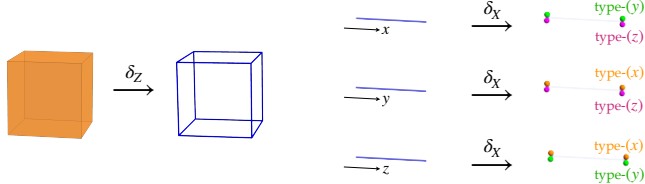

Figure 5: The X-cube model.

- For $[x, x+1] \times \{(y,z)\} \in \Delta_x$, we set

$$\delta_X\big([x,x+1] \times \{(y,z)\}\big) = \sum_{k=y,z} \{(x,y,z)\}^{(k)} + \sum_{k=y,z} \{(x+1,y,z)\}^{(k)}, \qquad (88)$$

where $\{(\bullet,\bullet,\bullet)\}^{(k)} \in \Delta_0^{(k)}$. Differentials for cells in $\Delta_y$ and $\Delta_z$ are defined in similar manners.

The chain $\delta_Z c$ ($c \in \Delta_{xyz}$) is a sum of twelve edges contained in the cube. The chain $\delta_X e$ ($e \in \bigcup_{k=x,y,z} \Delta_k$) is a sum of four vertices at the end of the edge, where the type labeled by $k$ differs from the direction of the edge. For illustration, take a vertex $\sigma_0^{(x)} \in \Delta_0^{(x)}$. The dual differential $\delta_X^* \sigma_0^{(x)}$ is the sum over four adjacent edges within the plane perpendicular to the $x$ direction. Using this set of definitions, stabilizers for the X-cube model can be written as

$$\mathcal{S}_Z = \big\{Z(\delta_Z c)\big\}_{c \in \Delta_{xyz}}, \qquad \mathcal{S}_X = \big\{X(\delta_X^* v)\big\}_{v \in \bigcup_{k=x,y,z} \Delta_0^{(k)}}, \qquad (89)$$

where the first is the set of cube terms, and the second of star terms in the defining Hamiltonian

$$H_{\mathrm{XC}} = -\sum_{k=x,y,z} \sum_{v \in \Delta_0^{(k)}} X(\delta_X^* v) - \sum_{c \in \Delta_{xyz}} Z(\delta_Z c). \qquad (90)$$

See Figure 5 for illustration. $\mathrm{Im}\, \delta_X$ dictates the linear mobility of lineons, and $\mathrm{Im}\, \delta_Z^*$ dictates the planar mobility of planons (pairs of fractons separated in the $x$-, $y$-, or $z$-direction).

Global symmetries are generated by $Z(z)$ and $X(z^*)$ ($z, z^* \in \bigoplus_{k=x,y,z} C_k$) such that $\delta_X z = 0$ and $\delta_Z^* z^* = 0$. An example of $z$ is a straight line (without ends) in the $k$-direction formed by edges in the $k$-direction ($k = x, y, z$), giving a string operator $Z(z)$ as a symmetry generator. For a line $c \in C_k$ with two ends, the string operator $Z(c)$ creates excitations at the ends that can be moved along the line by the application of another string operator; each of such excitations is known as a lineon, while the pair of lineons separated in the $k$-direction is mobile along the plane perpendicular to the $k$-direction and is known as a planon [67]. An example of $z^*$ is a surface with no corner in the $k'k''$-directions formed by edges in the $k$-direction ($k, k', k''$ distinct), giving a membrane operator $X(z^*)$ as a symmetry generator. For a rectangle $c^* \in C_k$ in the $k'k''$-directions with four corners, the membrane operator $X(c^*)$ creates excitations at the corners; these excitations are known as fractons [15].

The foliated cluster state is defined on a 4-dimensional hypercubic lattice. For 1-cells in the $w$-direction, we consider three copies of $\Delta_w$ and denote them by $\Delta_w^{(k)}$ with $k = x, y, z$. We place one qubit on each $c \in \Delta_{xyz}$, one qubit on $e \in \Delta_x \cup \Delta_y \cup \Delta_z$, one qubit on $e \in \Delta_w^{(k)}$ ($k = x, y, z$), and one qubit on $f \in \Delta_{xw} \cup \Delta_{yw} \cup \Delta_{zw}$. We write

$$\Delta_{Q_1} = \Delta_{xyz} \cup \bigcup_{k=x,y,z} \Delta_{kw}, \qquad \Delta_{Q_2} = \Big( \bigcup_{k=x,y,z} \Delta_k \Big) \cup \Big( \bigcup_{k=x,y,z} \Delta_w^{(k)} \Big), \qquad (91)$$

as well as

$$C_{Q_1} = C_{xyz} \oplus \bigoplus_{k=x,y,z} C_{kw}, \qquad C_{Q_2} = \Big( \bigoplus_{k=x,y,z} C_k \Big) \oplus \Big( \bigoplus_{k=x,y,z} C_w^{(k)} \Big). \qquad (92)$$

Note that qubits are placed at $\mathbf{\Delta}_{Q_1} \cup \mathbf{\Delta}_{Q_2}$. Then the relevant chain complex is

$$0 \xrightarrow{\delta} C_{xyzw} \xrightarrow{\delta} \underbrace{C_{xyz} \oplus \bigoplus_{k=x,y,z} C_{kw}}_{=C_{Q_1}} \xrightarrow{\delta} \underbrace{\Big( \bigoplus_{k=x,y,z} C_k \Big) \oplus \Big( \bigoplus_{k=x,y,z} C_w^{(k)} \Big)}_{=C_{Q_2}} \xrightarrow{\delta} \bigoplus_{k=x,y,z} C_0^{(k)} \xrightarrow{\delta} 0. \tag{93}$$

Differentials are nilpotent by construction. The foliated cluster state $|\psi_C^{(\mathrm{XCM})}\rangle$ is characterized by the stabilizers

$$\begin{aligned} K(\boldsymbol{\sigma}) &= X_{\boldsymbol{\sigma}} Z(\boldsymbol{\delta}\boldsymbol{\sigma}) && (\boldsymbol{\sigma} \in \mathbf{\Delta}_{Q_1}), \\ K(\boldsymbol{\sigma}') &= X_{\boldsymbol{\sigma}'} Z(\boldsymbol{\delta}^* \boldsymbol{\sigma}') && (\boldsymbol{\sigma}' \in \mathbf{\Delta}_{Q_2}). \end{aligned} \tag{94}$$

An excitation is supported by a cycle $\boldsymbol{z}_{Q_2} \in C_{Q_2}$ with $\boldsymbol{\delta}\boldsymbol{z}_{Q_2} = 0$ or a dual cycle $\boldsymbol{z}_{Q_1}^* \in C_{Q_1}$ with $\boldsymbol{\delta}^* \boldsymbol{z}_{Q_1}^* = 0$. Examples of a cycle $\boldsymbol{z}_{Q_2}$ are

- a straight line in the $w$-direction formed by edges in the $w$-direction and of type $x$, $y$, or $z$,

- a straight line in the $x$-, $y$-, or $z$-direction formed by edges in that direction, and more generally

- a curve on a $kw$-plane ($k = x, y, z$) formed by edges in the $w$-direction (of types $k'$ and $k''$) and edges in the $k$-direction ($k$, $k'$, and $k''$ distinct).

Examples of dual cycles $\boldsymbol{z}_{Q_1}^*$ are

- a straight line in the $w$-direction formed by cubes in the $xyz$-directions,

- a straight line in the $x$- or $y$-direction formed by faces in the $zw$-directions (and similar straight lines obtained by permuting $x$, $y$, and $z$), and more generally

- a curve on a $kw$-plane ($k = x, y, z$) formed by cubes in the $xyz$-directions and faces in the $k'w$-direction ($k' \neq k$).[19]

A symmetry generator of the foliated cluster state is supported by a cycle $\boldsymbol{z}_{Q_1} \in C_{Q_1}$ with $\boldsymbol{\delta}\boldsymbol{z}_{Q_1} = 0$ or a dual cycle $\boldsymbol{z}_{Q_2}^* \in C_{Q_2}$ with $\boldsymbol{\delta}^* \boldsymbol{z}_{Q_2}^* = 0$. Examples of a cycle $\boldsymbol{z}_{Q_1}$ are

- a plane in the $kw$-directions ($k = x, y, z$) formed by faces in the $kw$-directions,

- a plane in the $kk'$-directions ($k \neq k'$) formed by cubes in the $xyz$-directions, and more generally

- a straight line in the $k$-direction ($k = x, y, z$) times a curve on a $k'w$-plane ($k' \neq k$) formed by faces in the $kw$-directions and cubes in the $xyz$-directions.[20]

Examples of a dual cycle $\boldsymbol{z}_{Q_2}^*$ are

- a plane in the $k'w$-directions formed by edges in the $k$-direction ($k \neq k'$),

- a plane in the $kk'$-directions formed by of type $k''$ ($k, k', k''$ distinct) in the $w$-direction, and more generally

- a straight line in the $k$-direction times a curve on a $k'w$-plane formed by edges in the $k'$-direction and edges of type $k''$ ($k, k', k''$ distinct) in the $w$-direction. This is illustrated in Figure 6.

---

[19]To visualize this, one can imagine multiplying a segment $\{j \leq z \leq j+1\}$ to every cell in Figure 3.

[20]To visualize this for $k = z$ as an example, one can imagine multiplying $\{0 \leq z \leq L_z\}$ to every cell in Figure 4 (Left). The direction $k'$ would be $x$ in the figure.

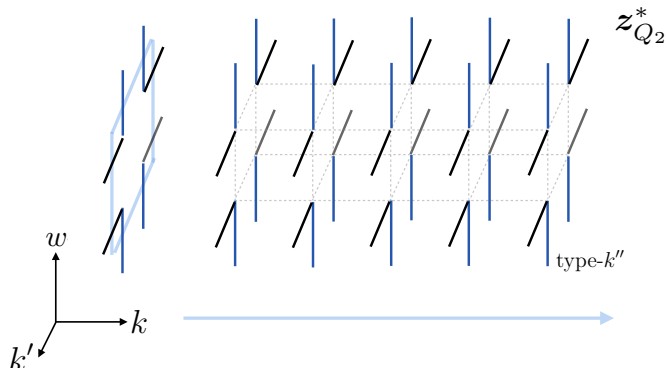

Figure 6: An example of symmetry generators in the cluster state $|\psi_C^{(\text{XCM})}\rangle$ described by $\boldsymbol{z}_{Q_2}^*$. The three-dimensional picture obtained by projecting the four-dimensional structure with respect to the $k''$-direction is illustrated. The edges in the $w$-direction is of type $k''$.

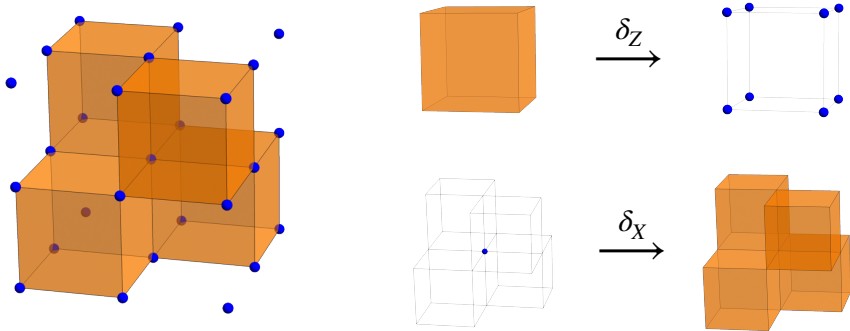

Figure 7: The checkerboard model.

Excitations that can be moved by symmetry generators are created by $Z(\boldsymbol{z})$ with $\boldsymbol{z} \in \boldsymbol{C}_{Q_2}$ satisfying $\boldsymbol{\delta z} = 0$ and by $Z(\boldsymbol{z}^*)$ with $\boldsymbol{z}^* \in \boldsymbol{C}_{Q_1}$ satisfying $\boldsymbol{\delta}^* \boldsymbol{z}^* = 0$. An example of $\boldsymbol{z} \in \boldsymbol{C}_w^{(x)}$ with $\boldsymbol{\delta z} = 0$ is a straight line in the $w$-direction at fixed $(x, y, z)$. An example of $\boldsymbol{z}^* \in \boldsymbol{C}_{xw}$ with $\boldsymbol{\delta}^* \boldsymbol{z}^* = 0$ is a straight line in the $y$-direction at fixed $(x, z, w)$, or in the $z$-direction at fixed $(x, y, w)$.

### 2.4.3 Checkerboard model

We consider a cubic lattice and group the cubes into shaded and unshaded ones as in Figure 7. The Hilbert space of the checkerboard model is given by qubits placed at vertices of the lattice. The Hamiltonian is

$$H_{\text{CBM}} = -\sum_c \prod_{v \subset c} X_v - \sum_c \prod_{v \subset c} Z_v, \tag{95}$$

where the summations are over shaded cubes only. See Figure 7 for illustration.

The checkerboard model also admits a CSS chain complex description. Let us write the set of vertices as $\Delta_0$. We double the set of shaded cubes and define $\Delta_{xyz,Z}^{(s)}$ and $\Delta_{xyz,X}^{(s)}$ as independent cells. We have a chain complex over the cells above:

$$0 \longrightarrow C_{xyz,Z}^{(s)} \xrightarrow{\delta_Z} C_0 \xrightarrow{\delta_X} C_{xyz,X}^{(s)} \longrightarrow 0. \tag{96}$$

The differentials are set as follows. For $c \in \Delta_{xyz,Z}^{(s)}$, $\delta_Z c$ is the sum of 8 vertices at the corners of $c$. For $v = \{(x, y, z)\} \in \Delta_0$, $\delta_X v$ is the sum of 4 shaded cubes adjacent to $(x, y, z)$. Note that $\delta_X \circ \delta_Z = 0$ (i.e., it is nilpotent). The dual boundary operator $\delta_i^*$ is obtained as the transpose of $\delta_i$,

$$0 \longrightarrow C_{xyz,X}^{(s)} \xrightarrow{\delta_X^*} C_0 \xrightarrow{\delta_Z^*} C_{xyz,Z}^{(s)} \longrightarrow 0\,. \tag{97}$$

The dual boundary operator is also nilpotent, i.e., $\delta_Z^* \circ \delta_X^* = 0$.

Global symmetries are generated by $Z(z)$ and $X(z^*)$ $(z, z^* \in C_0)$ such that $\delta_X z = \delta_Z^* z^* = 0$. An example of both $z$ and $z^*$ is a straight line in the $x$-, $y$-, or $z$-direction, giving $Z(z)$ and $X(z^*)$ as symmetry generators.[21]

Following the general construction, we obtain the foliated chain complex

$$0 \longrightarrow C_{xyzw}^{(s)} \xrightarrow{\delta} \underbrace{C_{xyz}^{(s)} \oplus C_w}_{=C_{Q_1}} \xrightarrow{\delta} \underbrace{C_0 \oplus C_{xyzw}^{(s)}}_{=C_{Q_2}} \xrightarrow{\delta} C_{xyz}^{(s)} \longrightarrow 0\,, \tag{98}$$

defined using a 4-dimensional hypercubic lattice. The stabilizers for the cluster state $|\psi_C^{(\mathrm{CBM})}\rangle$, the symmetry generators, and excitations can be described by cycles and dual cycles as in the previous cases.

An excitation is supported by a cycle $z_{Q_2} \in C_{Q_2}$ with $\delta z_{Q_2} = 0$ or a dual cycle $z_{Q_1}^* \in C_{Q_1}$ with $\delta^* z_{Q_1}^* = 0$. Examples of a cycle $z_{Q_2}$ are

- a straight line in the $x$-, $y$-, or $z$-direction formed by vertices,

- a straight line in the $w$-direction formed by shaded hypercubes, and more generally

- a curve in a $xw$-, $yw$-, or $zw$-plane formed by vertices and shaded hypercubes in the $w$-direction.

Examples of a dual cycle $z_{Q_1}^*$ are

- a straight line in the $w$-direction formed by shaded cubes in the $xyz$-directions,

- a straight line in the $x$-, $y$-, or $z$-direction formed by edges in the $w$-direction, and more generally

- a curve in a $xw$-, $yw$-, or $zw$-directions formed by shaded cubes in the $xyz$-directions and by edges in the $w$-direction.

A symmetry generator of the foliated cluster state is supported by a cycle $z_{Q_1} \in C_{Q_1}$ with $\delta z_{Q_1} = 0$ or a dual cycle $z_{Q_2}^* \in C_{Q_2}$ with $\delta^* z_{Q_2}^* = 0$. Examples of a cycle $z_{Q_1}$ are

- a plane in the $xy$-, $xz$-, or $yz$-directions formed by shaded cubes in the $xyz$-directions,

- a plane extended in the $xw$-, $yw$-, or $zw$-directions formed by edges in the $w$-direction, and more generally

- a straight line in the $k$-direction times a curve in the $k'w$-directions $(k' \neq k)$ formed by shaded cubes in the $xyz$-directions and by edges in the $w$-direction.

Examples of a dual cycle $z_{Q_2}^*$ are

- a plane in the $xw$-, $yw$-, or $zw$-directions formed by vertices,

- a plane in the $xy$-, $xz$-, or $yz$-directions formed by shaded hypercubes, and more generally

- a straight line in the $k$-direction times a curve in the $k'w$-plane $(k' \neq k)$ formed by vertices and shaded hypercubes.

---

[21] See [67] for the fractons, lineons, and planons created by $Z(c)$ and $X(c^*)$ $(c, c^* \in C_0)$ with $\partial_X c \neq 0$, $\partial_Z^* c^* \neq 0$.

# 3 Anomaly inflow for CSS codes

In general, the partition function of a quantum field theory (QFT) with symmetries is a functional of the background gauge fields coupled to the symmetries. The ('t Hooft) anomaly of a QFT is the non-invariance of the partition function under gauge transformations of the background fields [68]. The background fields are Poincaré dual to the world-volumes of the symmetry defects, and their gauge transformation is a local deformation of the world-volumes. It has been known that the anomalies of the boundary and bulk systems can be described in terms of symmetry defects. In the case of the Wegner models (more precisely generalizations of the toric code) the equality of the anomalous gauge transformations, i.e., the anomaly inflow, was demonstrated explicitly in [45]. Here we study the anomaly inflow for a general CSS code.

The basic manifestation of anomaly is the fact that the logical operators obey non-trivial commutation relations.[22] As in Section 2.3.1, we consider the foliated cluster state defined on $0 \leq w \leq L_w$ with boundaries at $w = 0$ and $w = L_w$. The operators $X(\boldsymbol{z}_2^*)$ and $X(\boldsymbol{z}_1^{\text{rel}})$ represent space-like defects.

## 3.1 Descriptions of defects by a chain complex

To describe general defects in a spacetime with boundary, we consider the following set-up. For the foliated cluster state, with $0 \leq w \leq L_w$, qubits are placed at

$$\sigma_q \otimes \{j\}, \qquad \sigma_Z \otimes \{j\}, \tag{99}$$

for $j = 0, 1, \ldots, L_w$, and

$$\sigma_q \otimes [j, j+1], \qquad \sigma_X \otimes [j, j+1], \tag{100}$$

for $j = 0, 1, \ldots, L_w - 1$, with $\sigma_q \in \Delta_q$, $\sigma_Z \in \Delta_Z$, and $\sigma_X \in \Delta_X$. As in Section 2.3.2, we model the time by a circle $M_\tau$ and the chain complex $C_\bullet(M_\tau)$ and the spacetime by the triple product of complexes $C_{\text{CSS}} \otimes C_\bullet(M_w) \otimes C_\bullet(M_\tau)$. The product complex includes a portion

$$\ldots \xrightarrow{\mathbf{d} = \delta \otimes \text{id} + \text{id} \otimes \partial} \boldsymbol{C}_{Q_1} \otimes C_0(M_\tau) \ \oplus \ \boldsymbol{C}_{Q_2} \otimes C_1(M_\tau) \xrightarrow{\mathbf{d}' = \delta \otimes \text{id} + \text{id} \otimes \partial} \ldots, \tag{101}$$

where we recall that $\boldsymbol{C}_{Q_1} = C_Z \otimes C_0(M_w) \oplus C_q \otimes C_1(M_w)$ and $\boldsymbol{C}_{Q_2} = C_q \otimes C_0(M_w) \oplus C_X \otimes C_1(M_w)$. The set-up is almost the same as that of Section 2.3.2, but we propose that the world-volume of the first type of defect is now given by a relative cycle $\boldsymbol{z}_{\text{rel}} \in \boldsymbol{C}_{Q_1} \otimes C_0(M_\tau) \ \oplus \ \boldsymbol{C}_{Q_2} \otimes C_1(M_\tau)$ such that

$$\mathbf{d}' \boldsymbol{z}_{\text{rel}} = \boldsymbol{z} \otimes \{w = 0\} + \boldsymbol{z}' \otimes \{w = L_w\}, \tag{102}$$

with

$$\boldsymbol{z}, \boldsymbol{z}' \in C_q \otimes C_0(M_\tau) \oplus C_X \otimes C_1(M_\tau), \tag{103}$$

and $(\delta \otimes \text{id} + \text{id} \otimes \delta)\boldsymbol{z} = (\delta \otimes \text{id} + \text{id} \otimes \delta)\boldsymbol{z}' = 0$. For the second type of defect, we propose that its world-volume is given by a chain (dual cycle) $\boldsymbol{z}^* \in \boldsymbol{C}_{Q_1} \otimes C_0(M_\tau) \ \oplus \ \boldsymbol{C}_{Q_2} \otimes C_1(M_\tau)$ such that $\mathbf{d}^* \boldsymbol{z}^* = 0$.

Let us decompose the relative cycle $\boldsymbol{z}_{\text{rel}}$ into space- and time-like parts:

$$\boldsymbol{z}_{\text{rel}} = \sum_{k=0}^{L_\tau - 1} \boldsymbol{c}_{Q_1}^{(k)} \otimes \{k\} + \sum_{k=0}^{L_\tau - 1} \boldsymbol{z}_{Q_2, \text{rel}}^{(k)} \otimes [k, k+1]. \tag{104}$$

---

[22]More generally, an anomaly implies that a group acts on the Hilbert space in a projective, rather than genuine, representation. We emphasize that the anomaly is not a property of the Hamiltonian or the Lagrangian.

They must obey

$$\sum_{k=0}^{L_\tau-1} \delta c_{Q_1}^{(k)} \otimes \{k\} + \sum_{k=0}^{L_\tau-1} \delta z_{Q_2,\text{rel}}^{(k)} \otimes [k,k+1] + \sum_{k=0}^{L_\tau-1} z_{Q_2,\text{rel}}^{(k)} \otimes (\{k\}+\{k+1\})$$
$$(= \mathbf{d}'\mathbf{z}_{\text{rel}}) = z \otimes \{w=0\} + z' \otimes \{w=L_w\}. \tag{105}$$

Writing $z = \sum_k c_q^{(k)} \otimes \{k\} + \sum_k z_X^{(k)} \otimes [k,k+1]$ with $\delta_X c_q^{(k)} = z_X^{(k)} - z_X^{(k-1)},$[23] and similarly for $z'$, we get

$$\delta z_{Q_2,\text{rel}}^{(k)} = z_X^{(k)} \otimes \{w=0\} + z_X'^{(k)} \otimes \{w=L_w\}, \tag{106}$$

$$z_{Q_2,\text{rel}}^{(k)} = z_{Q_2,\text{rel}}^{(k-1)} + \delta c_{Q_1}^{(k)} + c_q^{(k)} \otimes \{w=0\} + c_q'^{(k)} \otimes \{w=L_w\}. \tag{107}$$

In particular, $\{z_{Q_2,\text{rel}}^{(k)} | 1 \le k \le L_\tau - 1\}$ is determined from $z_{Q_2,\text{rel}}^{(0)}$, $\{c_{Q_1}^{(k)} | 0 \le k \le L_\tau - 1\}$, $\{c_q^{(k)} | 0 \le k \le L_\tau - 1\}$, and $\{c_q'^{(k)} | 0 \le k \le L_\tau - 1\}$.

Let us also decompose the dual cycle $\mathbf{z}^*$ into space- and time-like parts:

$$\mathbf{z}^* = \sum_{k=0}^{L_\tau-1} z_{Q_1}^{*(k)} \otimes \{k\} + \sum_{k=0}^{L_\tau-1} c_{Q_2}^{*(k+1/2)} \otimes [k,k+1]. \tag{108}$$

They must obey

$$\sum_{k=0}^{L_\tau-1} \delta^* z_{Q_1}^{*(k)} \otimes \{k\} + \sum_{k=0}^{L_\tau-1} z_{Q_1}^{*(k)} \otimes ([k-1,k]+[k,k+1]) + \sum_{k=0}^{L_\tau-1} \delta^* c_{Q_2}^{*(k+1/2)} \otimes [k,k+1] (= \mathbf{d}^*\mathbf{z}^*) = 0. \tag{109}$$

It follows that

$$\delta^* z_{Q_1}^{*(k)} = 0, \qquad z_{Q_1}^{*(k+1)} = z_{Q_1}^{*(k)} + \delta^* c_{Q_2}^{*(k+1/2)}. \tag{110}$$

Thus $\{z_{Q_1}^{*(k)} | 1 \le k \le L_\tau\}$ is determined from $z_{Q_1}^{*(0)}$ and $\{c_{Q_2}^{*(k+1/2)} | 0 \le k \le L_\tau - 1\}$.

We note that $z_{Q_1,\text{rel}}^{(0)}$ and $z_{Q_2}^{*(0)}$ specify the initial state, which we implement by appropriate projections. The chains instruct us to

$$\text{insert} \begin{cases} X(c_{Q_1}^{(k)}), Z\left(c_q^{(k)} \otimes \{w=0\}\right), \text{ and } Z\left(c_q'^{(k)} \otimes \{w=L_w\}\right), & \text{at } \tau = k, \\ X\left(c_{Q_2}^{*(k+1/2)}\right), & \text{at } \tau = k+1/2. \end{cases} \tag{111}$$

See Figure 8 for illustration.

## 3.2 The bulk partition function in the presence of boundary

Even in the presence of boundaries, we define the bulk partition function as

$$Z_{\text{bulk}}^{\text{CSS}}[\mathbf{z}_{\text{rel}}, \mathbf{z}^*] := \text{Tr}\left[\mathcal{O}_{\text{inserted}} \mathbb{P}(z_{Q_2,\text{rel}}^{(0)}, z_{Q_1}^{*(0)})\right] = \langle \mathcal{E} | \mathcal{O}_{\text{inserted}} | \mathcal{E} \rangle, \tag{112}$$

with the time-ordered operator

$$\mathcal{O}_{\text{inserted}} = \prod_{k=1}^{L_\tau} X(c_{Q_1}^{(k)}) Z(c_q^{(k)} \otimes \{w=0\}) Z(c_q'^{(k)} \otimes \{w=L_w\}) X(c_{Q_2}^{*(k-1/2)}), \tag{113}$$

---

[23] $\{z_X^{(k)} | 1 \le k \le L_\tau - 1\}$ is determined from $z_X^{(0)}$ and $\{c_q^{(k)} | 0 \le k \le L_\tau - 1\}$.

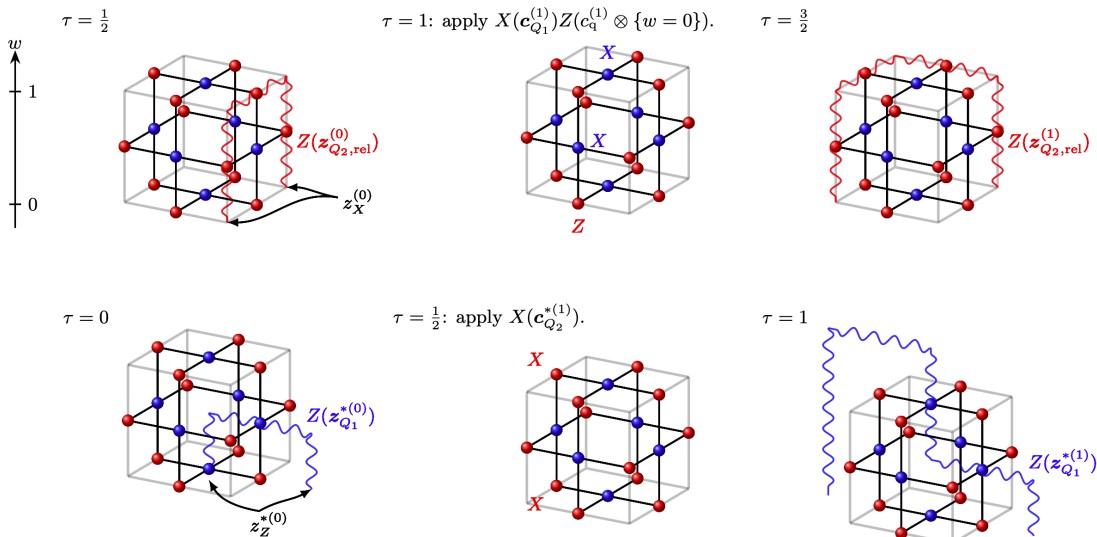

Figure 8: Evolution of excitations in the foliated cluster state due to insertion of defects. We take the toric code as an example of CSS codes, so the foliated CSS chain complex describes the RBH state; the chain $C_{Z,w}$ is the 3-chain, $C_{Q_1}$ the 2-chain, $C_{Q_2}$ the 1-chain, and $C_X$ the 0-chain. (Left column) Initially, we have a set of excitations described by $Z(z_{Q_2,\mathrm{rel}}^{(0)})$ (red curvy lines) and $Z(z_{Q_1}^{*(0)})$ (blue curvy lines), which terminate on the boundary at $z_X^{(0)} \otimes \{w = 0\}$ and $z_Z^{*(0)} \otimes \{w = 0\}$, respectively. (Middle column) We can move the excitations by applying operators as instructed in (111). (Right column) Due to the stabilizers of the RBH model, the new excitations become supported on deformed cycles.

where $k$ increases from right to left, the initial state

$$|\mathcal{E}\rangle := Z(z_{Q_2,\mathrm{rel}}^{(0)})Z(z_{Q_1}^{*(0)})|\psi_{\mathcal{C}}^{(\mathrm{CSS})}\rangle, \tag{114}$$

and the projection operator

$$\mathbb{P}(z_{Q_2,\mathrm{rel}}^{(0)}, z_{Q_1}^{*(0)}) = |\mathcal{E}\rangle\langle\mathcal{E}|. \tag{115}$$

We again find that the partition function is given in terms of an intersection number:

$$Z_{\mathrm{bulk}}^{\mathrm{CSS}}[\mathbf{z}_{\mathrm{rel}}, \mathbf{z}^*] = (-1)^{\#(\mathbf{z}_{\mathrm{rel}} \cap \mathbf{z}^*)}. \tag{116}$$

This is not gauge invariant due to the presence of boundaries. Under the gauge transformation of $\mathbf{z}_{\mathrm{rel}}$

$$\mathbf{z}_{\mathrm{rel}} \to \mathbf{z}_{\mathrm{rel}} + \mathbf{dc} + \mathrm{c} \otimes \{w = 0\} + \mathrm{c}' \otimes \{w = L_w\}, \tag{117}$$

with $\mathbf{c} \in C_{Z,w} \otimes C_0(M_\tau) \oplus C_{Q_1} \otimes C_1(M_\tau)$, $c, c' \in C_Z \otimes C_0(M_\tau) \oplus C_q \otimes C_1(M_\tau)$, the partition function changes as

$$Z_{\mathrm{bulk}}^{\mathrm{CSS}}[\mathbf{z}_{\mathrm{rel}}, \mathbf{z}^*] \to (-1)^{\#((\mathrm{c} \otimes \{w=0\} + \mathrm{c}' \otimes \{w=L_w\}) \cap \mathbf{z}^*)} Z_{\mathrm{bulk}}^{\mathrm{CSS}}[\mathbf{z}_{\mathrm{rel}}, \mathbf{z}^*]. \tag{118}$$

On the other hand, under the gauge transformation of $\mathbf{z}^*$

$$\mathbf{z}^* \to \mathbf{z}^* + \mathbf{d}^*\mathbf{c}^*, \tag{119}$$

with $\mathbf{c}^* \in C_{Q_2} \otimes C_0(M_\tau) \oplus C_X \otimes C_1(M_\tau)$, the partition function changes as

$$Z_{\mathrm{bulk}}^{\mathrm{CSS}}[\mathbf{z}_{\mathrm{rel}}, \mathbf{z}^*] \to (-1)^{\#((\mathrm{z} \otimes \{w=0\} + \mathrm{z}' \otimes \{w=L_w\}) \cap \mathbf{c}^*)} Z_{\mathrm{bulk}}^{\mathrm{CSS}}[\mathbf{z}_{\mathrm{rel}}, \mathbf{z}^*]. \tag{120}$$

### 3.3 Variations of the boundary partition function

As introduced earlier, we write the $w = 0$ part of the chain $\mathbf{d}'\mathbf{z}_{\mathrm{rel}}$ as $\mathbf{z} = \sum_k c_{\mathrm{q}}^{(k)} \otimes \{k\} + \sum_k z_X^{(k)} \otimes [k, k+1]$ with

$$\delta_X c_{\mathrm{q}}^{(k)} = z_X^{(k)} - z_X^{(k-1)}. \tag{121}$$

Let us write the dual cycle at $w = 0$ as $\mathbf{z}^* = \sum_k z_Z^{*(k)} \otimes \{k\} + \sum_k c_{\mathrm{q}}^{*(k+1/2)} \otimes [k, k+1]$ with

$$\delta_Z^* c_{\mathrm{q}}^{*(k-1/2)} = z_Z^{*(k)} - z_Z^{*(k-1)}. \tag{122}$$

We define the boundary partition function for a CSS code at the spatial boundary $w = 0$ as

$$Z_{\mathrm{bdry}}^{\mathrm{CSS}}[\mathbf{z}, \mathbf{z}^*] := \mathrm{Tr}\left[ Z\left(c_{\mathrm{q}}^{(L_\tau)}\right) X\left(c_{\mathrm{q}}^{*(L_\tau - 1/2)}\right) \dots Z\left(c_{\mathrm{q}}^{(1)}\right) X\left(c_{\mathrm{q}}^{*(1/2)}\right) \mathbb{P}\left(z_X^{(0)}, z_Z^{*(0)}\right) \right], \tag{123}$$

where $\mathbb{P}(z_X^{(0)}, z_Z^{*(0)})$ is the projector that specifies the initial configuration of excitations in the CSS code state at $\tau = 0$,

$$\mathbb{P}(z_X^{(0)}, z_Z^{*(0)}) = \prod_{\sigma_\alpha \in \Delta_X} \frac{1 + (-1)^{a(z_X^{(0)}; \sigma_\alpha)} X(\delta_X^* \sigma_\alpha)}{2} \prod_{\sigma_\beta \in \Delta_Z} \frac{1 + (-1)^{\#(\sigma_\beta \cap z_Z^{*(0)})} Z(\delta_Z \sigma_\beta)}{2}, \tag{124}$$

and the time-ordered operator insertions inside the trace describe the motion of excitations in spacetime according to the defects. One can regard the chain $c_{\mathrm{q}}^{(k)}$ as the location of string operators which move excitations on $z_X^{(k)}$ at time interval $[k-1, k]$ to those on $z_X^{(k+1)}$ at time interval $[k, k+1]$. Similarly, the chain $c_{\mathrm{q}}^{*(k-1/2)}$ describes the location of string operators which move excitations on $z_Z^{*(k-1)}$ at time $k-1$ to those on $z_Z^{*(k)}$ at time $k$. When both of the defects are absent the partition function gives us the ground state degeneracy of the CSS code.

The boundary partition function defined above may pick up a phase when the defects undergo gauge transformations, which signals the anomaly. The tensor product of chain complexes $C_{\mathrm{CSS}} \otimes C_\bullet(M_\tau)$ includes a portion

$$\dots \longrightarrow C_Z \otimes C_0(M_\tau) \oplus C_{\mathrm{q}} \otimes C_1(M_\tau) \xrightarrow{\mathrm{d} = \delta \otimes \mathrm{id} + \mathrm{id} \otimes \partial} C_{\mathrm{q}} \otimes C_0(M_\tau) \oplus C_X \otimes C_1(M_\tau) \longrightarrow \dots \tag{125}$$

Here, $\delta$ is $\delta_Z$ when acting on $C_Z$ and $\delta_X$ when acting on $C_{\mathrm{q}}$. We find the following anomaly in gauge transformations:

$$\frac{Z_{\mathrm{bdry}}^{\mathrm{CSS}}[\mathbf{z} + \mathrm{dc}_{\mathrm{gauge}}, \mathbf{z}^*]}{Z_{\mathrm{bdry}}^{\mathrm{CSS}}[\mathbf{z}, \mathbf{z}^*]} = (-1)^{\#(c_{\mathrm{gauge}} \cap \mathbf{z}^*)}, \tag{126}$$

$$\frac{Z_{\mathrm{bdry}}^{\mathrm{CSS}}[\mathbf{z}, \mathbf{z}^* + \mathrm{d}^* c_{\mathrm{gauge}}^*]}{Z_{\mathrm{bdry}}^{\mathrm{CSS}}[\mathbf{z}, \mathbf{z}^*]} = (-1)^{\#(\mathbf{z} \cap c_{\mathrm{gauge}}^*)}, \tag{127}$$

where $c_{\mathrm{gauge}} \in C_Z \otimes C_0(M_\tau) \oplus C_{\mathrm{q}} \otimes C_1(M_\tau)$ and $c_{\mathrm{gauge}}^* \in C_{\mathrm{q}} \otimes C_0(M_\tau) \oplus C_X \otimes C_1(M_\tau)$. We show the first equality for cases with $c_{\mathrm{gauge}} = \sigma_\beta \otimes \{k\}$ and $c_{\mathrm{gauge}} = \sigma_{\mathrm{q}} \otimes [k, k+1]$. The relation with general $c_{\mathrm{gauge}}$ follows from linearity of the intersection pairing, and the second equality can be shown analogously. For the first case with the space-like gauge transformation, we get the following transformation of the partition function:

$$\mathrm{Tr}\left[ \dots Z(c_{\mathrm{q}}^{(k+1)}) X(c_{\mathrm{q}}^{*(k+1/2)}) Z(c_{\mathrm{q}}^{(k)}) X(c_{\mathrm{q}}^{*(k-1/2)}) \dots \mathbb{P}(z_X^{(0)}, z_Z^{*(0)}) \right]$$

$$\xrightarrow{\mathbf{z} \to \mathbf{z} + \mathrm{dc}_{\mathrm{gauge}}} \mathrm{Tr}\left[ \dots Z(c_{\mathrm{q}}^{(k+1)}) X(c_{\mathrm{q}}^{*(k+1/2)}) Z(\delta_Z \sigma_\beta) Z(c_{\mathrm{q}}^{(k)}) X(c_{\mathrm{q}}^{*(k-1/2)}) \dots \mathbb{P}(z_X^{(0)}, z_Z^{*(0)}) \right]. \tag{128}$$

We note that the eigenvalue of $Z(\delta_Z \sigma_\beta)$ on the right hand side is determined by whether there is an excitation at $\sigma_\beta \otimes \{k\}$, and it is given by $(-1)^{\#(\sigma_\beta \cap z_Z^{*(k)})} = (-1)^{\#(c_{\text{gauge}} \cap z^*)}$. In the second case with the time-like gauge transformation, we get

$$\text{Tr}\Big[\ldots Z(c_q^{(k+1)}) X(c_q^{*(k+1/2)}) Z(c_q^{(k)}) X(c_q^{*(k-1/2)}) \ldots \mathbb{P}(z_X^{(0)}, z_Z^{*(0)})\Big]$$

$$\xrightarrow{z \to z + dc_{\text{gauge}}} \text{Tr}\Big[\ldots Z(c_q^{(k+1)}) Z(\sigma_q) X(c_q^{*(k+1/2)}) Z(\sigma_q) Z(c_q^{(k)}) X(c_q^{*(k-1/2)}) \ldots \mathbb{P}(z_X^{(0)}, z_Z^{*(0)})\Big], \qquad (129)$$

and the R.H.S. is equal to the original partition function up to the phase $(-1)^{\#(\sigma_q \cap c_q^{*(k+1/2)})} = (-1)^{\#(c_{\text{gauge}} \cap z^*)}$. Thus the anomalous transformations (126) and (127) are shown for the boundary at $w = 0$. Similar transformations occur on the other boundary $w = L_w$. They are identical to the anomalous variations of the bulk partition function in (118) and (120).

## 3.4 Relation between the bulk and the boundary systems

We now argue that our definitions of the bulk and the boundary partition functions are natural by showing that the boundary system is obtained by measuring the bulk in the $X$-basis.

We defined the bulk partition function (112) in terms of the excited state $|\mathcal{E}\rangle$ in (114) and the time evolution operator $\mathcal{O}_{\text{inserted}}$ in (113) for defects. Now, let us consider the states

$$|\Psi_0\rangle_{\text{bdry}} := \langle +|' |\mathcal{E}\rangle, \qquad |\Psi\rangle_{\text{bdry}} := \langle +|' \mathcal{O}_{\text{inserted}} |\mathcal{E}\rangle, \qquad (130)$$

where $\langle +|'$ is the product of $\langle +|$ states on all the qubits of the foliated cluster state (those on $\Delta_{Q_1} \cup \Delta_{Q_2}$) except at those at $\sigma_i \otimes \{w = 0\}$ and $\sigma_i \otimes \{w = L_w\}$ for $\sigma_i \in \Delta_q$. Then

$$|\Psi\rangle_{\text{bdry}} = \mathcal{O}_{w=0} \mathcal{O}_{w=L_w} |\Psi_0\rangle_{\text{bdry}}, \qquad (131)$$

with

$$\mathcal{O}_{w=0} := Z(c_q^{(L_\tau)}) X(c_q^{*(L_\tau - 1/2)}) \ldots Z(c_q^{(1)}) X(c_q^{*(1/2)}), \qquad (132)$$

where we omitted "$\otimes \{w = 0\}$" for simplicity, and with $\mathcal{O}_{w=L_w}$ similarly defined. The operator $\mathcal{O}_{w=0}$ is nothing but what appears in the boundary partition function (123). The state $|\Psi_0\rangle_{\text{bdry}}$ is characterized by the eigenvalue equations [23, 24] (we omit the eigenstate $|\Psi_0\rangle_{\text{bdry}}$ in the equations and only indicate the eigenvalues)

$$X(\delta_X^* \sigma_\alpha \otimes \{w = 0\}) = (-1)^{a(z_X^{(0)}; \sigma_\alpha)}, \quad \text{similar equations for } w = L_w,$$
$$Z(\delta_Z \sigma_\beta \otimes \{w = 0\}) = (-1)^{a(z_Z^{*(0)}; \sigma_\beta)}, \quad \text{similar equations for } w = L_w, \qquad (133)$$

for $\sigma_\alpha \in \Delta_X$, $\sigma_\beta \in \Delta_Z$, and

$$X(z_q^* \otimes \{w = 0\}) X(z_q^* \otimes \{w = L_w\}) = (-1)^{\#(z_{Q_2,\text{rel}}^{(0)} \cap z_q^* \otimes [-\frac{1}{2}, L_w + \frac{1}{2}])},$$
$$Z(z_q \otimes \{w = 0\}) Z(z_q \otimes \{w = L_w\}) = (-1)^{\#(z_{Q_1}^{*(0)} \cap z_q \otimes [0, L_w])}, \qquad (134)$$

for $z_q \in \text{Ker}\, \delta_X$ and $z_q^* \in \text{Ker}\, \delta_Z^*$. The equations (133) imply that the state $|\Psi_0\rangle_{\text{bdry}}$ is in the subspace specified by the projector (124) and the corresponding projector for $w = L_w$. On the other hand, the equations (134) imply that the state $|\Psi_0\rangle_{\text{bdry}}$ is a generalization of the Bell state and has long-range entanglement between the two boundaries at $w = 0$ and $w = L_w$. Other states in the subspace specified by the projector can be obtained by applying logical operators on one boundary. Thus the boundary system is obtained from the bulk by $X$-measurements and post-selection (to $|+\rangle'$). Moreover, (131) implies that the operator insertions in the bulk reduce to the desired ones on the boundary.

Taking the overlap with $\langle+|'$ in (130) corresponds to the measurement outcome such that every single-qubit measurement in the $X$-basis gives $X = +1$. Other outcomes also give states that are characterized by eigenvalue equations and therefore are Bell pairs [24].[24]

### 3.5 BF theory description of an arbitrary CSS code

We defined the partition function of a CSS code as a functional of defects in (123). We can derive a BF-type classical lattice model, i.e., a classical statistical model whose weight is the exponential of an action that looks like $S \sim \int b \wedge da$. For this, we replace the projector in (123) by the product of $L_\tau$ copies of it, commute each copy to the left through the $X$ and $Z$ insertions, rewrite each projector as a sum, and insert the completeness relation after each $X$ and $Z$. We get

$$
Z_{\text{bdry}}^{\text{CSS}}[z, z^*] = \frac{1}{2^{|\Delta_X| + |\Delta_Z|}} \sum_{a_{\text{q}}^{(k)}, b_{\text{q}}^{(r)}, a_Z^{(r)}, b_X^{(k)}} \exp\Bigg[ \pi i \sum_{k=1}^{L_\tau} \bigg( b_{\text{q}}^{(k-\frac{1}{2})} \cdot (a_{\text{q}}^{(k)} - a_{\text{q}}^{(k-1)}) + b_{\text{q}}^{(k-\frac{1}{2})} \cdot \delta_Z a_Z^{(k-\frac{1}{2})}
$$

$$
+ b_X^{(k)} \cdot \delta_X a_{\text{q}}^{(k)} + b_{\text{q}}^{(k-\frac{1}{2})} \cdot c_{\text{q}}^{(k)} + a_{\text{q}}^{(k)} \cdot c_{\text{q}}^{*(k+\frac{1}{2})} + a_Z^{(k-\frac{1}{2})} \cdot z_Z^{*(k)} + b_X^{(k)} \cdot z_X^{(k)} \bigg) \Bigg]. \quad (135)
$$

Here, we denoted the intersection number simply by a dot, and the summation is over the chains $a_{\text{q}}^{(k)}, b_{\text{q}}^{(r)} \in C_{\text{q}}$, $a_Z^{(r)} \in C_Z$, and , $b_X^{(k)} \in C_X$ with $k \sim k + L_\tau$, $r \sim r + L_\tau$. The summand is invariant under

$$
\begin{aligned}
a_{\text{q}}^{(k)} &\to a_{\text{q}}^{(k)} + \delta_Z \alpha_Z^{(k)}, \\
a_Z^{(r)} &\to a_Z^{(r)} + \alpha_Z^{(r+1/2)} - \alpha_Z^{(r-1/2)}, \\
b_{\text{q}}^{(r)} &\to b_{\text{q}}^{(r)} + \delta_X^* \beta_X^{(r)}, \\
b_X^{(k)} &\to b_X^{(k)} + \beta_X^{(k+1/2)} - \beta_X^{(k-1/2)},
\end{aligned}
\quad (136)
$$

due to (121) and (122), where $\alpha_Z^{(k)}$ and $\beta_X^{(r)}$ are arbitrary. The "path integral" (135) generalizes topological lattice gauge theories in [47]. See, also, [71–73]. The gauge variation computed in Section 3.3 can be rephrased in terms of the classical lattice model if desired. The cycles z and z* represent gauge-invariant observables, and their correlation functions are given essentially by linking numbers, as follows from the anomalies in (126) and (127). The overall picture of anomaly inflow is then quite similar to [44, 74], where the continuum BF-theory description was used for models with subsystem symmetries.

## 4 Duality and strange correlators

In this section, we consider Kramers-Wannier dualities constructed with a cluster-state entangler, local measurements, and feedforwarded corrections. Interestingly, we show that the pair of the Kramers-Wannier duality operator and its reverse (KW† ∘ KW) becomes a sum over all possible subsystem symmetry generators, i.e., a projector onto the subsystem symmetric Hilbert space, exhibiting non-invertibility. Our calculation is a generalization of the result by Cao et al. [48] for 2d plaquette Ising model with subsystem symmetries, giving operational interpretations and extending to general CSS codes.

---

[24]In particular, the measurement-induced entanglement introduced in [69] and studied in [70] is log 2 times the number of logical qubits.

Then we consider for general CSS codes a generalized notion of strange correlators. We show that an overlap between a product state with a CSS code state (such as the toric code or the fractonic state) which is written using the aforementioned KW operator gives the classical partition function.

Further, we generalize the Kramers-Wannier-Wegner duality between classical partition functions using the foliated cluster state for general CSS codes. This gives a natural generalization of Wegner's duality and the self duality of the 3d anisotropic plaquette Ising model [46].

## 4.1 Kramers-Wannier duality

Let us consider the CSS complex (34). To discuss the Kramers-Wannier transformation, let us introduce new qubits on $\sigma_\beta \in \Delta_Z$. We define the entangler

$$\mathcal{U}_{CZ} = \prod_{\substack{\sigma_i \in \Delta_q \\ \sigma_\beta \in \Delta_Z}} CZ_{\sigma_\beta,\sigma_i}^{a(\delta_Z \sigma_\beta; \sigma_i)}, \tag{137}$$

and define the Kramers-Wannier transformation

$$\mathsf{KW} = \langle + |^{\Delta_q} \mathcal{U}_{CZ} | + \rangle^{\Delta_Z}. \tag{138}$$

We also define

$$\mathsf{KW}^\dagger = \langle + |^{\Delta_Z} \mathcal{U}_{CZ} | + \rangle^{\Delta_q}. \tag{139}$$

Then, we find the following relations:

$$\mathsf{KW} \cdot X(\sigma_i) = Z(\delta_Z^* \sigma_i) \cdot \mathsf{KW}, \tag{140}$$

$$X(\sigma_\beta) \cdot \mathsf{KW} = \mathsf{KW} \cdot Z(\delta_Z \sigma_\beta). \tag{141}$$

Namely, the Kramers-Wannier transformation operator converts between two quantum models:

$$\widetilde{H}_{\mathrm{CSS}} = -\sum_{\sigma_i \in \Delta_q} X(\sigma_i) - \lambda \sum_{\sigma_\beta \in \Delta_Z} Z(\delta_Z \sigma_\beta) \text{ with symmetry } X(z_q^*) \, (\delta_Z^* z_q^* = 0)$$

$$\xrightarrow[\mathsf{KW}^\dagger]{\mathsf{KW}} \quad \widetilde{H}_{\mathrm{CSS,dual}} = -\sum_{\sigma_i \in \Delta_q} Z(\delta_Z^* \sigma_i) - \lambda \sum_{\sigma_\beta \in \Delta_Z} X(\sigma_\beta) \text{ with symmetry } X(z_Z) \, (\delta_Z z_Z = 0). \tag{142}$$

The former Hamiltonian $\widetilde{H}_{\mathrm{CSS}}$ is symmetric under the transformation generated by $X(z_q^*)$ with $\delta_Z^* z_q^* = 0$, and the latter is symmetric under the transformation generated by $X(z_Z)$ with $\delta_Z z_Z = 0$. The adjoint operator $\mathsf{KW}^\dagger$ simply implements the reverse of the above transformations.

Furthermore, we find the following relations:

$$\mathsf{KW} \cdot X(z_q^*) = \mathsf{KW}, \quad \text{for } \delta_Z^* z_q^* = 0, \tag{143}$$

$$X(z_Z) \cdot \mathsf{KW} = \mathsf{KW}, \quad \text{for } \delta_Z z_Z = 0. \tag{144}$$

As is well known in the literature (see for example [5]), Kramers-Wannier transformations can be seen as gauging global symmetries. In our case, the symmetry $X(z_q^*)$ in the CSS code model can be seen as being gauged in the transformation by the operator $\mathsf{KW}$. On the other hand, the symmetry $X(z_Z)$ is emergent after the duality transformation, as indicated in (144).

Composition of the two duality operators exhibits a non-invertible fusion rule. Let us consider a product state in the $Z$ basis, $|u_Z\rangle = \bigotimes_{\sigma_\beta \in \Delta_Z} |a(u_Z; \sigma_\beta)\rangle_{\sigma_\beta}$ with $u_Z$ a chain generated by $\Delta_Z$. We define $|v_Z\rangle$ in the same manner. According to the definition of the controlled-$Z$ gate and regarding the qubits on $\Delta_Z$ as the controlling qubits, we find

$$\langle u_Z | \mathsf{KW} \circ \mathsf{KW}^\dagger | v_Z \rangle = \frac{1}{2^{|\Delta_Z|}} \langle + |^{\Delta_q} \prod_{\sigma_i \in \Delta_q} Z_{\sigma_i}^{a(\delta_Z \sigma_\beta; \sigma_i)(a(u_Z; \sigma_\beta) + a(v_Z; \sigma_\beta))} | + \rangle^{\Delta_q}$$

$$= \frac{1}{2^{|\Delta_Z|}} \prod_{\sigma_i \in \Delta_q} \delta_{\#(\sigma_i \cap \delta_Z(u_Z + v_Z))}^{\bmod 2}, \tag{145}$$

which implies

$$\mathsf{KW} \circ \mathsf{KW}^\dagger = \frac{1}{2^{|\Delta_Z|}} \sum_{\delta_Z z_Z = 0} X(z_Z). \tag{146}$$

The right hand side is a sum of all possible symmetry generators. Similarly, we find

$$\mathsf{KW}^\dagger \circ \mathsf{KW} = \frac{1}{2^{|\Delta_q|}} \sum_{\delta_Z^* z_q^* = 0} X(z_q^*). \tag{147}$$

The two relations above hold for a general CSS code, and they are the main results in this subsection. In Section 5.1.1, we discuss examples. We provide a discussion of the Kramers-Wannier transformation and the fusion rules for the Chamon model, a non-CSS code, in Appendix B. As can be seen from (143) and (147), KW annihilates an eigenstate of $X(z_q^*)$ with eigenvalue $-1$ for some $z_q^*$, and acts as an isomorphism between the symmetric subspaces (the space of states stabilized by $X(z_q^*)$ for all $z_q^*$ or by $X(z_Z)$ for all $z_Z$).

Finally, following the idea of Ref. [5, 75], we give a physically feasible procedure of implementing the gauging operation. The same argument below works for $\mathsf{KW}^\dagger$ as well. Given a state in a CSS code, which is (demanded to be) symmetric $X(z_q^*)|\Psi\rangle = |\Psi\rangle$, we consider performing the following procedure.

- Introduce ancilla qubits $|+\rangle^{\Delta_Z}$.

- Apply the entangler $\mathcal{U}_{CZ}$ in (137).

- Measure qubits on $\Delta_q$ in the $X$ basis.

- Counter the randomness of the measurement.

Let us explain the last step of the procedure above. The state we get after applying the entangler $\mathcal{U}_{CZ}$ is

$$|\Psi_{\text{pre}}\rangle = \mathcal{U}_{CZ} |\Psi\rangle |+\rangle^{\Delta_Z}, \tag{148}$$

and it is also symmetric:

$$X(z_q^*) |\Psi_{\text{pre}}\rangle = |\Psi_{\text{pre}}\rangle. \tag{149}$$

The measurement outcome can be expressed as $\langle + |^{\Delta_q} Z(c_q)$ with some chain $c_q \in C_q$. Note the following relation:

$$\langle + |^{\Delta_q} Z(c_q) |\Psi_{\text{pre}}\rangle = \langle + |^{\Delta_q} Z(c_q) X(z_q^*)^2 |\Psi_{\text{pre}}\rangle = (-1)^{\#(z_q^* \cap c_q)} \langle + |^{\Delta_q} Z(c_q) |\Psi_{\text{pre}}\rangle. \tag{150}$$

The first equality follows from the fact that $X(z_q^*)^2 = 1$ and the second equality follows by moving one of the $X(z_q^*)$ to the right and the other one to the left. This implies that $\#(z_q^* \cap c_q) = 0 \bmod 2$ for any cycle $\delta_Z^* z_q^* = 0$. Hence in particular $\#(\delta_X^* c_X \cap c_q) = 0 \bmod 2$ for any $c_X$. Therefore by Poincare duality $\#(c_X \cap \delta_X c_q) = 0 \bmod 2$, which implies $\delta_X c_q = 0 \bmod 2$, i.e., $c_q$ is a cycle. By the non-degeneracy of the intersection pairing of homologies (see Appendix E for a proof) the homology class of $c_q$ is trivial, $[c_q] = 0$. Therefore, $c_q = \delta_Z c_Z$ for some chain $c_Z \in C_Z$. Using the map $\mathsf{KW} = \langle + |^{\Delta_q} \mathcal{U}_{CZ} |+\rangle^{\Delta_Z}$, the Pauli operator $Z(c_q = \delta_Z c_Z)$ representing the general measurement outcome is transformed as $\mathsf{KW} Z(\delta c_Z) = X(c_Z) \mathsf{KW}$, and the operator $X(c_Z)$ can be canceled after the measurement.

## 4.2 Strange correlator and dualities

A strange correlator is defined as the overlap between a symmetry-protected topological state or a topologically ordered state and a product state [51, 52]. In principle, one could consider this overlap with operator insertions, as explored in [51] for symmetry-protected topological states. However, in this manuscript, we do not incorporate operator insertions while discussing strange correlators.

The strange correlator that we define in this manuscript can be interpreted as a lattice manifestation of symmetry topological field theory (SymTFT) or the sandwich construction, which has been explored extensively in recent literature [76–79]. SymTFT is described by a $(d+1)$-dimensional bulk theory, a $d$-dimensional topological boundary, and a $d$-dimensional dynamical boundary. The bulk theory is topological in nature, and the two boundary theories serve as boundary conditions for the bulk. The topological boundary specifies the global properties (those affected by gauging symmetries) of the system, while the dynamical boundary encodes the local dynamics. Since the bulk is topological, the topological boundary can be moved freely, making the overall system appear as a quasi-$d$-dimensional system.

Let us denote the non-trivial state, which is the ground state of a topological or fracton order, by $|\Psi\rangle$, and the product state by $|\omega\rangle$. The overlap $\langle \omega | \Psi \rangle$ can be interpreted as specifying the boundary conditions in the SymTFT framework in the following way:

1. Topological boundary condition specified by $|\Psi\rangle$,

2. Dynamical boundary condition specified by $|\omega\rangle$.

In this manuscript, the state $|\Psi\rangle$ refers to the ground states of CSS codes. Consequently, the bulk theory in the SymTFT/sandwich construction can be described by a BF theory. An illustration of this is provided in Figure 9.

Using $\mathsf{KW}$ in (139), we provide a generalization of strange correlators to general CSS codes. The state $\mathsf{KW}^\dagger |+\rangle^{\Delta_Z}$ is a ground state of a Hamiltonian whose terms are given by the stabilizers of the CSS code. Passing $X$ operators through $CZ$ operators and evaluating a wave function overlap gives us the following formula:

$$\mathbf{Z}_Z(K) = \mathcal{N} \times \langle \omega(K) | \mathsf{KW}^\dagger |+\rangle^{\Delta_Z}, \tag{151}$$

where

$$\mathbf{Z}_Z(K) = \sum_{\{s(\sigma_\beta)=\pm 1\}_{\sigma_\beta \in \Delta_Z}} \exp\Big[K \sum_{\sigma_i \in \Delta_q} s(\delta_Z^* \sigma_i)\Big], \tag{152}$$

$$\langle \omega(K)| = \bigotimes_{\sigma_i \in \Delta_q} \langle 0| e^{K X_{\sigma_i}}, \tag{153}$$

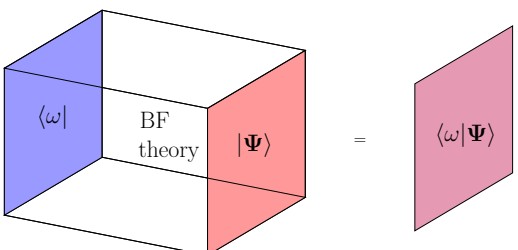

Figure 9: A pictorial illustration of the SymTFT/sandwich construction. On the left-hand side, we depict a quasi-$d$-dimensional system consisting of a bulk and two boundaries. The red boundary represents the topological (or reference) boundary, which we specify as the ground state $|\Psi\rangle$ of a CSS code. The blue boundary corresponds to the dynamical boundary, described by the product state $|\omega\rangle$. The bulk theory is represented by a BF theory. On the right-hand side, we show the resulting $d$-dimensional system, whose partition function is given by the overlap $\langle\omega|\Psi\rangle$.

and $\mathcal{N}$ is a normalization constant independent of the parameter $K$. Here, we defined $s(c_k) := \prod_{\sigma_k \in \Delta_k} s(\sigma_k)^{a(c_k;\sigma_k)}$ for a chain $c_k$ generated by $\Delta_k$. The exponent in the partition function is (proportional to) a classical Hamiltonian,

$$-\beta\mathcal{H} = K \sum_{\sigma_i \in \Delta_q} s(\delta_Z^* \sigma_i), \tag{154}$$

where $\beta$ is the inverse temperature. The Hamiltonian is symmetric under a transformation that flips spins on cycles $z_Z \in C_Z$ such that $\delta_Z z_Z = 0$. The general result above can be applied to CSS codes that describe topological orders [45, 58, 80], fracton models, etc. See Table 1 for a summary of examples. To the best of our knowledge, this is the first construction of a strange correlator for fracton models.

Now, we claim a duality between two partition functions that arise from the strange correlator we just described. We write the the space of logical operators as $\mathcal{L}^* = \mathrm{Ker}\,\delta_Z^* / \mathrm{Im}\,\delta_X^*$, and we denote the homology class represented by $\bullet$ as $[\bullet]$. We define $K^* = -\frac{1}{2}\log\tanh K$. Then, we claim the following duality of partition functions (see Appendix D for a proof):

$$\mathbf{Z}_Z(K) = \frac{2^{|\Delta_Z|}(\sinh 2K)^{|\Delta_q|/2}}{2^{|\Delta_X|}|\mathcal{L}^*|} \sum_{[\ell^*] \in \mathcal{L}^*} \mathbf{Z}_X^{\text{twisted}}(K^*, \ell^*), \tag{155}$$

where we introduced the twisted partition function given by

$$\mathbf{Z}_X^{\text{twisted}}(K^*, \ell^*) = \sum_{\{s_{\sigma_\alpha} = \pm 1\}_{\sigma_\alpha \in \Delta_X}} \exp\left(K^* \sum_{\sigma_i \in \Delta_q} (-1)^{\#(\sigma_q \cap \ell^*)} s(\delta_X \sigma_i)\right). \tag{156}$$

This duality can be thought of as changing the boundary condition in the SymTFT.

### 4.2.1 Example: 2d classical Ising model from the 2d toric code

An illuminating example, which we have already discussed in Section 1.2.2, is the case with the chain complex $C_2 \xrightarrow{\delta_Z} C_1 \xrightarrow{\delta_X} C_0$ defined on the two-dimensional periodic square lattice. Here $C_i$ denotes the group of $i$-chains. The partition function $\mathbf{Z}_Z(K)$ is then the classical 2d Ising model with spins introduced at every plaquette, where interactions take place over every edge with the coupling constant $K$. The twisted partition function $\mathbf{Z}_X^{\text{twisted}}(K^*, \ell^*)$ is a classical 2d Ising model with spins introduced at every vertex, where interactions take place over every

Table 1: Summary of setups in Kramers-Wannier transformation for fracton models seen as a CSS code. Here, $\Delta_k$ is the set of $k$-cells, and additional labels are used to distinguish between different cells. We provide the definitions of CSS chain complexes for the above models in Appendix A.

| | $\Delta_Z$ | $\Delta_q$ | $\Delta_X$ | $z_q^*$ | $z_Z$ | statistical model |
|---|---|---|---|---|---|---|
| 2d quantum plaquette Ising model (2d-qPIM) | plaquette $\Delta_2$ | vertex $\Delta_0$ | $\emptyset$ | rigid line | rigid line | 2d classical plaquette Ising model (2d-cPIM) |
| X-cube model (XCM) | cube $\Delta_3$ | edge $\Delta_1$ | three copies of vertices $\bigcup_{k=x,y,z}\Delta_0^{(k)}$ | "subsystem 1-form" | rigid plane | 3d classical plaquette Ising model (3d-cPIM) |
| Checkerboard model (CBM) | shaded cube $\Delta_{3,Z}^{(s)}$ | vertex $\Delta_0$ | shaded cube $\Delta_{3,X}^{(s)}$ | line | rigid plane | 3d classical tetrahedral Ising model (3d-cTIM) |
| Haah's code (HC) | cube $\Delta_{3,Z}$ | two copies of vertices $\Delta_0^R \cup \Delta_0^B$ | cube $\Delta_{3,X}$ | fractal | fractal | 3d clasical fractal Ising model (3d-cFIM) |

edge with the coupling constant $K^*$. The twists — i.e., the flip of the sign in terms in the classical Hamiltonian — in the latter partition function are introduced along logical operators of the underlying 2d toric code; $x$-directional edges along a non-contractible loop in the $y$-direction, or vice versa.

This construction generalizes to Wegner's models, and we provided a similar derivation of the dualities in our previous paper [45].

### 4.2.2 Example: 2d classical plaquette Ising model from 2d quantum plaquette Ising model

Another illuminating example of the strange correlator (151) is a mapping from the ground state of (2+1)d plaquette Ising model defined by the complex (71) to the 2d classical plaquette Ising model. We take an overlap between the ground state of the 2d qPIM $|\Psi_{\mathrm{qPIM}}\rangle := \mathsf{KW}^\dagger|+\rangle^{\Delta_2}$ on plaquettes (which is characterized by the stabilizer condition $\prod_{v\subset f} Z_v|\Psi_{\mathrm{qPIM}}\rangle = |\Psi_{\mathrm{qPIM}}\rangle$ and subsystem symmetries) and a product state $\langle\omega(K)| := \bigotimes_{v\in\Delta_0}\langle 0|e^{KX}$. We obtain

$$\mathbf{Z}_{\mathrm{2d\text{-}cPIM}}(K) = \mathcal{N} \times \langle\omega(K)|\Psi_{\mathrm{qPIM}}\rangle, \tag{157}$$

where $\mathbf{Z}_{\mathrm{2d\text{-}cPIM}}$ is the 2d classical plaquette Ising model,

$$\mathbf{Z}_{\mathrm{2d\text{-}cPIM}}(K) := \sum_{\{s_f=\pm 1\}} e^{K\sum_{v\in\Delta_0}\prod_{f\supset v}s_f}. \tag{158}$$

The exponent is a classical 2d Hamiltonian which consists of the product of spins at the four corners of a dual plaquette $v \simeq f^*$ in the dual lattice. The mapping is a relation between a fractonic ground state (although it is an unstable fracton in this specific case) with a classical model by taking an overlap with a product state, and can be viewed as a generalization of the so-called strange correlator [51, 52]. The classical Hamiltonian is symmetric under a flip of spins on a rigid line in $x$ or $y$ directions, each of which is an example of $z_Z$.

The duality of the equation (155) implies the relation

$$\mathbf{Z}_Z(K) = \frac{2^{|\Delta_2|}(\sinh 2K)^{|\Delta_0|/2}}{|\mathcal{L}^*|} \sum_{[\ell^*]\in\mathcal{L}^*} \exp\left(K^*\sum_{\sigma_0}(-1)^{\#(\sigma_0\cap\ell^*)}\right), \tag{159}$$

with $|\mathcal{L}^*| = 2^{L_x + L_y - 1}$. It states that the 2d classical plaquette Ising model is dual to the classical statistical model defined by the sum in the equation. We note that $\mathcal{L}^*$ is generated by horizontal and vertical straight lines with one relation, namely the sum of all such lines being zero.

### 4.2.3 Example: 3d classical tetrahedral Ising model from the checkerboard model

Another example of the strange correlator (151) is a mapping from the ground state of (3+1)d checkerboard model to the 3d classical tetrahedral Ising model. The CSS chain complex for the checkerboard model was given in (96) and illustrated in Figure 7. We write the lattice linear sizes as $L_x$, $L_y$, and $L_y$, each of which is even.

The partition function $\mathbf{Z}_Z(K)$ in (152), which is obtained as the strange correlator from a checkerboard ground state is the 3d classical tetrahedral Ising model (3d-cTIM). In the classical Hamiltonian of 3d-cTIM, spins live on the shaded cubes, and the interaction is defined as the product of four spins adjacent to every vertex. The duality (155) tells us that the partition function $\mathbf{Z}_Z(K)$ is proportional to the sum of the dual twisted partition functions $\mathbf{Z}_X^{\text{twisted}}(K^*, \ell^*)$ in (156). Interestingly, the geometric structure of the Hamiltonian terms is the exactly the same in both sides of the duality, as $\delta_X v$ and $\delta_Z^* v$ for $v \in \Delta_0 = \Delta_q$ look identical.

On the right hand side of (155), the twists are inserted along non-trivial logical operators in the checkerboard model: rigid lines in $x$, $y$, or $z$ directions. We have $2L_x + 2L_y + 2L_z - 6$ independent straight lines as logical operators [81].[25]

## 4.3 Strange correlator for foliated cluster states and dualities

We further introduce another type of strange correlators, now derived from foliated cluster states, and present dualities between them. The special case of the RBH model was explained in Section 1.2.3. We focus on the general construction in this subsection, and we give another concrete example in Section 5.2.1.

We begin by reminding readers that, given a CSS code, we have a set of foliated cells, a foliated chain complex $0 \xrightarrow{\delta} C_{Z,w} \xrightarrow{\delta} C_{Q_1} \xrightarrow{\delta} C_{Q_2} \xrightarrow{\delta} C_X \xrightarrow{\delta} 0$ with $C_{Q_1} = C_Z \oplus C_{q,w}$ and $C_{Q_2} = C_q \oplus C_{X,w}$, and the foliated cluster state $|\psi_C^{(\text{CSS})}\rangle$. It is a straightforward calculation to show the following relation:

$$\langle \Omega(J,K) | \psi_C^{(\text{CSS})} \rangle = 2^{-|\Delta_{Q_2}|} 2^{-|\Delta_{Q_1}|/2} \times \mathcal{Z}^{(\text{CSS})}(J,K), \tag{160}$$

where $\mathcal{Z}^{(\text{CSS})}$ is a partition function

$$\mathcal{Z}^{(\text{CSS})}(J,K) = \sum_{\{s(\sigma)=\pm 1\}_{\sigma \in \Delta_{Q_2}}} \exp\left(J \sum_{\sigma' \in \Delta_Z} s(\delta \sigma') + K \sum_{\sigma'' \in \Delta_{q,w}} s(\delta \sigma'')\right), \tag{161}$$

and $\langle \Omega(J,K)|$ is a product state

$$\langle \Omega(J,K)| = \bigotimes_{\sigma \in \Delta_{Q_2}} \langle + |_\sigma \bigotimes_{\sigma' \in \Delta_Z} \langle 0 |_{\sigma'} e^{J X_{\sigma'}} \bigotimes_{\sigma'' \in \Delta_{q,w}} \langle 0 |_{\sigma''} e^{K X_{\sigma''}}. \tag{162}$$

We claim that, with $J^* = -\frac{1}{2}\log\tanh J$, $K^* = -\frac{1}{2}\log\tanh K$ and up to contributions from non-trivial cycles discussed below, we have

$$\mathcal{Z}^{(\text{CSS})}(J,K) \sim \mathcal{N}' \times (\sinh J)^{|\Delta_Z|/2} (\sinh K)^{|\Delta_{q,w}|/2} \mathcal{Z}_{\text{dual}}^{(\text{CSS})}(J^*, K^*), \tag{163}$$

---

[25]As four parallel straight lines are related by a product of local $X$ stabilizers, there are $L_y + L_z - 1$ independent straight lines in the $x$ direction, etc. Hence, we have $2L_x + 2L_y + 2L_z - 3$ straight lines. However, there are three constraints among them: An $xy$ plane can be formed either as a sum of $x$-lines or of $y$-lines, and similar constraints for $yz$ and $zx$ planes. Therefore, we have $2L_x + 2L_y + 2L_z - 6$ independent straight lines as logical operators.

where $\mathcal{N}'$ is a normalization constant independent of $(J,K)$ and $\mathcal{Z}^{(\text{CSS})}_{\text{dual}}(J,K)$ is a dual statistical partition function given by

$$\mathcal{Z}^{(\text{CSS})}_{\text{dual}}(J,K) = \sum_{\{s(\boldsymbol{\sigma})=\pm 1\}_{\boldsymbol{\sigma}\in\Delta_{Z,w}}} \exp\Big(J\sum_{\boldsymbol{\sigma}'\in\Delta_Z} s(\boldsymbol{\delta}^*\boldsymbol{\sigma}') + K\sum_{\boldsymbol{\sigma}''\in\Delta_{\text{q},w}} s(\boldsymbol{\delta}^*\boldsymbol{\sigma}'')\Big). \tag{164}$$

Below, we derive a refined version of this duality.

We decompose the product state as

$$\langle\Omega(J,K)| = \langle +|^{\Delta_{Q_2}}\langle\omega(J,K)|, \qquad \langle\omega(J,K)| = \bigotimes_{\Delta_Z}\langle 0|e^{JX}\bigotimes_{\Delta_{\text{q},w}}\langle 0|e^{KX}. \tag{165}$$

We note the state given by

$$|\Phi^{(\text{CSS})}\rangle = \langle +|^{\Delta_{Q_2}}|\psi^{(\text{CSS})}_{\mathcal{C}}\rangle, \tag{166}$$

is stabilized by $\{Z(\boldsymbol{\delta}^*\boldsymbol{\sigma})\}_{\boldsymbol{\sigma}\in\Delta_{Q_2}}$ and $\{X(\boldsymbol{\delta}\boldsymbol{\sigma})\}_{\boldsymbol{\sigma}\in\Delta_{Z,w}}$ as well as the logical operator $X(\boldsymbol{z}_{Q_1})$ with $\boldsymbol{\delta}\boldsymbol{z}_{Q_1} = 0$.

We introduce a dual cluster state $|\psi^{(\text{CSS})*}_{\mathcal{C}}\rangle$ defined on $\Delta_{Q_1}\cup\Delta_{Z,w}$ which is a simultaneous $+1$ eigenstate of stabilizers $K(\boldsymbol{\sigma}) = X(\boldsymbol{\sigma})Z(\boldsymbol{\delta}\boldsymbol{\sigma})$ $(\boldsymbol{\sigma}\in\Delta_{Z,w})$, $K(\boldsymbol{\sigma}') = X(\boldsymbol{\sigma}')Z(\boldsymbol{\delta}^*\boldsymbol{\sigma}')$ $(\boldsymbol{\sigma}'\in\Delta_{Q_1})$. We note the state given by

$$|\Phi^{(\text{CSS})*}\rangle = \langle +|^{\Delta_{Z,w}}|\psi^{(\text{CSS})*}_{\mathcal{C}}\rangle, \tag{167}$$

is stabilized by $\{Z(\boldsymbol{\delta}\boldsymbol{\sigma})\}_{\boldsymbol{\sigma}\in\Delta_{Z,w}}$ and $\{X(\boldsymbol{\delta}^*\boldsymbol{\sigma})\}_{\boldsymbol{\sigma}\in\Delta_{Q_2}}$ as well as the logical operator $X(\boldsymbol{z}^*_{Q_1})$ with $\boldsymbol{\delta}^*\boldsymbol{z}^*_{Q_1} = 0$.

We write the simultaneous Hadamard transform as $\mathsf{H} = \bigotimes_{\boldsymbol{\sigma}\in\Delta_{Q_1}} H_{\boldsymbol{\sigma}}$. The states $\mathsf{H}|\Phi^{(\text{CSS})}\rangle$ and $|\Phi^{(\text{CSS})*}\rangle$ are almost the same as they are both stabilized by the same local operators. However, they are different as logical states since the former is stabilized by $Z(\boldsymbol{z}_{Q_1})$ while the latter is stabilized by $X(\boldsymbol{z}^*_{Q_1})$. The difference can be accounted for by summing over logical operators:

$$\mathsf{H}|\Phi^{(\text{CSS})}\rangle = \frac{1}{|\mathcal{L}|}\sum_{[\boldsymbol{z}_{Q_1}]\in\mathcal{L}} Z(\boldsymbol{z}_{Q_1})|\Phi^{(\text{CSS})*}\rangle, \tag{168}$$

where $\mathcal{L} = \text{Ker}(\boldsymbol{\delta}:C_{Q_1}\to C_{Q_2})/\text{Im}(\boldsymbol{\delta}:C_{Z,w}\to C_{Q_1})$ is the homology group and $[\bullet]$ is a homology class represented by the element $\bullet$.

We make use of identities,

$$\langle 0|e^{JX}H = (\sinh 2J)^{1/2}\langle 0|e^{J^*X}, \qquad \langle 0|e^{KX}H = (\sinh 2K)^{1/2}\langle 0|e^{K^*X}. \tag{169}$$

The identity

$$\langle\omega(J,K)|\Phi^{(\text{CSS})}\rangle = \langle\omega(J,K)|\mathsf{H}\cdot\mathsf{H}|\Phi^{(\text{CSS})}\rangle, \tag{170}$$

yields a refined version of the duality (163):

$$\mathcal{Z}^{(\text{CSS})}(J,K) = \frac{2^{|\Delta_{Q_2}|}(\sinh 2J)^{|\Delta_Z|/2}(\sinh 2K)^{|\Delta_{\text{q},w}|/2}}{2^{|\Delta_{Z,w}|}|\mathcal{L}|}\sum_{[\boldsymbol{z}_{Q_1}]\in\mathcal{L}} \mathcal{Z}^{(\text{CSS}),\text{twisted}}_{\text{dual}}(J^*,K^*;\boldsymbol{z}_{Q_1}), \tag{171}$$

where we introduced a twisted partition function

$$\mathcal{Z}^{(\text{CSS}),\text{twisted}}_{\text{dual}}(J^*,K^*;\boldsymbol{z}_{Q_1})$$
$$= \sum_{\{s(\boldsymbol{\sigma})=\pm 1\}_{\boldsymbol{\sigma}\in\Delta_{Z,w}}} \exp\Big(J^*\sum_{\boldsymbol{\sigma}'\in\Delta_Z}(-1)^{\#(\boldsymbol{z}_{Q_1}\cap\boldsymbol{\sigma}')}s(\boldsymbol{\delta}^*\boldsymbol{\sigma}') + K^*\sum_{\boldsymbol{\sigma}''\in\Delta_{\text{q},w}}(-1)^{\#(\boldsymbol{z}_{Q_1}\cap\boldsymbol{\sigma}'')}s(\boldsymbol{\delta}^*\boldsymbol{\sigma}'')\Big). \tag{172}$$

We can interpret the strange correlator overlap $\langle \omega(J,K)|\Phi^{(\text{CSS})}\rangle$ as specifying the boundary conditions in the SymTFT in the following sense:

1. Topological boundary condition specified by $|\Phi^{(\text{CSS})}\rangle$,
2. Dynamical boundary condition specified by $|\omega(J,K)\rangle$.

This gives the partition function as an overlap $\mathcal{Z}^{(\text{CSS})}(J,K) \propto \langle \omega(J,K)|\Phi^{(\text{CSS})}\rangle$. Similarly, a different topological boundary condition, specified by $\mathsf{H}|\Phi^{(\text{CSS})*}\rangle$, yields a different partition function $\mathcal{Z}^{(\text{CSS})}_{\text{dual}}(J^*,K^*) \propto \langle \omega(J,K)|\mathsf{H}|\Phi^{(\text{CSS})*}\rangle$. The two partition functions are related as in (171), which implements Kramers-Wannier duality at the level of the classical statistical model. The Kramers-Wannier duality can be understood as gauging the classical spin-flip symmetry of the respective models.

The classical spin-flip symmetries of the statistical model $\mathcal{T}^{(\text{CSS})}(J,K)$, defined by its partition function $\mathcal{Z}^{(\text{CSS})}(J,K)$, are the group

$$\mathcal{G}^{(\text{CSS})} = \{F_{Q_2}(z^*)\,|\,\delta^* z^* = 0, z^* \in C_{Q_2}\}, \tag{173}$$

where the element $F_{Q_2}(z^*)$ acts on spins via the action $F_{Q_2}(z^*)s(\sigma) = (-1)^{\#(z^* \cap \sigma)}s(\sigma)$ and the spins take values $s(\sigma) = \pm 1$ with $\sigma \in \Delta_{Q_2}$. Similarly, one can consider the classical spin-flip symmetries for the statistical model $\mathcal{T}^{(\text{CSS})}_{\text{dual}}(J^*,K^*)$ defined by the partition function $\mathcal{Z}^{(\text{CSS})}_{\text{dual}}(J^*,K^*)$:

$$\mathcal{G}^{(\text{CSS})}_{\text{dual}} = \{F_{Z,w}(z)\,|\,\delta z = 0, z \in C_{Z,w}\}, \tag{174}$$

where the element $F_{Z,w}(z)$ act on spins via the action $F_{Z,w}(z)s(\sigma) = (-1)^{\#(z \cap \sigma)}s(\sigma)$ and the spins take values $s(\sigma) = \pm 1$ with $\sigma \in \Delta_{Z,w}$. The equality (171) implies the duality relation

$$\mathcal{T}^{(\text{CSS})}(J,K) \simeq \mathcal{T}^{(\text{CSS})}_{\text{dual}}(J^*,K^*)/\mathcal{G}^{(\text{CSS})}_{\text{dual}}. \tag{175}$$

Modifying the derivation of (171) by inverting the relation (168) between $\mathsf{H}|\Phi^{(\text{CSS})}\rangle$ and $|\Phi^{(\text{CSS})*}\rangle$, we also obtain the duality relation

$$\mathcal{T}^{(\text{CSS})}_{\text{dual}}(J^*,K^*) \simeq \mathcal{T}^{(\text{CSS})}(J,K)/\mathcal{G}^{(\text{CSS})}. \tag{176}$$

We find again that the Kramers-Wannier transformation is implemented by exchanging the topological boundary conditions specified by $\mathsf{H}|\Phi^{(\text{CSS})}\rangle$ and $|\Phi^{(\text{CSS})*}\rangle$ in the SymTFT.

# 5 Self-dual models, non-invertible symmetry, and measuring cluster states

In this section, we consider a family of self-dual quantum models constructed from classical codes described by

$$\mathbf{CC}^{\text{self-dual}}_{(d,k)}: \quad 0 \longrightarrow C_{d-k} \xrightarrow{\delta_Z} C_k \longrightarrow 0, \tag{177}$$

where $2k < d$. We denote the coordinate directions by $x_1,\dots,$ and $x_d$. The chain group $C_{d-k}$ ($C_k$) is generated by $(d-k)$-dimensional ($k$-dimensional) cells of a $d$-dimensional hypercubic lattice. The differential $\delta_Z$ instructs us to append all the $k$-cells that appear within a $(d-k)$-cell. The Hamiltonian for this classical code or classical spin model can be written as

$$H^{\mathbf{CS}}_{(d,k)} = -\sum_{\sigma_\beta \in \Delta_{d-k}} Z(\delta_Z \sigma_\beta). \tag{178}$$

We note that this classical code can be promoted to a quantum CSS code described by

$$\mathbf{QC}_{(d,k,l)}^{\text{self-dual}}: \quad 0 \longrightarrow C_{d-k} \xrightarrow{\delta_Z} C_k \xrightarrow{\delta_X} C_l \longrightarrow 0 \tag{179}$$

when

$$\binom{d-k-l}{k-l} = 0 \bmod 2 \tag{180}$$

for some $0 \le l < k$. The constraint (180) guarantees the nilpotency $\delta_X \circ \delta_Z = 0$. Explicitly, the CSS code Hamiltonian that can be constructed from the chain complex (179) is

$$H_{(d,k,l)}^{\mathbf{QC}} = -\sum_{\sigma_\alpha \in \Delta_l} X(\delta_X^* \sigma_\alpha) - \sum_{\sigma_\beta \in \Delta_{d-k}} Z(\delta_Z \sigma_\beta). \tag{181}$$

The excitations of this model are of two types: 1) $X$-type stabilizer violations and 2) $Z$-type stabilizer violations. In some cases, for example $d = 5$, $k = 2$ case, the electric excitations (i.e., violations of $X$ type stabilizers) behave like fractons. This can be understood as follows: suppose a violation of the X-type stabilizer occurs at a 1-cell pointing in the $x_1$ coordinate direction, then this excitation cannot move in the $x_1$ coordinate direction without incurring additional energy cost.

We define a quantum Hamiltonian from (177) that is self-dual under gauging its symmetries.

$$H_{(d,k)}^{\mathbf{CC}} = -\sum_{\sigma_i \in \Delta_k} X(\sigma_i) - \lambda \sum_{\sigma_\beta \in \Delta_{d-k}} Z(\delta_Z \sigma_\beta). \tag{182}$$

This Hamiltonian is invariant under subsystem symmetry transformations generated by the $X$ operators on $(k+1)$-dimensional hyperplanes, each of which consists of $k$-cells orthogonal to it (see Appendix F.3 for explicit examples). When there exists an $l$ ($0 \le l < k$) such that the constraint (180) is satisfied, $\mathbf{CC}_{(d,k)}^{\text{self-dual}}$ can be extended to $\mathbf{QC}_{(d,k)}^{\text{self-dual}}$ and the Hamiltonian (182) has symmetries generated by the $X$ operators supported on $\delta_X^* \sigma_\alpha$ where $\sigma_\alpha \in \Delta_l$ is an elementary $l$-cell. We will refer to such symmetries as local symmetries; in the present case, they form a global symmetry generator supported on a homologically trivial dual cycle. We note that the subsystem symmetries and local symmetries are not completely independent. It may happen that an appropriate sum of local generators gives a sum of subsystem generators. This can be easily seen in the example $\mathbf{CC}_{(3,1)}^{\text{self-dual}}$ and more such examples are discussed in Appendix F.3.

Under Krames-Wannier transformation (138), Hamiltonian (182) is dual to

$$H_{(d,k),\text{dual}}^{\mathbf{CC}} = -\lambda \sum_{\sigma_\beta \in \Delta_{d-k}} X(\sigma_\beta) - \sum_{\sigma_i \in \Delta_k} Z(\delta_Z^* \sigma_i). \tag{183}$$

In $d$ dimensions, a $k$-cell is dual to a $(d-k)$-cell. With this identification, the Hamiltonians Eq. (182) and Eq. (183) are the same and the Kramers-Wannier duality is a self-duality. This self-duality was studied for $k = 0$ case in $d$ dimensions by [57].

## 5.1 Non-invertible symmetry

With a tuned coupling constant $\lambda = 1$, the Kramers-Wannier duality becomes a symmetry of the models (up to shift of lattices[26]). To be concrete, the Kramers-Wannier duality operator

$$\mathrm{KW} = \langle + |^{\Delta_k} \prod_{\substack{\sigma_i \in \Delta_k \\ \sigma_\beta \in \Delta_{d-k}}} CZ_{\sigma_i, \sigma_\beta}^{a(\delta_Z \sigma_\beta; \sigma_i)} | + \rangle^{\Delta_{d-k}}, \tag{184}$$

---

[26]Recently, Kramers-Wannier duality operators acting on the same Hilbert space (rather than introducing ancillas on dual lattices) have been studied [56,57,82,83]. Our study here is complementary to these approaches.

obeys the algebra

$$\text{KW} \cdot H^{\text{CC}}_{(d,k)} = \left( H^{\text{CC}}_{(d,k)} \right)^{\mathsf{T}} \cdot \text{KW}, \qquad X(z_{d-k}) \cdot \text{KW} = \text{KW}, \qquad \text{KW} \cdot X(z_k^*) = \text{KW}, \qquad (185)$$

where $z_{d-k} \in C_{d-k}$ and $z_k^* \in C_k$ are the chains that describe symmetries of the model $\text{CC}^{\text{self-dual}}_{(d,k)}$; they are (dual) cycles which satisfy $\delta_Z z_{d-k} = 0$ and $\delta_Z^* z_k^* = 0$. The superscript $\mathsf{T}$ denotes the shift of the lattice by $\frac{1}{2}$ in every direction so that the dual lattice matches the original one. The transformation generated by KW is a symmetry of the Hamiltonian and it is non-invertible:

$$\text{KW} \circ \text{KW}^\dagger = \frac{1}{2^{|\Delta_{d-k}|}} \sum_{\delta_Z z_{d-k}=0} X(z_{d-k}), \qquad \text{KW}^\dagger \circ \text{KW} = \frac{1}{2^{|\Delta_k|}} \sum_{\delta_Z^* z_k^*=0} X(z_k^*). \qquad (186)$$

In other words, the set of symmetry operators forms a non-invertible fusion rule.

### 5.1.1 Example: 3d quantum cube Ising model

The case with $\text{CC}^{\text{self-dual}}_{(2,0)}$ corresponds to the 2d quantum plaquette Ising model. For this model, the non-invertible duality transformation involving subsystem symmetries and the duality operator was found in Ref. [48], and our formalism reproduces their result.

Let us consider $\text{CC}^{\text{self-dual}}_{(3,0)}$ as the next non-trivial example. We may call the model the quatum cube Ising model. Let us denote a cube by $c \in \Delta_3$ and a vertex by $v \in \Delta_0$. In the chain complex $C_3 \xrightarrow{\delta_Z} C_0$ generated by those cells, we use the boundary operator $\delta_Z$ which takes the eight corners of a cube. The Hamiltonian

$$H_{\text{CIM}} = - \sum_{v \in \Delta_0} X_v - \lambda \sum_{c \in \Delta_3} \prod_{v \subset c} Z_v \,, \qquad (187)$$

is symmetric under transformations supported on cycles $z_0^* \in C_0$, which are line symmetries in $x$, $y$, or $z$ directions such as

$$S_x(y,z) := \prod_{k \in \mathbb{Z}/\mathbb{Z}_{L_x}} X(v = \{(k, y, z)\}), \qquad (188)$$

with $(y, z)$ an arbitrary coordinate to specify a line in the $x$ direction. Similarly, we define generators $\{S_y(x, z)\}$ and $\{S_z(x, y)\}$ for $y$ and $z$ directions, respectively. However, not all such generators are independent. There are constraints such as $\prod_{y \in \mathbb{Z}/\mathbb{Z}_{L_y}} S_x(y, z) = \prod_{x \in \mathbb{Z}/\mathbb{Z}_{L_x}} S_y(x, z)$. The number of independent symmetry generators is $L_x L_y + L_y L_x + L_z L_x - L_x - L_y - L_z + 1$; this is also the base-2 logarithm of the ground state degeneracy at $\lambda \to \infty$. We denote the set of independent line symmetry generators that act on vertices by $\mathbb{L}^*$, and those that act on dual vertices by $\mathbb{L}$.

The model is self-dual. For each vertex, one can associate the center of a dual cube, and vice versa. The general formula (185) gives us the duality $\text{KW} \cdot H_{\text{CIM}} = (H_{\text{CIM}})^{\mathsf{T}} \cdot \text{KW}$ and the gauging relations $S \cdot \text{KW} = \text{KW} \cdot S^* = \text{KW}$ with $S \in \mathbb{L}$ and $S^* \in \mathbb{L}^*$. Furthermore, the non-invertible algebra (186) becomes

$$\text{KW} \circ \text{KW}^\dagger = \frac{1}{2^{|\Delta_3|}} \sum_{S \in \mathbb{L}} S, \qquad \text{KW}^\dagger \circ \text{KW} = \frac{1}{2^{|\Delta_0|}} \sum_{S^* \in \mathbb{L}^*} S^*. \qquad (189)$$

The set of symmetry operators, KW, $\text{KW}^\dagger$, and line symmetry operators, thus form a non-invertible fusion rule. As a remark, an explicit operator representation of the Kramers-Wannier duality symmetry for the plaquette Ising model, cubic Ising model and other hypercubic Ising model was constructed in [57] where the symmetry operator is realized as a map on the same Hilbert space where the qubits live.

## 5.2 Self-duality of classical partition functions from strange correlators for foliated cluster states

Here, we apply the construction in Section 4.3 to the self-dual models in the previous section. From the chain complex for $\mathbf{CC}^{\text{self-dual}}_{(d,k)}$, we obtain a foliated chain complex (49) with

$$\boldsymbol{C}_{Z,w} = C_{d-k} \otimes C_w, \qquad \boldsymbol{C}_{Q_1} = C_{d-k} \otimes C_0 \oplus C_k \otimes C_w, \qquad \boldsymbol{C}_{Q_2} = C_k \otimes C_0, \quad \text{and} \quad \boldsymbol{C}_X = 0, \quad (190)$$

where the second factor represents the cells in the $w$ direction. Then, the partition function $\mathcal{Z}^{(\text{CSS})}(J,K)$ in (163) is described by the classical Hamiltonian whose spins are on $\Delta_k \otimes \Delta_0$ (the $k$-cells at points in the $w$ direction). Interactions consist of two types:

(i) One is the boundary of a cell in $\Delta_{d-k} \otimes \Delta_0$ within $w = \{pt\}$ slices with strength $J$.

(ii) The other is the boundary of a cell in $\Delta_k \otimes \Delta_w$ and thus simply an Ising two-body interaction with strength $K$.

In the duality (163), the right hand side is a model with the partition function $\mathcal{Z}^{(\text{CSS})}_{\text{dual}}(J^*, K^*)$, whose spins are placed on $\Delta_{d-k} \otimes C_w$. Two types of interactions are:

(i)* An Ising-type two-body interaction in the $w$ direction with strength $J^*$.

(ii)* An interaction with strength $K^*$ among spins in the dual boundary $C_k \otimes C_w \xrightarrow{\delta_Z^*} C_{d-k} \otimes C_w$.

The form of interaction (i)* is identical to (ii), and (ii)* to (i), respectively. Upon identification between $(J,K)$ and $(K^*, J^*)$, we see that the partition functions are self-dual to each other. On finite lattices, one can improve the duality by including the sum over twists as in (171). This will be explained with an example below.

### 5.2.1 Example: 3d classical anisotropic plaquette Ising model

Let us consider the 2d plaquette Ising model, which is a self-dual code $\mathbf{CC}^{\text{self-dual}}_{(2,0)}$. The foliated chain complex was given in (76), where we set $\boldsymbol{\Delta}_Z = \boldsymbol{\Delta}_{xy}$ (horizontal faces), $\boldsymbol{\Delta}_{q,w} = \boldsymbol{\Delta}_w$ (vertical edges), $\boldsymbol{\Delta}_q = \boldsymbol{\Delta}_0$ (vertices). We get $\boldsymbol{C}_{Z,w} = \boldsymbol{C}_{xyw}$, $\boldsymbol{C}_{Q_1} = \boldsymbol{C}_{xy} \oplus \boldsymbol{C}_w$, $\boldsymbol{C}_{Q_2} = \boldsymbol{C}_0$, and $\boldsymbol{C}_{X,w} = \boldsymbol{C}_X = 0$. For the left hand side of (163), we get a statistical partition function (161)

$$\mathcal{Z}^{(\text{qPIM})}(J,K) = \sum_{\{s(v)=\pm 1\}_{v\in\Delta_0}} \exp\Big[J \sum_{f\in\Delta_{xy}} s(\delta f) + K \sum_{e\in\Delta_w} s(\delta e)\Big]. \quad (191)$$

It is the 3d classical anisotoropic plaquette Ising model (cAPIM) [46]. The first term is the plaquette term with the product of four spins at corners of the plaquette. The second term is the ordinary Ising term in the $w$ direction.

For the right hand side of (163), we get a dual statistical partition function (164):

$$\mathcal{Z}^{(\text{qPIM})}_{\text{dual}}(J,K) = \sum_{\{s(c)=\pm 1\}_{c\in\Delta_{xyw}}} \exp\Big[J \sum_{f\in\Delta_{xy}} s(\delta^* f) + K \sum_{e\in\Delta_w} s(\delta^* e)\Big]. \quad (192)$$

It is illustrative to interpret interactions in the dual lattice using identifications $\boldsymbol{\Delta}_{xyw} \simeq \boldsymbol{\Delta}_0^*$, $\boldsymbol{\Delta}_{xy} \simeq \boldsymbol{\Delta}_w^*$, and $\boldsymbol{\Delta}_w \simeq \boldsymbol{\Delta}_{xy}^*$. In the dual lattice picture, the first term is the ordinary Ising interaction in the $w$ direction. The second term is a product of four spins on dual vertices around a dual plaquette. Thus, the dual model is also the 3d cAPIM. The duality (163) reproduces the self-duality of 3d cAPIM, which has been known in the literature, see e.g. [46, 84]. (We give an illustration for the derivation of the duality in Figure 10.)

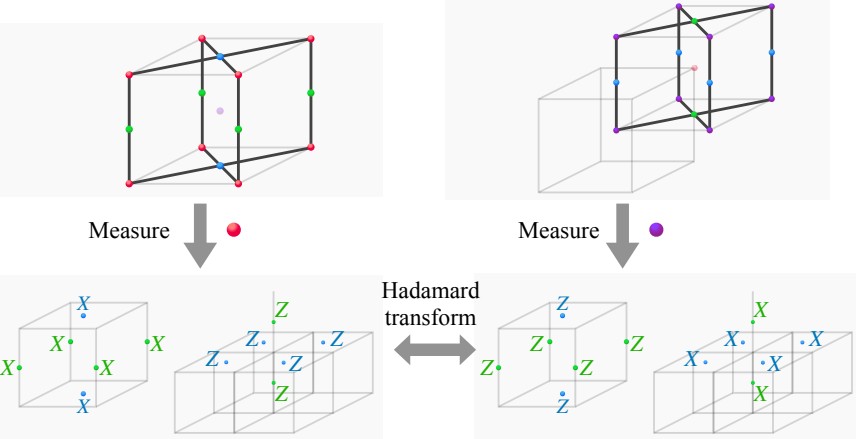

Figure 10: The Kramers-Wannier duality in the plaquette Ising model from the Hadamard transformation in the resource states.

On finite lattices, we can improve the duality as in (171) by taking into account twist defects. They are inserted along representatives in homology classes, $[z_{Q_1}] \in \mathcal{L}$, where we write $\mathcal{L} = \text{Ker}(\delta : C_{Q_1} \to C_{Q_2})/\text{Im}(\delta : C_{Z,w} \to C_{Q_1})$. It is generated by the basis

$$\left( \bigcup_{\substack{v=\{(x,y)\} \\ \text{s.t. } xy=0}} \sum_{k\in\mathbb{Z}_{L_w}} v \otimes [k, k+1] \right) \cup \left( \bigcup_{y\in\mathbb{Z}_{L_y}} \sum_{k\in\mathbb{Z}_{L_x}} [k, k+1] \otimes [y, y+1] \otimes \{w=0\} \right)$$

$$\cup \left( \bigcup_{x\in\mathbb{Z}_{L_x}} \sum_{k\in\mathbb{Z}_{L_y}} [x, x+1] \otimes [k, k+1] \otimes \{w=0\} \right). \quad (193)$$

The first part is a union of vertical lines in the $w$ direction at $(x, y)$ that satisfies $x = 0$ or $y = 0$. Note that all the other vertical lines in the bulk ($x \geq 1$ and $y \geq 1$) can be generated from them by $\delta c$ (with $c \in \Delta_{xyw}$) summed over the $w$ direction. The second and the third part is a union of the sum of plaquettes at $w = 0$ along lines in $x$ or $y$ directions, respectively. Lines in the bulk ($w \geq 1$) can be generated from lines at $w = 0$ using $\delta c$ summed over either $x$ or $y$ directions.

## 5.3 Ground state degeneracy

We exhibited a general fusion algebra involving invertible and non-invertible global symmetries of $H_{(d,k)}^{\textbf{CC}}$. To be more concrete, we study the number of independent (invertible) global symmetries in the model, which amounts to counting the ground state degeneracy of the classical-code Hamiltonian $H_{(d,k)}^{\textbf{CS}}$. For the low dimensional models that we study in the Appendix.F.3, the ground state degeneracy can be computed explicitly using Gröbner-basis method [85] in the algebraic formalism. We summarize the ground state degeneracy in Table 2 for the classical model (177) and in Table 3 for the CSS codes (179).

Note that in both Tables 2 and 3, we have taken the number of vertices in all $d$ directions $x_1, \ldots, x_d$ to be $L_1 = L_2 = \ldots = L_d = L$. This does not lose generality as one can deduce the polynomial with different $L_1, \ldots, L_d$ by promoting $L^k$ for $1 \leq k \leq d$ to $\frac{1}{\binom{d}{k}} \sum_{\{i_1,\ldots,i_k\}\subset\{1,\ldots,d\}} L_{i_1} \ldots L_{i_k}$ from the polynomials that we have written down in the two tables. In the case of classical code $\textbf{CC}_{(d,k)}^{\text{self-dual}}$, all the models except $(4, 1)$ and $(6, 1)$ have leading

terms $L^d$. This is due to the existence of a local symmetry, which we explained below (182), in those models — i.e., there exits $l$ that satisfies the condition (180). However, for the cases $(4,1)$ and $(6,1)$, the ground state degeneracy scales as $L^{d-1}$ indicating an absence of local symmetry.

Let us define $\mathcal{S}$ to be the set of all values of $0 \le l < k$ for which (180) holds.

From the general argument given above, we come to the following asymptotics of ground-state degeneracy of self-dual codes.

1. $\mathbf{CC}_{(d,k)}^{\text{self-dual}}$ satisfying (180) for some $l \in \{0, 1, \ldots, k-1\}$. Note that for $l \in \mathcal{S}$, $\mathbf{CC}_{(d,k)}^{\text{self-dual}}$ can be promoted to $\mathbf{QC}_{(d,k,l)}^{\text{self-dual}}$. Let $\mathcal{N}_l$ be the set of local symmetry generators of the form $\delta_X^* \sigma_\alpha$ where $\sigma_\alpha \in \Delta_l$. We define $\mathcal{I}$ to be the number of independent local symmetry generators among all possible local symmetry generators $\bigcup_{l \in \mathcal{S}} \mathcal{N}_l$. Then

$$\log_2 \text{GSD} = \mathcal{I} + \mathcal{O}(L^{d-1}). \tag{194}$$

Note that $\mathcal{I} \sim \mathcal{O}(L^d)$ and also captures the coefficient in front as in the $(d,k) = (5,2)$ model.

2. $\mathbf{CC}_{(d,k)}^{\text{self-dual}}$ not satisfying (180) for all $0 \le l < k$.

$$\log_2 \text{GSD} = \mathcal{O}(L^{d-1}). \tag{195}$$

3. $\mathbf{QC}_{(d,k,l)}^{\text{self-dual}}$. We define $\mathcal{I}'$ to be the number of independent local symmetry generators among all possible local symmetry generators $\bigcup_{l' \in \mathcal{S} \setminus \{l\}} \mathcal{N}_{l'}$.

$$\log_2 \text{GSD} = \mathcal{I}' + \mathcal{O}(L^{d-1}). \tag{196}$$

Note that in this case there is no local symmetry generators of the form $\delta_X^* \sigma_\alpha$ where $\sigma_\alpha \in \Delta_l$; they are $X$ type stabilizers in the quantum CSS code.

Note that these asymptotics agree with the examples given in the Tables 2 and 3. Determining a more concrete form of the ground state degeneracy in general is a non-trivial and interesting problem, which we leave as a future work.

Table 2: Summary of ground state degeneracy in $\mathbf{CC}_{(d,k)}^{\text{self-dual}}$.

| $(d,k)$ | $\log_2(\text{GSD})$ polynomial | leading local/subsystem symmetry |
|---|---|---|
| $(3,1)$ | $L^3 + 2$ | six-edges star (the fourth column of (F.35)) |
| $(4,1)$ | $8L^3 - 12L^2 + 16L - 8$ | four-edges stars within a plane stacked in an orthogonal direction (F.40) |
| $(5,1)$ | $L^5 + 10L^4 - 20L^3$ $+ 40L^2 - 40L + 14$ | ten-edges star (the first one in (F.48)) |
| $(5,2)$ | $4L^5 + 6$ | eight-plaquettes star (F.55) |
| $(6,1)$ | $24L^5 - 60L^4 + 120L^3$ $- 150L^2 + 96L - 24$ | four-edges stars within a plane stacked in an orthogonal direction (F.61) |
| $(6,2)$ | $L^6 + 24L^5 - 60L^4$ $+ 160L^3 - 240L^2 + 204L - 74$ | sixty-plaquettes star (the first one in (F.67)) |

Table 3: Summary of ground state degeneracy in $\mathbf{QC}^{\text{self-dual}}_{(d,k,l)}$.

| $(d,k,l)$ | $\log_2(\text{GSD})$ polynomial |
|---|---|
| $(3,1,0)$ | 3 |
| $(5,1,0)$ | $10L^4 - 20L^3 + 40L^2 - 40L + 15$ |
| $(5,2,1)$ | 10 |
| $(6,2,0)$ | $24L^5 - 60L^4 + 160L^3 - 240L^2 + 210L - 79$ |

# 6  Discussion and conclusion

We systematically studied dualities, non-invertible operators, and the anomaly inflow mechanism for a large class of spin models described by CSS codes. Our study involved the CSS chain complex as the starting point, and we reformulated the foliation construction [23] of a cluster state using the tensor product of chain complexes. Moreover, we used a further tensor product with another chain complex that models the physical time, to describe the spacetime motion of defects, which was used to show the anomaly inflow. Along the way, we provided a BF theory description for any model that can be viewed as a CSS code. It would be interesting to compare with other works that studied similar BF theory descriptions for models with subsystem symmetries [66,72,73,86–88]. Recently, quantum low-density parity check codes [25] have attracted significant attention. In Ref. [89], a generic construction of such quantum error correction codes was elucidated using the so-called product construction from classical codes (see also Ref. [90]). Our construction of SPT states using the tensor product of a CSS code and the chain for the foliation direction — i.e., the repetition code — would be relevant to studies along this line. We further note here that our bulk SPT states map to topologically ordered states upon partial measurements.

Originally, the foliation construction of a cluster state from a CSS code was introduced to discuss quantum error correction. It would be interesting to consider quantum error correction for fractonic codes from this view point. The foliation construction has been extended to non-CSS stabilizer codes [91]. An interesting direction along these lines would be to dynamical codes [92] such as the Floquet color code [93,94]. The foliation construction based on Ref. [91] for the Floquet color code was given in Ref. [95], where some similarity to the RBH state was pointed out. It would be interesting to extend the analysis in this paper to such codes.

We provided a construction of statistical models associated with general CSS codes via strange correlators. We provided different kinds of duality between partition functions, where examples included Wegner's dualities (see also Ref. [45]), the self-duality of the anisotropic plaquette Ising model, as well as some new examples. We utilized the stabilizer formalism and showcased the effectiveness of the cluster state measurement for deriving precise dualities. As discussed in our previous work [45], the duality between partition functions can be understood as imposing different topological boundary conditions on the so-called SymTFT [76–79]. Measuring resource states would serve as a useful language in discussing dualities that involve more intricate topological field theories or fusion categories; see e.g. [54,96] for related ideas.

We studied non-invertible symmetries in various spin models. Recently, non-invertible symmetry proved to be useful to constrain the nature of ground states in spin models [83], much in the spirit of the Lieb-Schulz-Mattis theorem [97]. Although our construction differs from that in Ref. [83] in that we introduce new qubits to implement dualities, it may be interesting to investigate highly non-trivial ground state degeneracy of subsystem symmetric models that we introduced in this paper.

To conclude, our framework based on cluster-state measurements and CSS chain complexes and their generalizations unites various prominent models, and has provided a number of rigorous and generalized results. We believe that our framework can be developed further, and will further strengthen the links between various branches in theoretical physics.

## Acknowledgments

T.O. thanks T. Nishioka for useful discussions on CSS codes. A.P.M. and H.S. thank Y. Li, M. Litvinov, S.-H. Shao, and T.-C. Wei for useful discussion. A.P.M. would like to thank M. Swaminathan for discussions on intersection pairing between homology classes.

**Funding information** A.P.M. and H.S. were partially supported by the Materials Science and Engineering Divisions, Office of Basic Energy Sciences of the U.S. Department of Energy under Contract No. DESC0012704. The research of T. O. was supported in part by Grant-in-Aid for Transformative Research Areas (A) "Extreme Universe" No. 21H05190 and by JST PRESTO Grant Number JPMJPR23F3. We are grateful to the long term workshop YITP-T-23-01 held at YITP, Kyoto University, where a part of this work was done.

**Author contributions** The three authors contributed equally.

## A Examples of CSS codes and dualities

In this appendix, we will look at explicit examples of Kramers-Wannier transformations between Hamiltonians constructed out of CSS code chain complexes for fracton models. In particular, we will illustrate KW, its fusion rule with KW$^\dagger$, and strange correlators for the X-cube model, the checkerboard model and Haah's code. We note that the strange correlator construction gives the partition functions of classical statistical models provided in Table I of Ref. [15].

### A.1 X-cube model

Let us recall that the chain complex for X-cube model is

$$0 \xrightarrow{\delta} C_{xyz} \xrightarrow{\delta_Z} \bigoplus_{k=x,y,z} C_k \xrightarrow{\delta_X} \bigoplus_{k=x,y,z} C_0^{(k)} \xrightarrow{\delta} 0, \tag{A.1}$$

defined on a 3-dimensional cubic lattice with the differentials defined as in (87) and (88) satisfying $\delta_X \circ \delta_Z = 0$. The dual chain complex is

$$0 \longrightarrow \bigoplus_{k=x,y,z} C_0^{(k)} \xrightarrow{\delta_X^*} \bigoplus_{k=x,y,z} C_k \xrightarrow{\delta_Z^*} C_{xyz} \longrightarrow 0. \tag{A.2}$$

The Kramers-Wannier operator converts the Hamiltonians as

$$\widetilde{H}_{\text{XC}} = \sum_{\sigma_0^{(k)} \in \Delta_0^{(k)}} X(\sigma_0^{(k)}) - \lambda \sum_{\sigma_{xyz} \in \Delta_{xyz}} \prod_{\sigma_k \subset \sigma_{xyz}} Z(\sigma_k)$$

$$\xrightleftharpoons[\text{KW}^\dagger]{\text{KW}} \widetilde{H}_{\text{XC,dual}} = - \sum_{\sigma_k \in \Delta_k} \prod_{\substack{\sigma_{xyz} \supset \sigma_k \\ \sigma_{xyz} \in \Delta_{xyz}}} Z(\sigma_{xyz}) - \lambda \sum_{\sigma_{xyz} \in \Delta_{xyz}} X(\sigma_{xyz}). \tag{A.3}$$

on the symmetric Hilbert spaces as in (142). The second term in the Hamiltonian $H_{\text{XC}}$ is the cube term in the X-cube model and is given by the product of twelve edges in a cube. It is invariant under subsystem one-form symmetry transformations generated by

$$X(\delta_X^* \sigma_0^{(x)}), \qquad X(\delta_X^* \sigma_0^{(y)}), \qquad X(\delta_X^* \sigma_0^{(z)}) \quad \text{etc.,} \tag{A.4}$$

which is an example of the dual cycle $\delta_Z^* z_{\text{q}} = 0$ ($z_{\text{q}} \in \bigoplus_{k=x,y,z} C_k$). The Hamiltonian $\widetilde{H}_{\text{XC,dual}}$ is defined on the vertices of the dual lattice. This is the Hamiltonian of the 3d plaquette Ising model. It is invariant under rigid planar symmetry transformations generated by

$$\prod_{\sigma_{xyz} \in \Delta_{xyz} \cap \left( x=[k,k+1] \right)} X(\sigma_{xyz}) \quad \text{etc.,} \tag{A.5}$$

which is an example of the cycle $\delta_Z z_Z = 0$ ($z_Z \in C_{xyz}$). The composition $\text{KW}^\dagger \circ \text{KW}$ gives us the sum of the symmetry generators in (A.4). The reversed composition $\text{KW} \circ \text{KW}^\dagger$ yields the sum over the symmetry generators in (A.5). The strange correlator gives the 3d plaquette Ising model. We refer to Table I in Ref [15] for figures.

## A.2 Checkerboard model

We begin by recapitulating the chain complex for the checkerboard model in (96):

$$0 \longrightarrow C_{xyz,Z}^{(s)} \xrightarrow{\delta_Z} C_0 \xrightarrow{\delta_X} C_{xyz,X}^{(s)} \longrightarrow 0, \tag{A.6}$$

defined on a 3-dimensional cubic lattice. The dual chain complex reads

$$0 \longrightarrow C_{xyz,X}^{(s)} \xrightarrow{\delta_X^*} C_0 \xrightarrow{\delta_Z^*} C_{xyz,Z}^{(s)} \longrightarrow 0. \tag{A.7}$$

The set of dual boundary operators is also nilpotent: $\delta_Z^* \circ \delta_X^* = 0$.

The Kramers-Wannier operator converts Hamiltonians as

$$\widetilde{H}_{\text{CBM}} = - \sum_{\sigma_0 \in \Delta_0} X(\sigma_0) - \lambda \sum_{\sigma_{xyz} \in \Delta_{xyz}^{(s)}} \prod_{\sigma_0 \subset \sigma_{xyz}} Z(\sigma_0)$$

$$\underset{\text{KW}^\dagger}{\overset{\text{KW}}{\rightleftharpoons}} \quad \widetilde{H}_{\text{CBM,dual}} = - \sum_{\sigma_0 \in \Delta_0} \prod_{\substack{\sigma_{xyz} \supset \sigma_0 \\ \sigma_{xyz} \in \Delta_{xyz}^{(s)}}} Z(\sigma_{xyz}) - \lambda \sum_{\sigma_{xyz} \in \Delta_{xyz}^{(s)}} X(\sigma_{xyz}), \tag{A.8}$$

on the symmetric Hilbert spaces as in (142). The second term in the Hamiltonian $\widetilde{H}_{\text{CBM}}$ is the product over eight vertices in a shaded cube. It is invariant under rigid line symmetry transformations generated by

$$\prod_{j \in \{0,\dots,L_z-1\}} X(\{x\} \times \{y\} \times \{j\}) \quad \text{etc.,} \tag{A.9}$$

which is an example of the dual cycle $\delta_Z^* z_{\text{q}} = 0$ ($z_{\text{q}} \in C_0$). On the other hand, the Hamiltonian $\widetilde{H}_{\text{CBM,dual}}$ is defined on the shaded cubes. The first term in the Hamiltonian is the tetrahedral Ising term, which is a product over four shaded cubes around a vertex, see Figure 11. The Hamiltonian $\widetilde{H}_{\text{CBM,dual}}$ is invariant under rigid plane symmetry transformations generated by

$$\prod_{\sigma_{xyz} \in \Delta_{xyz}^{(s)} \cap \left( x=[k,k+1] \right)} X(\sigma_{xyz}) \quad \text{etc.,} \tag{A.10}$$

which is an example of the cycle $\delta_Z z_Z = 0$ ($z_Z \in C_{xyz}^{(s)}$), see Figure 11.

The composition $\text{KW}^\dagger \circ \text{KW}$ gives us the sum of the symmetry generators in (A.9). The reversed composition $\text{KW} \circ \text{KW}^\dagger$ yields the sum over the symmetry generators in (A.10). The strange correlator gives the 3d tetrahedral Ising model, which is illustrated in Figure 11.

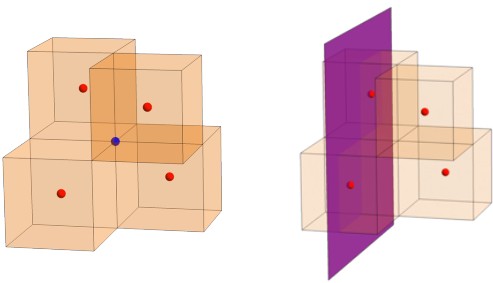

Figure 11: The tetrahedral Ising model and its global symmetry.

### A.3 Haah's code

To describe Haah's code [98], we consider a copy of 3-cells ($\Delta_{xyz,X}$ and $\Delta_{xyz,Z}$), two copies of cells at each vertex ($\Delta_0^R$ and $\Delta_0^B$). We consider a chain complex,

$$0 \longrightarrow C_{xyz,Z} \xrightarrow{\delta_Z} C_0^R \oplus C_0^B \xrightarrow{\delta_X} C_{xyz,X} \longrightarrow 0. \tag{A.11}$$

Here, for $[x, x+1] \times [y, y+1] \times [z, z+1] \in \Delta_{xyz,Z}$ we define

$$
\begin{aligned}
\delta_Z\big([x, x+1] &\times [y, y+1] \times [z, z+1]\big) \\
&= \{x+1\} \times \{y\} \times \{z\}^R + \{x\} \times \{y\} \times \{z+1\}^R + \{x+1\} \times \{y\} \times \{z+1\}^R \\
&\quad + \{x+1\} \times \{y+1\} \times \{z+1\}^R + \{x\} \times \{y\} \times \{z\}^B + \{x+1\} \times \{y+1\} \times \{z\}^B \\
&\quad + \{x+1\} \times \{y\} \times \{z+1\}^B + \{x\} \times \{y+1\} \times \{z+1\}^B,
\end{aligned} \tag{A.12}
$$

for $\{x\} \times \{y\} \times \{z\}^R \in \Delta_0^R$ we define

$$
\begin{aligned}
\delta_X\big(\{x\} &\times \{y\} \times \{z\}^R\big) \\
&= [x, x+1] \times [y, y+1] \times [z-1, z] + [x, x+1] \times [y-1, y] \times [z, z+1] \\
&\quad + [x-1, x] \times [y, y+1] \times [z, z+1] + [x-1, x] \times [y-1, y] \times [z-1, z],
\end{aligned} \tag{A.13}
$$

and for $\{x\} \times \{y\} \times \{z\}^B \in \Delta_0^B$ we define

$$
\begin{aligned}
\delta_X\big(\{x\} &\times \{y\} \times \{z\}^B\big) \\
&= [x, x+1] \times [y, y+1] \times [z, z+1] + [x, x+1] \times [y-1, y] \times [z, z+1] \\
&\quad + [x-1, x] \times [y-1, y] \times [z, z+1] + [x, x+1] \times [y-1, y] \times [z-1, z].
\end{aligned} \tag{A.14}
$$

See Figure 12 for illustration. The above set of boundary operators satisfies the nilpotency condition $\delta_X \circ \delta_Z = 0$. The dual boundary operators are defined as the transpose of $\delta$:

$$0 \longrightarrow C_{xyz,X} \xrightarrow{\delta_X^*} C_0^{(R)} \oplus C_0^{(B)} \xrightarrow{\delta_Z^*} C_{xyz,Z} \longrightarrow 0. \tag{A.15}$$

The set of dual boundary operators is also nilpotent: $\delta_Z^* \circ \delta_X^* = 0$. Haah's code is described by the stabilizers

$$Z(\delta_Z \sigma_{xyz}), \qquad X(\delta_X^* \sigma_{xyz}). \tag{A.16}$$

The Kramers-Wannier operator converts the Hamiltonians as

$$\widetilde{H}_{\text{HC}} = - \sum_{\sigma_0 \in \Delta_0^R \cup \Delta_0^B} X(\sigma_0) - \lambda \sum_{\sigma_{xyz} \in \Delta_{xyz,Z}} Z(\delta_Z \sigma_{xyz})$$

$$\xrightleftharpoons[\text{KW}^\dagger]{\text{KW}} \widetilde{H}_{\text{HC,dual}} = \sum_{\sigma_0 \subset \Delta_0^R \cup \Delta_0^B} Z(\delta_Z^* \sigma_0) - \sum_{\sigma_{xyz} \in \Delta_{xyz,Z}} X(\sigma_{xyz}), \tag{A.17}$$

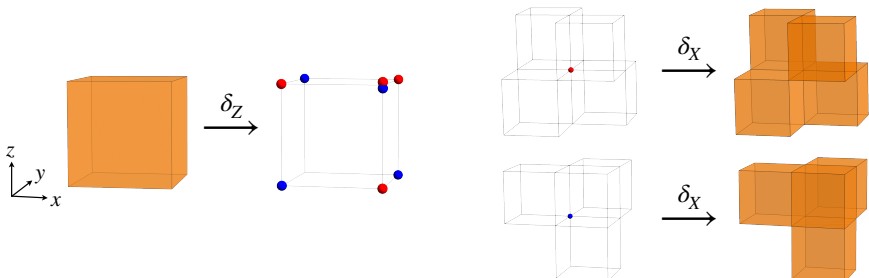

Figure 12: Haah's code.

on the symmetric Hilbert spaces as in (142). The second term in the Hamiltonian $\widetilde{H}_{\text{HC}}$ is the product over the vertices coloured blue and red in Figure 12. This is invariant under the fractal symmetry [98] (see [15,99] for a description of fractal symmetry using algebraic formalism).

On the other hand, the first term in the Hamiltonian $\widetilde{H}_{\text{HC,dual}}$ is the fractal Ising term, which is the product of four cubes as given in Figure 12. There are two types of fractal Ising term as depicted in the Figure 12. The Hamiltonian $\widetilde{H}_{\text{HC,dual}}$ is invariant under the fractal symmetry generated by $X(z)$ where $z \in C_{xyz,Z}$ with $\delta_Z z = 0$. The strange correlator gives the 3d fractal Ising model.

## B    Measurement-based gauging for a non-CSS code: The Chamon model

In the main text 4, we discussed chain complexes for CSS codes, the Kramers-Wannier duality, and correctability for preparing the CSS code or the dual CSS code states. In this section, we will discuss the Kramers-Wannier duality and correctability for the Chamon model, an example of non-CSS codes. We will also give a prescription to prepare a ground state of the Chamon model.[27]

Let us consider a cubic lattice. Qubits are placed on the vertices. Let us denote the set of vertices by $\mathbf{\Delta}_v$ and set of cubes by $\mathbf{\Delta}_c$. The Hamiltonian is a sum of stabilizers given by

$$H = -\sum_{\sigma_\beta \in \mathbf{\Delta}_c} \mathcal{O}(\sigma_\beta), \tag{B.1}$$

where $\mathcal{O}_c$ is defined as follows. Let $C_c$ and $C_v$ be the chain group formed by cells in $\mathbf{\Delta}_c$ and $\mathbf{\Delta}_v$. The stabilizers can be described by the chain complex

$$C_c \xrightarrow{\delta_c} C_v, \quad \text{and} \quad C_c \xrightarrow{\delta'_c} C_v, \tag{B.2}$$

where

$$\delta_c(\sigma_\beta = [x, x+1] \times [y, y+1] \times [z, z+1])$$
$$= \sigma_{(x,y,z+1)} + \sigma_{(x,y+1,z+1)} + \sigma_{(x+1,y,z)} + \sigma_{(x+1,y+1,z)}, \tag{B.3}$$
$$\delta'_c(\sigma_\beta = [x, x+1] \times [y, y+1] \times [z, z+1])$$
$$= \sigma_{(x,y,z)} + \sigma_{(x,y,z+1)} + \sigma_{(x+1,y+1,z)} + \sigma_{(x+1,y+1,z+1)}, \tag{B.4}$$

---

[27]A prescription for preparing the ground state of Wen's plaquette model and double semion model were given in [5]. These are also examples of non-CSS codes.

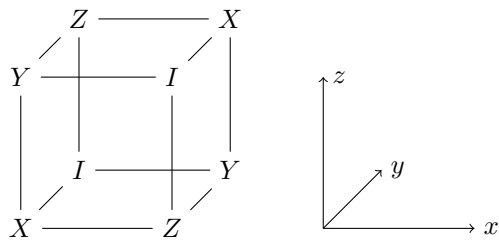

Figure 13: Stabilizer $\mathcal{O}$ of the Chamon model.

and

$$\mathcal{O}(\sigma_\beta) = -Z(\delta_c \sigma_\beta) X(\delta'_c \sigma_\beta). \tag{B.5}$$

The Chamon model has symmetries of the form $X(z_v^*)$ with $\delta_c^* z_v^* = 0$ and $Z(z_v')$ with $\delta_c'^* z_v' = 0$. Let us consider the Chamon model on a periodic lattice with the number of sites in $x$, $y$ and $z$ directions equal to $L$. Examples of $X(z_v^*)$ are

$$\prod_{j=0}^{L-1} X(\{(x,j,z)\}) \qquad \text{and} \qquad \prod_{j=0}^{L-1} X(\{(j,y,k-j)\}), \quad \text{for } 0 \le k < L. \tag{B.6}$$

Similarly examples of $Z(z_v')$ are

$$\prod_{j=0}^{L-1} Z(\{(x,y,j)\}) \qquad \text{and} \qquad \prod_{j=0}^{L-1} Z(\{(k+j,j,z)\}), \quad \text{for } 0 \le k < L. \tag{B.7}$$

We set the entangler as

$$\mathcal{U}_{CO} = \prod_{\sigma_i \in \Delta_v, \sigma_\beta \in \Delta_c} CO_{\sigma_\beta, \sigma_i}, \tag{B.8}$$

where

$$CO_{\sigma_\beta, \sigma_i} = |0\rangle_{\sigma_\beta} \langle 0| + |1\rangle_{\sigma_\beta} \langle 1| \otimes \mathcal{O}_{\sigma_i}, \tag{B.9}$$

and $\mathcal{O}_{\sigma_i} \in \{I, X, Y, Z\}$ depending on the position of the vertex $\sigma_i$ as depicted in Figure 13 so that $\mathcal{O}(\sigma_\beta) = \prod_{\sigma_i \subset \sigma_\beta} \mathcal{O}_{\sigma_i}$. It is easy to verify relations $X(\sigma_v)\mathcal{U}_{CO} = \mathcal{U}_{CO}X(\sigma_v)Z(\delta_c^* \sigma_v)$ and $X(\sigma_\beta)\mathcal{U}_{CO} = \mathcal{U}_{CO}\mathcal{O}(\sigma_\beta)X(\sigma_\beta)$.

We define the Kramers-Wannier operator as

$$\text{KW} = \langle+|^{\Delta_v} \mathcal{U}_{CO} |+\rangle^{\Delta_c}, \tag{B.10}$$

and

$$\text{KW}^\dagger = \langle+|^{\Delta_c} \mathcal{U}_{CO} |+\rangle^{\Delta_v}. \tag{B.11}$$

They satisfy the following relations for $\sigma_v \in \Delta_v$, $\sigma_\beta \in \Delta_c$, cycles $\delta_c z_c = 0$, and dual cycles $\delta_c^* z_v^* = 0$:

$$\text{KW}X(\sigma_v) = Z(\delta_c^* \sigma_v)\text{KW}, \tag{B.12}$$

$$\text{KW}\mathcal{O}(\sigma_\beta) = X(\sigma_\beta)\text{KW}, \tag{B.13}$$

$$\text{KW}X(z_v^*) = \text{KW}, \tag{B.14}$$

and

$$KW^\dagger X(\sigma_\beta) = \mathcal{O}(\sigma_\beta) KW^\dagger, \tag{B.15}$$

$$KW^\dagger Z(\delta_c^* \sigma_v) = X(\sigma_v) KW^\dagger, \tag{B.16}$$

$$KW^\dagger X(z_c) = \prod_{\sigma_\beta \in z_c} \mathcal{O}(\sigma_\beta) KW^\dagger. \tag{B.17}$$

The Kramers-Wannier operator converts Hamiltonians as

$$\widetilde{H}_{CM} = -\sum_{\sigma_\beta \in \Delta_c} \mathcal{O}(\sigma_\beta) - \lambda \sum_{\sigma_v \in \Delta_v} X(\sigma_v)$$

$$\underset{KW^\dagger}{\overset{KW}{\rightleftharpoons}} \quad \widetilde{H}_{CM,dual} = -\lambda \sum_{\sigma_v \in \Delta_v} Z(\delta_c^* \sigma_v) - \sum_{\sigma_\beta \in \Delta_c} X(\sigma_\beta), \tag{B.18}$$

on the symmetric Hilbert spaces. $\widetilde{H}_{CM}$ has symmetry $X(z_v^*)$ with $\delta_c^* z_v^* = 0$ and $\widetilde{H}_{CM,dual}$ has symmetry $X(z_c)$ with $\delta_c z_c = 0$. We can compute the composition

$$KW \circ KW^\dagger = \frac{1}{2^{|\Delta_c|}} \sum_{\delta_c z_c = 0} X(z_c), \tag{B.19}$$

and

$$KW^\dagger \circ KW = \frac{1}{2^{|\Delta_v|}} \sum_{\delta_v^* z_v^* = 0} X(z_v^*). \tag{B.20}$$

The state $KW^\dagger |+\rangle^{\Delta_c}$ is a ground state of the Chamon model Hamiltonian (B.1). It can be interpreted as the following procedure: (i) prepare the product state $|+\rangle^{\Delta_c}$, (ii) introduce ancillas $|+\rangle^{\Delta_v}$, (iii) apply the entangler $\mathcal{U}_{CO}$, and (iv) measure the cube degrees of freedom in the $X$ basis. At the fourth stage, if one obtains the outcome $\langle +|^{\Delta_c} Z(c_c)$ for some chain $c_c \in C_c$, the correction procedure can be done as follows. Consider the state after applying the entangler $\mathcal{U}_{CO}$

$$|\Psi_{pre}\rangle = \mathcal{U}_{CO} |\Psi\rangle |+\rangle^{\Delta_v}, \tag{B.21}$$

which is also symmetric

$$X(z_c) |\Psi_{pre}\rangle = |\Psi_{pre}\rangle. \tag{B.22}$$

One can invoke the argument involving non-degenerate intersection pairing as we discussed in the CSS case in Section 4.1, which implies $c_c = \delta_c^* c_v$. The correction can be done by applying $X(c_v)$ on the post-measurement state because

$$X(c_v) \langle +|^{\Delta_c} Z(c_c) |\Psi_{pre}\rangle = \langle +|^{\Delta_c} Z(c_c) \mathcal{U}_{CO} Z(\delta_c^* c_v) X(c_v) |\Psi\rangle |+\rangle^{\Delta_v} = \langle +|^{\Delta_c} |\Psi_{pre}\rangle. \tag{B.23}$$

## C Argument for SPT order via gauging

Here we provide another piece of evidence to show that the foliated cluster state from a given local CSS code defined on a lattice has an SPT order. Following [37, 39], we consider gauging the global symmetries (55) and (56). The overview of the argument is as follows. We define a gauging map $\Gamma$, which is locality preserving and gap preserving, such that the global symmetries (55) and (56) becomes trivial in the image of the map. We will take two different ungauged Hamiltonians $H_1$ and $H_2$, and obtain two gauged Hamiltonians $H_1^\Gamma$ and $H_2^\Gamma$, respectively. If there exists a path [1] generated by local unitary transformations between two gauged

Hamiltonians, then there also exits a symmetry-preserving path between two ungauged Hamiltonians. We demonstrate that gauging the foliated cluster state Hamiltonian $H_{\mathcal{C}}$, which is the sum of stabilizers (up to minus signs), yields the Hadamard transform of $H_{\mathcal{C}}$. As the ground state of $H_{\mathcal{C}}$ is short-range entangled, so is the ground state of the gauged Hamiltonian. For comparison, we also gauge a trivial Hamiltonian $H_{\text{trivial}}$ which is symmetric under (55) and (56) and whose gapped ground state is a symmetric product state. We show that the gauged Hamiltonian describes a model with a pair of non-trivial logical operators that intersect with respect to the foliated direction when the original CSS code has non-trivial logical operators. Thus the trivial Hamiltonian becomes a topologically-ordered model upon gauging. As the two gauged models belong to different topological orders, the two original models have to belong to different SPT orders. Since the model with $H_{\text{trivial}}$ belongs to the trivial SPT order, then the foliated cluster state must possess a nontrivial SPT order.

Let us begin by defining the gauging map $\Gamma$. We make use of the foliated chain complex and the lattice is assumed to be periodic in the $w$-direction. The models to be gauged are defined on $\Delta_{Q_1} \cup \Delta_{Q_2}$. We write the basis for a wave function in $\mathcal{H}_{Q_1} \otimes \mathcal{H}_{Q_2}$ as $|c_{Q_1}\rangle \otimes |c_{Q_2}\rangle$. The gauging map $\Gamma : \mathcal{H}_{Q_1} \otimes \mathcal{H}_{Q_2} \to \mathcal{H}_{Q_1} \otimes \mathcal{H}_{Q_2}$ is defined at the level of basis as

$$|c_{Q_1}\rangle \otimes |c_{Q_2}\rangle \xrightarrow{\Gamma} |\delta^* c_{Q_2}\rangle \otimes |\delta c_{Q_1}\rangle. \tag{C.1}$$

Note that the locality of the map is ensured by the locality of differentials. The map is extended to arbitrary wave functions in the Hilbert space by linearity. Operators acting on the Hilbert space is gauged according to

$$\mathcal{O}^{\Gamma}(\Gamma|\psi\rangle) = \Gamma(\mathcal{O}|\psi\rangle). \tag{C.2}$$

In particular, the symmetry generators $X(z)$ in (55) and $X(z^*)$ in (56) in the original models are gauged:

$$X(z)^{\Gamma} = X(z^*)^{\Gamma} = 1. \tag{C.3}$$

The image of the map is subject to emergent symmetry conditions:

$$Z(z)(\Gamma|\psi\rangle) = Z(z^*)(\Gamma|\psi\rangle) = (\Gamma|\psi\rangle), \qquad \forall|\psi\rangle \in \mathcal{H}_{Q_1} \otimes \mathcal{H}_{Q_2}, \tag{C.4}$$

with $\delta z = 0$ ($z \in C_{Q_1}$) and $\delta z^* = 0$ ($z^* \in C_{Q_2}$). This can be seen from $Z(z)|\delta^* c_{Q_2}\rangle = (-1)^{\#(z \cap \delta^* c_{Q_2})}|\delta^* c_{Q_2}\rangle = |\delta^* c_{Q_2}\rangle$ due to $\#(z \cap \delta^* c_{Q_2}) = \#(\delta z \cap c_{Q_2}) = 0$ etc.

We consider gauging the foliated cluster state Hamiltonian,

$$H_{\mathcal{C}} = -\sum_{\sigma \in \Delta_{Q_1}} K(\sigma) - \sum_{\tau \in \Delta_{Q_2}} K(\tau). \tag{C.5}$$

According to the definition (C.2), we obtain[28]

$$\left(H_{\mathcal{C}}\right)^{\Gamma} = H^{\otimes \Delta_{Q_1} \cup \Delta_{Q_2}} H_{\mathcal{C}} H^{\otimes \Delta_{Q_1} \cup \Delta_{Q_2}}. \tag{C.6}$$

Thus the ground state of the gauged Hamiltonian is $H^{\otimes \Delta_{Q_1} \cup \Delta_{Q_2}}|\psi_{\mathcal{C}}^{(\text{CSS})}\rangle$, which is short-range entangled [39, 58].

On the other hand, we consider the trivial Hamiltonian,

$$H_{\text{trivial}} = -\sum_{\sigma \in \Delta_{Q_1}} X(\sigma) - \sum_{\sigma \in \Delta_{Q_2}} X(\tau). \tag{C.7}$$

---

[28]For example, we have $\Gamma(K(\sigma)|c_{Q_1}\rangle \otimes |c_{Q_2}\rangle) = \Gamma((-1)^{\#(\delta \sigma \cap c_{Q_2})}|c_{Q_1} + \sigma\rangle \otimes |c_{Q_2}\rangle) = (-1)^{\#(\delta \sigma \cap c_{Q_2})}|\delta^* c_{Q_2}\rangle \otimes |\delta(c_{Q_1} + \sigma)\rangle$, while due to the relation $H^{\otimes \Delta_{Q_1} \cup \Delta_{Q_2}} K(\sigma) H^{\otimes \Delta_{Q_1} \cup \Delta_{Q_2}} = Z(\sigma)X(\delta \sigma)$ we calculate $Z(\sigma)X(\delta \sigma)\Gamma(|c_{Q_1}\rangle \otimes |c_{Q_2}\rangle) = Z(\sigma)X(\delta \sigma)|\delta^* c_{Q_2}\rangle \otimes |\delta c_{Q_1}\rangle = (-1)^{\#(\sigma \cap \delta^* c_{Q_2})}|\delta^* c_{Q_2}\rangle \otimes |\delta c_{Q_1} + \delta \sigma\rangle$, thus they are equal. We note that $\#(\sigma \cap \delta^* c_{Q_2}) = \#(\delta \sigma \cap c_{Q_2})$.



Gauging this Hamiltonian gives

$$\left(H_{\text{trivial}}\right)^{\Gamma} = -\sum_{\boldsymbol{\sigma}\in\Delta_{Q_1}} X(\delta\boldsymbol{\sigma}) - \sum_{\boldsymbol{\sigma}\in\Delta_{Q_2}} X(\delta^*\boldsymbol{\tau}). \tag{C.8}$$

To fix the ground state of the gauged Hamiltonian according to the symmetry conditions, we add the $Z$ terms, $Z(\delta^*\boldsymbol{\sigma}')$ $(\boldsymbol{\sigma}' \in \Delta_X)$ and $Z(\delta\boldsymbol{\tau}')$ $(\boldsymbol{\tau}' \in \Delta_{Z,w})$. The upshot is two decoupled models described by

$$H_{Q_1} = -\sum_{\boldsymbol{\tau}\in\Delta_{Q_2}} X(\delta^*\boldsymbol{\tau}) - \sum_{\boldsymbol{\tau}'\in\Delta_{Z,w}} Z(\delta\boldsymbol{\tau}'), \tag{C.9}$$

$$H_{Q_2} = -\sum_{\boldsymbol{\sigma}\in\Delta_{Q_1}} X(\delta\boldsymbol{\sigma}) - \sum_{\boldsymbol{\sigma}'\in\Delta_X} Z(\delta^*\boldsymbol{\sigma}'). \tag{C.10}$$

The former is defined on $\Delta_{Q_1}$ and the latter on $\Delta_{Q_2}$. Each of them possesses a non-trivial topological order. To see this, we construct a pair of non-trivial logical operators whose mutual commutativity is susceptible to the background topology. Namely, in the model (or a new CSS code) $H_{Q_1}$, we have logical operators of the form $Z(\boldsymbol{z})$ and $X(\boldsymbol{z}^*)$ ($\delta\boldsymbol{z} = 0$, $\delta^*\boldsymbol{z}^* = 0$; $\boldsymbol{z} \in C_{Q_1}, \boldsymbol{z}^* \in C_{Q_1}^* \simeq C_{Q_1}$). An example of an anti-commuting pair is given by

$$\boldsymbol{z} = \sum_{j\in\{0,\dots,L_w-1\}} z_{\text{q}} \times [j, j+1] \quad (\delta_X z_{\text{q}} = 0, z_{\text{q}} \in C_{\text{q}}), \tag{C.11}$$

$$\boldsymbol{z}^* = z_{\text{q}}^* \times [w, w+1] \quad (\delta_Z^* z_{\text{q}}^* = 0, z_{\text{q}}^* \in C_{\text{q}}^*). \tag{C.12}$$

We assume that the cycles $z_{\text{q}}$ and $z_{\text{q}}^*$ have a non-zero intersection number. As the existence of the anomalous commutator between logical operators above is due to the nontrivial intersection between a slice in the $w$ direction and a cycle wrapping around the $w$ direction (analogous to the $\alpha$- and $\beta$-cycles on the torus), the gauged model has a topological order.

# D  Duality between strange correlators for CSS codes

The aim of this appendix is to provide a CSS generalization of the Kramers-Wannier-Wegner duality with some careful treatment of the global contributions that arise on finite lattices. As discussed in the main text, given a CSS code described by the chain complex 34, we obtain a classical partition function:

$$\langle\omega(K)|\Psi_{\text{CSS}}\rangle = 2^{-|\Delta_Z|}2^{-|\Delta_{\text{q}}|/2} \times \mathbf{Z}_Z, \tag{D.1}$$

where $\langle\omega(K)| = \bigotimes_{\sigma_{\text{q}}\in\Delta_{\text{q}}}\langle 0|e^{KX}$ is a product state, $|\Psi_{\text{CSS}}\rangle$ is a particular state (see below) stabilized by $\mathcal{S}_Z = \langle Z(\delta_Z\sigma_\beta), \sigma_\beta \in \Delta_Z\rangle$ as well as $\mathcal{S}_X = \langle X(\delta_X^*\sigma_\alpha), \sigma_\alpha \in \Delta_X\rangle$, and $\mathbf{Z}_Z$ is a partition function

$$\mathbf{Z}_Z = \sum_{\{s_{\sigma_\beta}=\pm1\}_{\sigma_\beta\in\Delta_Z}} \exp\left(K\sum_{\sigma_i\in\Delta_{\text{q}}} s(\delta_Z^*\sigma_i)\right). \tag{D.2}$$

This can be obtained from an explicit calculation using the commutation relation between the $CZ$ gate the $X$ operator etc. via the expression

$$|\Psi_{\text{CSS}}\rangle = \langle+|^{\Delta_Z} \prod_{\substack{\sigma_i\in\Delta_{\text{q}}\\\sigma_\beta\in\Delta_Z}} CZ_{\sigma_i,\sigma_\beta}^{a(\delta_Z\sigma_\beta;\sigma_i)}|+\rangle^{\Delta_{\text{q}}}|+\rangle^{\Delta_Z}. \tag{D.3}$$

Note that the CSS state is the $+1$ eigenstate of the $X$ operator including the $X$ logical operators, $X(z_{\text{q}}^*)|\Psi_{\text{CSS}}\rangle = |\Psi_{\text{CSS}}\rangle$ with $\delta_Z^* z_{\text{q}}^* = 0$ ($z_{\text{q}} \in C_{\text{q}}$).

Let us consider a different state constructed in a similar manner but with the differential $\delta_X$ instead of $\delta_Z$,

$$|\Psi'_{\text{CSS}}\rangle = \langle + |^{\Delta_X} \prod_{\substack{\sigma_i \in \Delta_q \\ \sigma_\alpha \in \Delta_X}} CZ^{a(\delta_X^* \sigma_\alpha; \sigma_i)}_{\sigma_i, \sigma_\alpha} |+\rangle^{\Delta_q} |+\rangle^{\Delta_X}. \tag{D.4}$$

Let $\mathsf{H} = H^{\otimes \Delta_q}$ be the simultaneous Hadamard transform. The state $|\Psi'_{\text{CSS}}\rangle$ is almost the same as $\mathsf{H}|\Psi_{\text{CSS}}\rangle$, as they are both stabilized by the same stabilizers, $\mathcal{S}'_X = \langle X(\delta_Z \sigma_\beta), \sigma_\beta \in \Delta_Z \rangle$ and $\mathcal{S}'_Z = \langle Z(\delta_X^* \sigma_\alpha), \sigma_\alpha \in \Delta_X \rangle$. However, the state $|\Psi'_{\text{CSS}}\rangle$ is stabilized by the logical operator $X(z_q)$ with $\delta_X z_q = 0$, $X(z_q)|\Psi'_{\text{CSS}}\rangle = |\Psi'_{\text{CSS}}\rangle$. On the other hand, $Z(z_q^*)\mathsf{H}|\Psi_{\text{CSS}}\rangle = \mathsf{H}|\Psi_{\text{CSS}}\rangle$ with $\delta_Z^* z_q^* = 0$. The difference can be accounted for by summing over logical operators:

$$|\Psi'_{\text{CSS}}\rangle = \frac{1}{|\mathcal{L}|} \left( \sum_{[\ell] \in \mathcal{L}} X(\ell) \right) \mathsf{H}|\Psi_{\text{CSS}}\rangle, \tag{D.5}$$

$$\mathsf{H}|\Psi_{\text{CSS}}\rangle = \frac{1}{|\mathcal{L}^*|} \left( \sum_{[\ell^*] \in \mathcal{L}^*} Z(\ell^*) \right) |\Psi'_{\text{CSS}}\rangle, \tag{D.6}$$

where $\mathcal{L} = \text{Ker}\,\delta_X / \text{Im}\,\delta_Z$ and $\mathcal{L}^* = \text{Ker}\,\delta_Z^* / \text{Im}\,\delta_X^*$ and $[\bullet]$ is the homology class represented by $\bullet$. We note that

$$\langle \omega(K)|\mathsf{H} = (\sinh 2K)^{|\Delta_q|/2} \langle \omega(K^*)|, \tag{D.7}$$

where $K^* = -\frac{1}{2} \log \tanh K$. From the identity $\langle \omega(K)|\Psi_{\text{CSS}}\rangle = \langle \omega(K)|\mathsf{H} \cdot \mathsf{H}|\Psi_{\text{CSS}}\rangle$ and (D.6), we get the relation

$$\mathbf{Z}_Z(K) = \frac{2^{|\Delta_Z|}(\sinh 2K)^{|\Delta_q|/2}}{2^{|\Delta_X|}|\mathcal{L}^*|} \sum_{[\ell^*] \in \mathcal{L}^*} \mathbf{Z}_X^{\text{twisted}}(K^*, \ell^*), \tag{D.8}$$

where we introduced the twisted partition function given by

$$\mathbf{Z}_X^{\text{twisted}}(K^*, \ell^*) = \sum_{\{s_{\sigma_\alpha} = \pm 1\}_{\sigma_\alpha \in \Delta_X}} \exp\left( K^* \sum_{\sigma_q} (-1)^{\#(\sigma_q \cap \ell^*)} s(\delta_X \sigma_q) \right). \tag{D.9}$$

As we discussed in the main text in Section 4.3, we can interpret the strange correlator overlap $\langle \omega(K)|\Psi_{\text{CSS}}\rangle$ as specifying the boundary conditions in the SymTFT in the following sense:

1. Topological boundary condition specified by $|\Psi_{\text{CSS}}\rangle$,

2. Dynamical boundary condition specified by $|\omega(K)\rangle$.

This gives the partition function $\mathbf{Z}_Z(K) \propto \langle \omega(K)|\Psi_{\text{CSS}}\rangle$. Similarly, a different topological boundary condition specified by $\mathsf{H}|\Psi'_{\text{CSS}}\rangle$ gives us a different partition function as an overlap $\mathbf{Z}_X(K^*) \propto \langle \omega(K)|\mathsf{H}|\Psi'_{\text{CSS}}\rangle$. The two partition functions are related as in (D.8) implementing the Kramers-Wannier duality at the level of classical statistical models. The Kramers-Wannier duality can be understood as gauging the classical spin-flip symmetry of the respective models.

The classical spin-flip symmetries of the statistical model $\mathbf{T}_Z(K)$, defined by the partition function given by $\mathbf{Z}_Z(K)$, are a group

$$\mathcal{G}_Z = \{ F_Z(z_Z) | \delta_Z z_Z = 0 \}, \tag{D.10}$$

where the element $F_Z(z_Z)$ acts on spins via the action $F_Z(z_Z) s(\sigma_\beta) = (-1)^{\#(z_Z \cap \sigma_\beta)} s(\sigma_\beta)$ and the spins take values $s(\sigma_\beta) = \pm 1$ with $\sigma_\beta \in \Delta_Z$. Similarly one can consider the classical spin flip symmetry for the statistical model $\mathbf{T}_X(K^*)$ defined by the partition function $\mathbf{Z}_X(K^*)$

$$\mathcal{G}_X = \{ F_X(z_X^*) | \delta_X^* z_X^* = 0 \}, \tag{D.11}$$

where the element $\mathbf{F}_X(z_X^*)$ acts on spins via the action $F(z_X^*)s(\sigma_\alpha) = (-1)^{\#(z_X^* \cap \sigma_\alpha)}s(\sigma_\alpha)$ and the spins take values $s(\sigma_\alpha) = \pm 1$ with $\sigma_\alpha \in \Delta_X$. The equality (D.8) implies the duality relation

$$\mathbf{T}_Z(K) \simeq \mathbf{T}_X(K^*)/\mathcal{G}_X. \tag{D.12}$$

The same consideration based on (D.5) implies another duality relation

$$\mathbf{T}_X(K^*) \simeq \mathbf{T}_Z(K)/\mathcal{G}_Z. \tag{D.13}$$

As in Section 4.3, we find that the Kramers-Wannier transformation is implemented by exchanging the topological boundary conditions specified by $\mathsf{H}|\Psi_{\text{CSS}}\rangle$ and $|\Psi'_{\text{CSS}}\rangle$ in the SymTFT.

# E Proof of non-degeneracy for the intersection pairing between homology classes

In this appendix, we prove the non-degeneracy of the intersection pairing between homologies, a fact that we use in Section 4.1 to show the correctability of outcomes in the measurement-based gauging procedure.

Let $R$ be a commutative ring and consider a chain complex $C$ of free $R$-modules

$$\ldots \xrightarrow{\delta} C_{n+1} \xrightarrow{\delta} C_n \xrightarrow{\delta} C_{n-1} \xrightarrow{\delta} \ldots, \tag{E.1}$$

with $\delta^2 = 0$. The homology $R$-modules are given by $H_n(C,R) = \text{Ker}\,\delta/\text{Im}\,\delta$. We denote $\text{Ker}\,\delta$ by $Z_n$ and $\text{Im}\,\delta$ by $B_n$. This chain complex can be dualized by applying $\text{Hom}(-,R)$ to the chain complex. The cochain complex is given by

$$\ldots \xleftarrow{\delta^*} C_{n+1}^* \xleftarrow{\delta^*} C_n^* \xleftarrow{\delta^*} C_{n-1}^* \xleftarrow{\delta^*} \ldots \tag{E.2}$$

The cohomology $R$-modules are given by $H^n(C,R) = \text{Ker}\,\delta^*/\text{Im}\,\delta^*$. Now, let us consider the natural map between $H^n(C,R)$ and $\text{Hom}(H_n(C,R),R)$ [100]

$$h : H^n(C,R) \to \text{Hom}(H_n(C,R),R). \tag{E.3}$$

The map $h$ is defined as follows. Let $\phi \in H^n(C,R)$ be a dual cycle: $\delta^*\phi = 0$. This means that $\phi \circ \delta = 0$ — i.e., $\phi$ vanishes on the image of $\delta$ which is denoted by $B_n$. The restriction $\phi_0 = \phi|_{Z_n}$ induces a quotient homomorphism $\phi_0 : Z_n/B_n \to R$, and $\phi_0$ is an element of $\text{Hom}(H_n(C,R),R)$. The map $h$ sends $\phi$ to $\phi_0$. $h$ is a homomorphism and it is surjective [100]. The failure of $h$ to be an isomorphism is captured by the module $\text{Ext}_R^1(H_{n-1}(C,R),R)$ in the short exact sequence [101]

$$0 \to \text{Ext}_R^1(H_{n-1}(C,R),R) \to H^n(C,R) \xrightarrow{h} \text{Hom}(H_n(C,R),R) \to 0. \tag{E.4}$$

An $R$-module $B$ is injective if and only if $\text{Ext}_R^1(A,B) = 0$ for every $R$-module $A$ [101]. Since $\mathbb{Z}_n$ is an injective $\mathbb{Z}_n$ module [102], $\text{Ext}_R^1(H_{n-1}(C,R),R)$ vanishes and $h$ is an isomorphism if $R = \mathbb{Z}_n$. Now we restrict to the case $R = \mathbb{Z}_n$. Consider the canonical homomorphism

$$\Phi : C_n \times C_n^* \to R, \tag{E.5}$$

which send $(v,f) \to f(v)$. Now consider the same map at the level of homology

$$\Phi' : H_n \times H_n^* \to R, \tag{E.6}$$

which sends $([v],[f]) \rightarrow f(v)$. One can check that $\Phi'$ is well defined from the following observations

$$f(v + \delta v') = f(v) + f(\delta v') = f(v) + f \circ \delta v' = f(v), \tag{E.7a}$$

$$(f + \delta^* f')(v) = f(v) + \delta^*(f')(v) = f(v) + f' \circ \delta(v) = f(v) + f'(\delta(v)) = f(v). \tag{E.7b}$$

The last equality in the first line follows from the fact that $f \in \operatorname{Ker} \delta^*$ and the last equality in the second line follows from $v \in \operatorname{Ker} \delta$. The map $\Phi'$ is clearly a non-degenerate map. We define the intersection pairing between homology and cohomology classes as the composition

$$H_n \times H^n \xrightarrow{(id,h)} H_n \times H_n^* \xrightarrow{\Phi'} R, \tag{E.8}$$

which is non-degenerate because $(id, h)$ is an isomorphism.

# F  Algebraic formalism for various spin models

In this section, we will look at the algebraic formalism developed by J. Haah for translationally invariant Pauli-stabilizer Hamiltonians [103]. Following [99], we first review the Kramers-Wannier duality in the algebraic formalism in Section F.1. In Section F.2, we give a general proof of correctability of the measurement outcomes in measurement based preparation of ground states of CSS codes in algebraic formalism. Finally in Section F.3, we provide examples of self-dual classical codes $\mathbf{CC}_{(d,k)}^{\text{self-dual}}$ together with their symmetry generators and ground state degeneracy.

## F.1  Kramers-Wannier duality in algebraic formalism

Here we provide a brief review of algebraic formalism for translationally invariant Pauli Hamiltoninas [103] and a review of Kramers-Wannier duality between quantum spin models in this formalism [99].

### F.1.1  Generating map and excitation map

Let us consider a hypercubic lattice in $d$ dimensions. We denote the coordinate directions of the lattice by $x_1, \ldots, x_d$. We will restrict to periodic hypercubic lattice throughout this Appendix and denote the number of unit cells in each of the $x_1, \ldots, x_d$ directions by $L_1, \ldots, L_d$. We assume a single-qubit degree of freedom at each site of the lattice and a two-dimensional Hilbert space $\mathcal{H}_i$. We assume there are $K$ sites per unit cell in the lattice. We represent the total Hilbert by $\otimes_i \mathcal{H}_i$. Pauli operators are acting on each d.o.f.

We start with a Hamiltonian defined on a hypercubic lattice with $K$ sites per unit cell, expressible by $r$ independent terms up to translation:

$$H = \sum_{i \in \mathbb{Z}_{L_1} \times \ldots \times \mathbb{Z}_{L_d}} \left( w_1 H_i^{(1)} + \ldots + w_r H_i^{(r)} \right). \tag{F.1}$$

Here $H_i^{(j)}$ are Pauli operators and $w_i$ are real coefficients.

We will use the algebraic representation of a Pauli operator defined as follows. Let us denote by $\mathbb{F}_2[x_1, \ldots, x_d]$ the ring of polynomials of $x_1, \ldots, x_d$ with coefficients in $\mathbb{F}_2$ ($\mathbb{Z}_2$ regarded as a field). First we define a ring

$$\mathbf{R} := \frac{\mathbb{F}_2[x_1, \ldots, x_d]}{\langle x_1^{L_1} + 1, \ldots, x_d^{L_d} + 1 \rangle}, \tag{F.2}$$

where we quotient out by an ideal generated by the $d$ polynomials representing identifications in the torus. Let $Z_{i,k}$ and $X_{i,k}$ ($k = 1, \ldots, K$) be the Pauli $Z$- and $X$-operators for the $k$-th site in the unit cell located at $i = (i_1, \ldots, i_d)$. Then we define the algebraic representation of the Pauli operator $P = \otimes_{i \in \mathbb{Z}_{L_1} \times \ldots \times \mathbb{Z}_{L_d}} \otimes_{k=1}^{K} Z_{i,k}^{a_{i,k}} X_{i,k}^{b_{i,k}}$ to be

$$
p = \begin{pmatrix} \sum_i a_{i,1} x_1^{i_1} \ldots x_d^{i_d} \\ \vdots \\ \sum_i a_{i,K} x_1^{i_1} \ldots x_d^{i_d} \\ \hline \sum_i b_{i,1} x_1^{i_1} \ldots x_d^{i_d} \\ \vdots \\ \sum_i b_{i,K} x_1^{i_1} \ldots x_d^{i_d} \end{pmatrix} \in \mathbf{R}^{2K} =: \mathbf{P}. \tag{F.3}
$$

Note that product of Pauli operators correspond to addition of elements in $\mathbf{R}^{2K}$.

Let $h^{(j)}$ be the algebraic representation of $H_0^{(j)}$ and consider the $\mathbf{R}$-module $\mathbf{H}$ generated by $\{h^{(j)}\}_{j=1}^{r}$. Let $\{\sigma^{(j)}\}_{j=1}^{T}$ be a minimal set of generators of $\mathbf{H}$ and $\Sigma^{(j)}$ the operator represented by $\sigma^{(j)}$. We define the generator-label module as $\mathbf{G} \equiv \mathbf{R}^T$. Then the generating map is defined as a map $\sigma : \mathbf{G} \to \mathbf{P}$ represented by the $2K \times T$ matrix

$$
\sigma = \begin{pmatrix} \sigma^{(1)} & \cdots & \sigma^{(T)} \end{pmatrix}, \tag{F.4}
$$

which we denote by the same symbol as the map. The terms in the Hamiltonian (F.1) are given by a specific choice of vectors (generator labels) in $\mathbf{G}$. Thus a generating map can be thought of as a matrix which, when acting on the generator labels, gives all the terms in the Hamiltonian.

The commutation value of operators $A$ and $B$, which we represent algebraically as $\mathbf{A}$ and $\mathbf{B}$, is defined as the symplectic inner product

$$
\langle \mathbf{A}, \mathbf{B} \rangle = \mathbf{A}^{\dagger} \lambda_K \mathbf{B} \in \mathbf{R}, \tag{F.5}
$$

where $\dagger$ denote matrix transpose followed by the antipode map $x_i \to \bar{x}_i \equiv x_i^{-1}$ and $\lambda_K$ is the symplectic form

$$
\lambda_K := \begin{pmatrix} 0_{K \times K} & \mathbb{I}_{K \times K} \\ \mathbb{I}_{K \times K} & 0_{K \times K} \end{pmatrix}. \tag{F.6}
$$

The coefficient of $x_1^{i_1} \ldots x_d^{i_d}$ in $\langle \mathbf{A}, \mathbf{B} \rangle$ equals $n_{ZX,i} + n_{XZ,i}$ modulo 2, where $n_{ZX,i}$ ($n_{XZ,i}$) is the number of Pauli $Z$'s ($X$'s) from $B$ appearing in the position translated by $i = (i_1, \ldots, i_d)$ relative to the positions of the Pauli $X$'s ($Z$'s) from $A$. We define the excitation map $\epsilon : \mathbf{P} \to \mathbf{E} \equiv \mathbf{R}^T$ by its matrix representation

$$
\epsilon := \sigma^{\dagger} \lambda_K, \tag{F.7}
$$

denoted by the same symbol. One can read off where anti-commutation occurs between the operators $\Sigma^{(j)}$ and $P$ from $\epsilon(p)$. $\mathbf{E}$ is called the excitation module.

### F.1.2   Symmetries and identity generators

With the generating map and the excitation map, we have the sequence

$$
\mathbf{G} \xrightarrow{\sigma} \mathbf{P} \xrightarrow{\epsilon} \mathbf{E}. \tag{F.8}
$$

Here, $\mathbf{G}$, $\mathbf{P}$ and $\mathbf{E}$ are vector spaces over $\mathbb{F}_2$ with basis elements of the form $x_1^{i_1} x_2^{i_2} \ldots x_d^{i_d}$. They are also $\mathbf{R}$ modules. Hence $\sigma$ and $\epsilon$ are both linear maps as well as module homomorphisms.

The operators represented by the elements of $\mathrm{Ker}(\sigma)$ are called identity generators [99]. The elements of $\mathrm{Ker}(\epsilon)$ represent the symmetry generators (the Pauli operators which commute with $\Sigma^{(j)}$ for all $j$). $\mathrm{Ker}(\sigma)$ and $\mathrm{Ker}(\epsilon)$ are submodules and subspaces of the domains of the respective maps. Choosing a basis for the subspaces determine the independent identity and symmetry generators. For stabilizer Hamiltonians all the terms in the Hamiltonian commute and hence $\langle \sigma, \sigma \rangle = 0 = \epsilon \circ \sigma$.

### F.1.3 KW duality between generalized transverse field Ising models

The Hamiltonian of the generalized Ising model that we are going to discuss consists of two types of terms: $K$ transverse fields and $N$ Ising interactions. We algebraically represent them by $\mathbf{X}_k$ ($k = 1,\ldots,K$) and $\mathbf{Z}_n$ ($n = 1,\ldots,N$), respectively. Let us define a $2K \times N$ matrix $\mathbf{Z} = \begin{pmatrix} \mathbf{Z}_1 & \ldots & \mathbf{Z}_N \end{pmatrix}$ and

$$\mathbf{X} = \begin{pmatrix} \mathbf{X}_1 & \ldots & \mathbf{X}_K \end{pmatrix} = \begin{pmatrix} 0_{K \times K} \\ \mathbf{I}_{K \times K} \end{pmatrix}. \tag{F.9}$$

The generating map for this model is

$$\sigma = \begin{pmatrix} \mathbf{Z} & \mathbf{X} \end{pmatrix}. \tag{F.10}$$

Due to the transverse fields, $\langle \sigma, \sigma \rangle$ and $\epsilon \circ \sigma$ are non-zero. In the main text, we defined the Kramers-Wannier transformation in (138) and derived the operator relations (140)- (144). In the algebraic formalism, the KW transformation corresponds to the dual generating map given by $\tilde{\sigma} = \begin{pmatrix} \tilde{\mathbf{X}} & \tilde{\mathbf{Z}} \end{pmatrix}$ where [99]

$$\tilde{\mathbf{X}} = \begin{pmatrix} 0_{N \times N} \\ \mathbf{I}_{N \times N} \end{pmatrix}, \qquad \tilde{\mathbf{Z}} = \begin{pmatrix} \langle \mathbf{Z}, \mathbf{X} \rangle \\ 0_{N \times K} \end{pmatrix}. \tag{F.11}$$

The dual excitation map is given by $\tilde{\epsilon} = \tilde{\sigma}^\dagger \lambda_N$. The dual excitation map satisfies the relations $\tilde{\epsilon} \circ \tilde{\sigma} = \langle \tilde{\sigma}, \tilde{\sigma} \rangle = \langle \sigma, \sigma \rangle = \epsilon \circ \sigma$. The generating map and the dual generating map satisfy the following identities

$$\mathrm{Ker}(\tilde{\epsilon}) = \tilde{\sigma}(\mathrm{Ker}(\sigma)), \qquad \mathrm{Ker}(\epsilon) = \sigma(\mathrm{Ker}(\tilde{\sigma})). \tag{F.12}$$

See Appendix A of [99] for a proof.

### F.1.4 Example: Transverse field Ising model on a one dimensional lattice

Here we give an example of Kramers-Wannier transformation for transverse field Ising model on a one dimensional lattice with $L$ sites. Interested readers can find more examples and detailed analysis in [99]. We consider the following Hamiltonian

$$\widetilde{H}_{\mathrm{Ising}} = -\sum_i Z_i Z_{i+1} - \lambda \sum_i X_i. \tag{F.13}$$

So we have the number of sites per unit cell $K = 1$ and the number of Ising type interactions $N = 1$. The generating map is given by

$$\sigma = \begin{pmatrix} 1+x & 0 \\ 0 & 1 \end{pmatrix}, \tag{F.14}$$

whose two columns represent $Z_0 Z_1$ and $X_0$. The terms in the Hamiltonian are generated by the generating labels $g_1 = \begin{pmatrix} 1 \\ 0 \end{pmatrix}$ and $g_2 = \begin{pmatrix} 0 \\ 1 \end{pmatrix}$. The kernel $\mathrm{Ker}(\sigma)$ is generated by $\begin{pmatrix} \sum_i x^i \\ 0 \end{pmatrix}$, and

corresponds to the product of the Ising interactions $Z_i Z_{i+1}$ on all sites $i$. This product is the identity since $Z_i^2 = 1$. So, indeed Ker$(\sigma)$ gives the identity generator. The excitation map is

$$\epsilon = \sigma^\dagger \lambda_1 = \left( \begin{array}{c|c} 1+\bar{x} & 0 \\ 0 & 1 \end{array} \right) \left( \begin{array}{cc} 0 & 1 \\ 1 & 0 \end{array} \right) = \left( \begin{array}{c|c} 0 & 1+\bar{x} \\ 1 & 0 \end{array} \right). \tag{F.15}$$

The symmetry generators of the Ising Hamiltonian (F.13) are represented by the elements of Ker$(\epsilon)$, which is generated by $\left( \begin{array}{c} 0 \\ \sum_i x^i \end{array} \right)$. This is indeed the product $\prod_i X_i$ which is the global symmetry of the Ising Hamiltonian (F.13). To find the KW dual model we note that

$$\tilde{\mathbf{X}} = \begin{pmatrix} 0 \\ 1 \end{pmatrix}, \qquad \tilde{\mathbf{Z}} = \begin{pmatrix} 1+\bar{x} \\ 0 \end{pmatrix}. \tag{F.16}$$

Then

$$\tilde{\sigma} = \begin{pmatrix} 0 & 1+\bar{x} \\ 1 & 0 \end{pmatrix}. \tag{F.17}$$

The generating map acting on the generating labels $g_1$ and $g_2$ gives the dual Hamiltonian which is the same as the Ising Hamiltonian. This is because the KW transformation exchanges the $X_i$ with the $Z_{i-1} Z_i$ term and the $Z_i Z_{i+1}$ term with the $X_i$ term. Ker$(\tilde{\sigma})$ is generated by $\left( \begin{array}{c} 0 \\ \sum_i x^i \end{array} \right)$ which is indeed the product of the Ising interactions at all the dual sites. The excitation map is given by

$$\tilde{\epsilon} = \left( \begin{array}{c|c} 1 & 0 \\ 0 & 1+x \end{array} \right). \tag{F.18}$$

Ker$(\tilde{\epsilon})$ is generated by $\left( \begin{array}{c} 0 \\ \sum_i x^i \end{array} \right)$ which represent $\prod_i X_i$ and is a symmetry of the KW dual Ising Hamiltonian. One can easily verify the identities Ker$(\tilde{\epsilon}) = \tilde{\sigma}(\text{Ker}(\sigma))$ and Ker$(\epsilon) = \sigma(\text{Ker}(\tilde{\sigma}))$ for this example.

### F.2 Correctability in algebraic formalism

In this subsection, we prove that the correction against measurement outcome in the preperation of CSS code ground states is possible. This procedure follows the discussion of correctability of measurement outcomes given in Section 4.1. Here, we discuss correctability in the algebraic formalism. Let us consider the generalized Ising model on an arbitrary dimensional torus with the following generating map and excitation map

$$\sigma = \begin{pmatrix} \mathbf{Z} & \mathbf{X} \end{pmatrix} = \left( \begin{array}{c|c} \sigma_z & 0 \\ 0 & I \end{array} \right), \qquad \epsilon = \sigma^\dagger \lambda_K = \left( \begin{array}{c|c} 0 & \sigma_z^\dagger \\ I & 0 \end{array} \right), \tag{F.19}$$

where $\sigma_z$ is a $K \times N$ matrix and $I$ is a $K \times K$ identity matrix. Note that here we chose $\mathbf{Z} = \begin{pmatrix} \sigma_z \\ 0 \end{pmatrix}$, i.e., $\mathbf{Z}$ does not have any Pauli $X$ operator in it. Referring to the discussion given in Section 4.1, the generating map $\sigma$ and excitation map $\epsilon$ can be written in terms of the chain complex notation as

$$\sigma = \left( \begin{array}{c|c} \delta_Z & 0 \\ 0 & I \end{array} \right), \qquad \epsilon = \left( \begin{array}{c|c} 0 & \delta_Z^* \\ I & 0 \end{array} \right), \tag{F.20}$$

where $\delta_Z$ should be thought of as boundary map acting on the polynomial representation of the chains in $C_Z$ to give chains in $C_q$. Similarly, $\delta_Z^*$ should be thought of as dual boundary maps acting on the chains in $C_q$ to give chains in $C_Z$. Explicitly this is exactly the maps $\sigma_z$

and $\sigma_z^\dagger$. This generalized Ising model is exactly the same quantum model considered in the L.H.S. of (142). The dual generating and excitation maps are given by

$$\tilde{\sigma} = \begin{pmatrix} 0 & \sigma_z^\dagger \\ I & 0 \end{pmatrix}, \qquad \tilde{\epsilon} = \left( \begin{array}{c|c} I & 0 \\ 0 & \sigma_z \end{array} \right). \tag{F.21}$$

In terms of the chain-complex notation, the dual generating and excitation maps are given by

$$\tilde{\sigma} = \begin{pmatrix} 0 & \delta_Z^* \\ I & 0 \end{pmatrix}, \qquad \tilde{\epsilon} = \left( \begin{array}{c|c} I & 0 \\ 0 & \delta_Z \end{array} \right). \tag{F.22}$$

The dual generalized Ising model is exactly the same quantum model appearing in the R.H.S. of (142). The blocks $\sigma_z$ and $\sigma_z^\dagger$ can be thought of as maps

$$\sigma_z : \mathbf{R}^N \to \mathbf{R}^K, \qquad \sigma_z^\dagger : \mathbf{R}^K \to \mathbf{R}^N. \tag{F.23}$$

They are both a module homomorphism as well as a linear map. Now we define a conjugation map

$$C : \mathbf{R} \to \mathbf{R}, \tag{F.24}$$

given by $x \to x^{-1}, y \to y^{-1}$ and $z \to z^{-1}$, and extending the definition to other elements in $\mathbf{R}$ to make it a ring homomorphism. $C$ is also a vectorspace isomorphism when $\mathbf{R}$ is viewed as a vectorspace over $\mathbb{F}_2$. Then, one can define

$$C_N := (C, \ldots, C) : \mathbf{R}^N \to \mathbf{R}^N, \qquad C_K := (C, \ldots, C) : \mathbf{R}^K \to \mathbf{R}^K, \tag{F.25}$$

as multiple copies of the ring homomorphism $C$. Clearly $C_N$ or $C_K$ are vectorspace isomorphisms. Now $\sigma_z^\dagger$ is defined by

$$\sigma_z^\dagger C_K(v) = C_N(\sigma_z^T v), \qquad \forall v \in \mathbf{R}^K. \tag{F.26}$$

This tells us that $\operatorname{Ker} \sigma_z^T \cong \operatorname{Ker} \sigma_z^\dagger$ as vectorspaces.

One can easily check the relations $\operatorname{Ker} \tilde{\sigma} \cong \operatorname{Ker} \epsilon \cong \operatorname{Ker} \sigma_z^\dagger = \operatorname{Ker} \delta_Z^*$ as well as the relations $\operatorname{Ker} \sigma \cong \operatorname{Ker} \tilde{\epsilon} \cong \operatorname{Ker} \sigma_z = \operatorname{Ker} \delta_Z$. This is consistent with the fact that $\operatorname{Ker} \delta_Z^*$ is the symmetries of the quantum model defined on the L.H.S. of (142), and $\operatorname{Ker} \epsilon \cong \operatorname{Ker} \sigma_z^\dagger$ is the symmetries of the same model in the algebraic formalism. Similarly, $\operatorname{Ker} \delta_Z$ is the symmetries of the quantum model defined on the R.H.S. of (142), and $\operatorname{Ker} \tilde{\epsilon} \cong \operatorname{Ker} \sigma_z$ is the symmetries of the same model in the algebraic formalism.

Let us start with the generalized Ising model given by $\sigma$. We define a charge configuration for this model as an element in $\mathbf{R}^K$ (Laurent polynomial). This polynomial is equivalent to specifying a measurement outcome $\langle +|^{\otimes \Delta_q} Z(c_q)$ as in the correction procedure mentioned in Section 4.1. To be precise, the operator $Z(c_q)$ can be represented by $\begin{pmatrix} v \\ 0 \end{pmatrix}$ where $v \in R^K$.

The symmetries of the Hamiltonian described by $\sigma$ are described by $\operatorname{Ker} \epsilon = \begin{pmatrix} 0 \\ \operatorname{Ker} \sigma_z^\dagger \end{pmatrix}$. This is equivalent to operators of the form $X(z_q^*)$ with $\delta_Z^* z_q^* = 0$. Now, let us define the symmetric charge configurations as the configurations $Z(c_q)$ which commute with $X(z_q^*) \ \forall z_q^*$ such that $\delta_Z^* z_q^* = 0$. The commutation property between two operators represented by $\begin{pmatrix} v \\ 0 \end{pmatrix}$ and $\begin{pmatrix} 0 \\ v' \end{pmatrix}$ in the algebraic formalism is captured by the vanishing inner product

$$\begin{pmatrix} v \\ 0 \end{pmatrix}^\dagger \lambda_K \begin{pmatrix} 0 \\ v' \end{pmatrix} = 0. \tag{F.27}$$

Let us denote $\begin{pmatrix} v \\ 0 \end{pmatrix} \equiv \mathbf{v}$ and $\begin{pmatrix} 0 \\ v' \end{pmatrix} \equiv \mathbf{v}'$. Then the vanishing inner product is the statement $\langle \mathbf{v}, \mathbf{v}' \rangle = 0$. Now, the symmetric charge configurations is given by $\mathbf{v}$ such that $v \in \mathbf{R}^K$ and $\langle \mathbf{v}, \mathbf{v}' \rangle = 0$ for all $\mathbf{v}'$ such that $v' \in \mathrm{Ker}\,\sigma_z^\dagger$. In other words, the symmetric charge configurations are represented by $\mathbf{v} = \begin{pmatrix} v \\ 0 \end{pmatrix}$ such that $v \in \left( \mathrm{Ker}\,\sigma_z^\dagger \right)^\perp$. Now let $\mathbf{w} = \begin{pmatrix} 0 \\ w \end{pmatrix}$ such that $w \in \mathrm{Ker}\,\sigma_z^\dagger$, and $\mathbf{u} = \begin{pmatrix} u \\ 0 \end{pmatrix}$ such that $u \in \mathbf{R}^N$. Consider the inner product

$$0 = \left\langle \begin{pmatrix} u \\ 0 \end{pmatrix}, \begin{pmatrix} 0 \\ \sigma_z^\dagger w \end{pmatrix} \right\rangle = \left\langle \begin{pmatrix} \sigma_z u \\ 0 \end{pmatrix}, \begin{pmatrix} 0 \\ w \end{pmatrix} \right\rangle . \tag{F.28}$$

So we find that $\mathrm{Im}\,\sigma_z \subset \left( \mathrm{Ker}\,\sigma_z^\dagger \right)^\perp$. As vector spaces, we know that $\left( \mathrm{Ker}\,\sigma_z^\dagger \right)^\perp \cong \mathrm{Im}\,\sigma_z^\dagger \cong \mathrm{Im}\,\sigma_z$ when they are finite dimensional.[29] This implies that

$$\mathrm{Im}\,\sigma_z = \left( \mathrm{Ker}\,\sigma_z^\dagger \right)^\perp . \tag{F.29}$$

Here $\mathrm{Im}\,\sigma_z$ is the vector space spanned by the independent Ising interactions. Hence the relation (F.29) implies that all the symmetric charge configurations $Z(c_q)$ obeying the condition $[Z(c_q), X(z_q^*)] = 0$ for all $z_q^*$ with $\delta_Z^* z_q^* = 0$ can be obtained by taking products of Ising interations $\prod_{\sigma_\beta \in A \subset \Delta_Z} Z(\delta_Z \sigma_\beta)$ for some subset $A \subset \Delta_Z$. By applying $\prod_{\sigma_\beta \in A} X(\sigma_\beta)$ we perform the correction procedure. In terms of algebraic representation, this correction procedure can be stated as follows: a symmetric charge configuration given by $\mathbf{v} = \begin{pmatrix} v \\ 0 \end{pmatrix}$, $v \in \mathbf{R}^K$ can be corrected by applying $\tilde{\mathbf{v}} = \begin{pmatrix} 0 \\ \tilde{v} \end{pmatrix}$, for $\tilde{v} \in \mathbf{R}^N$ satisfying $\sigma_z \tilde{v} = v$.

We mention that this argument of correction procedure also works in the case of preparing certain non-CSS stabilizer code ground states. Alternative transformations between (F.19) and

$$\widetilde{\sigma} = \begin{pmatrix} 0 & \sigma_z^\dagger \\ I & \sigma_x^\dagger \end{pmatrix}, \qquad \widetilde{\epsilon} = \left( \begin{array}{c|c} I & 0 \\ \sigma_x & \sigma_z \end{array} \right) , \tag{F.30}$$

are also valid Kramers-Wannier transformations; an example is the Chamon model with transverse field, for which the KW operators are given in (B.10) and (B.11) and the dual Hamiltonians are given in (B.18). We still have $\mathrm{Ker}\,\widetilde{\epsilon} \cong \mathrm{Ker}\,\sigma_z$ and $\mathrm{Ker}\,\widetilde{\sigma} \cong \mathrm{Ker}\,\sigma_z^\dagger$. We still have $\mathrm{Im}\,\sigma_z = \left( \mathrm{Ker}\,\sigma_z^\dagger \right)^\perp$ and the correction procedure can still be performed by applying the counter $X$ operators as before.

## F.3 Self-dual models

We consider a class of self-dual models specified by the chain complex

$$0 \longrightarrow C_{d-k} \xrightarrow{\delta_Z} C_k \longrightarrow 0 , \tag{F.31}$$

in $d$ spatial dimensions ($0 \leq k < \frac{d}{2}$). The Hamiltonian in terms of the chain complex notation is

$$H_{(d,k)}^{\mathbf{CS}} = - \sum_{\sigma_\beta \in \Delta_{d-k}} Z(\delta_Z \sigma_\beta) . \tag{F.32}$$

---

[29]These vector spaces are finite dimensional when we assume periodic boundary condition on the underlying lattice.

Below, we wish to study symmetries of the model, and compute the ground state degeneracy. For example, for the 1d Ising model $(d, k) = (1, 0)$, the ground state subspace is formed by $|\bar{0}\rangle := |00\ldots0\rangle$ and $|\bar{1}\rangle := |11\ldots1\rangle$. The two logical qubits are related by the logical $X$ operator $\prod_i X_i$, which is the global symmetry of the 1d Ising model. Hereafter, we describe the family of models using the algebraic formalism to avoid clutters. Also, irrelevant zero matrices due to the zero transverse field limit will be omitted.

### F.3.1 $d = 3$, $k = 1$ model

Let us denote the three coordinates in three dimensions by $x$, $y$, and $z$ with $L_x$, $L_y$, and $L_z$ the number of unit cells in the cubic lattice. We assume periodic boundary conditions on all the three directions. This model can be specified by the generating map,

$$\sigma = \left(\frac{\sigma_Z}{0_{3\times3}}\right), \qquad \sigma_Z = \begin{pmatrix} 1+y & 0 & 1+z \\ 1+x & 1+z & 0 \\ 0 & 1+y & 1+x \end{pmatrix}, \tag{F.33}$$

where each row corresponds to the qubit on an edge in the $x$-, $y$-, or $z$-direction, while each column corresponds to a face in the $xy$-, $yz$-, or $zx$-directions. The symmetries of this model can be obtained by computing

$$\mathrm{Ker}\,\epsilon = \left(\frac{0_{3\times4}}{\mathrm{Ker}\,\sigma_Z^\dagger}\right). \tag{F.34}$$

$\mathrm{Ker}\,\sigma_Z^\dagger$ is a submodule generated by the columns of the matrix

$$\begin{pmatrix} s_y s_z & 0 & 0 & 1+\bar{x} \\ 0 & s_x s_z & 0 & 1+\bar{y} \\ 0 & 0 & s_x s_y & 1+\bar{z} \end{pmatrix}, \tag{F.35}$$

where we define a notation

$$s_\alpha := \sum_{i=0}^{L_\alpha-1} \alpha^i \quad \text{for } \alpha = x, y, \ldots \tag{F.36}$$

The last column describes the local symmetry of the model, which consists of a product of six edges around every vertex. This local symmetry generate the one-form symmetries which lies in the trivial homology class. Note that, unlike the 3d toric code, operators that violate the local symmetry do not raise energy, so the model contains a significantly larger number of states in the ground state subspace. There are $L_x L_y L_z$ local symmetry generators. However, product of all of them is the identity operator, so there are $L_x L_y L_z - 1$ independent terms. Planar symmetries given in the first three columns of (F.35) give the one-form symmetries in the non-trivial homology class. Given two parallel planes in $xy$, $yz$, or $zx$ directions which consist of edges in $z$, $x$, and $y$ directions, respectively, the corresponding symmetry generator can be formed by a product of local symmetry generators along the plane. Hence, there are three independent generators from the first three columns. So there are $L_x L_y L_z - 1 + 3 = L_x L_y L_z + 2$ independent logical $X$ operators. We get the ground state degeneracy,

$$\log_2 \mathrm{GSD} = L_x L_y L_z + 2. \tag{F.37}$$

### F.3.2 $d = 4$, $k = 1$ model

Let us denote the four coordinates in four dimensions by $x$, $y$, $z$, and $w$. Let $L_x$, $L_y$, $L_z$, and $L_w$ be the number of unit cells in the $x$, $y$, $z$, and $w$ directions of the hypercubic lattice. We

assume the periodic boundary condition on all the four directions. The model is specified by the chain complex

$$0 \longrightarrow C_3 \xrightarrow{\delta_Z} C_1 \longrightarrow 0. \tag{F.38}$$

This can be explicitly written using the generating map $\sigma$,

$$\sigma = \left(\frac{\sigma_Z}{0_{4\times 4}}\right),$$

$$\sigma_Z = \begin{pmatrix} 0 & 1+z+w+zw & 1+y+w+yw & 1+y+z+yz \\ 1+z+w+zw & 0 & 1+x+w+xw & 1+x+z+xz \\ 1+y+w+yw & 1+x+w+xw & 0 & 1+x+y+xy \\ 1+y+z+yz & 1+x+z+xz & 1+x+y+xy & 0 \end{pmatrix}. \tag{F.39}$$

Each row from top to bottom represents the four edges $x$, $y$, $z$ and $w$ respectively in a unit cell, and each columns from left to right describes an Ising interaction obtained as the boundary of a cube perpendicular to the four directions $x$, $y$, $z$ and $w$ respectively. Global symmetries of this model can be obtained by computing $\mathrm{Ker}\,\epsilon \cong \mathrm{Ker}\,\sigma_Z^\dagger$. $\mathrm{Ker}\,\sigma_Z^\dagger$ is a submodule generated by the columns of

$$\begin{pmatrix} 0 & 0 & s_y(1+\bar{x}) & s_y(1+\bar{x}) & s_z(1+\bar{x}) & s_z(1+\bar{x}) & s_w(1+\bar{x}) & s_w(1+\bar{x}) \\ s_x(1+\bar{y}) & s_x(1+\bar{y}) & 0 & 0 & s_z(1+\bar{y}) & 0 & s_w(1+\bar{y}) & 0 \\ s_x(1+\bar{z}) & 0 & 0 & s_y(1+\bar{z}) & 0 & 0 & 0 & s_w(1+\bar{z}) \\ 0 & s_x(1+\bar{w}) & s_y(1+\bar{w}) & 0 & 0 & s_z(1+\bar{w}) & 0 & 0 \end{pmatrix}, \tag{F.40}$$

and

$$\begin{pmatrix} s_y s_z & s_y s_w & s_w s_z & 0 & 0 & 0 & 0 & 0 & 0 & 0 & 0 & 0 \\ 0 & 0 & 0 & s_x s_z & s_x s_w & s_w s_z & 0 & 0 & 0 & 0 & 0 & 0 \\ 0 & 0 & 0 & 0 & 0 & 0 & s_x s_y & s_x s_w & s_y s_w & 0 & 0 & 0 \\ 0 & 0 & 0 & 0 & 0 & 0 & 0 & 0 & 0 & s_x s_y & s_x s_z & s_y s_z \end{pmatrix}. \tag{F.41}$$

Every generator represented by each column in the above matrix can be placed at arbitrary positions in the remaining coordinate. For example, $s_x(1+\bar{y})$ can be placed at $y^j z^l w^m$ for any $(j,l,m)$ or $s_x s_y$ can be placed at $z^l w^m$ for any $(l,m)$. We note that not all the generators are independent as there are constraints between generators from different columns. For example, the first two columns of the second matrix satisfy the constraint

$$s_w \begin{pmatrix} s_y s_z \\ 0_3 \end{pmatrix} = s_z \begin{pmatrix} s_y s_w \\ 0_3 \end{pmatrix}. \tag{F.42}$$

Here $0_m$ denotes $m$ repeated entries of 0. Similarly there is a constraint between the first column of first matrix and fourth column of the second matrix

$$s_z \begin{pmatrix} 0 \\ s_x(1+\bar{y}) \\ s_x(1+\bar{z}) \\ 0 \end{pmatrix} = (1+\bar{y}) \begin{pmatrix} 0 \\ s_x s_z \\ 0 \\ 0 \end{pmatrix}. \tag{F.43}$$

The ground state degeneracy can be computed by counting all the independent symmetry generators. It is given by

$$\begin{aligned} \log_2 \mathrm{GSD} = {} & 2(L_x L_y L_z + L_x L_y L_w + L_x L_z L_w + L_y L_z L_w) \\ & - 2(L_x L_y + L_x L_z + L_z L_w + L_y L_z + L_y L_w + L_z L_w) + 4(L_x + L_y + L_z + L_w) - 8. \end{aligned} \tag{F.44}$$

We obtained this polynomial using a method with the Gröbner basis, which we explain in Appendix F.4.

### F.3.3  $d = 5$, $k = 1$ **model**

Let us denote the coordinates by $x_1$, $x_2,\ldots,$ and $x_5$. Let $L_i$ be the number of unit cell in the $i^{\text{th}}$ direction of the hypercubic lattice. We assume periodic boundary conditions along all the five directions. The model is specified by the chain complex

$$0 \longrightarrow C_4 \xrightarrow{\delta_Z} C_1 \longrightarrow 0. \tag{F.45}$$

There are five edges per unit cell and five Ising type interactions obtained as the boundary of a four dimensional hypercube perpendicular to one of the five directions. Let us denote

$$f_1^5(i,j,l) = (1+x_i)(1+x_j)(1+x_l). \tag{F.46}$$

The generating map is given by

$$\sigma = \left(\frac{\sigma_Z}{0_{5\times 5}}\right),$$

$$\sigma_Z = \begin{pmatrix} 0 & f_1^5(3,4,5) & f_1^5(2,4,5) & f_1^5(2,3,5) & f_1^5(2,3,4) \\ f_1^5(3,4,5) & 0 & f_1^5(1,4,5) & f_1^5(1,3,5) & f_1^5(1,3,4) \\ f_1^5(2,4,5) & f_1^5(1,4,5) & 0 & f_1^5(1,2,5) & f_1^5(1,2,4) \\ f_1^5(2,3,5) & f_1^5(1,3,5) & f_1^5(1,2,5) & 0 & f_1^5(1,2,3) \\ f_1^5(2,3,4) & f_1^5(1,3,4) & f_1^5(1,2,4) & f_1^5(1,2,3) & 0 \end{pmatrix}, \tag{F.47}$$

where each row from top to bottom represent an edge along one of the five directions in ascending order in the subscript of the coordinates. Each column represents a four-dimensional hypercube by deleting one of the five coordinates. They are ordered from left to right in the ascending order in the subscript of deleted coordinates. The symmetry of this model is given by $\operatorname{Ker}\epsilon \cong \operatorname{Ker}\sigma_Z^\dagger$, which is a submodule generated by vectors of the form

$$\begin{pmatrix} 1+\bar{x}_1 \\ 1+\bar{x}_2 \\ 1+\bar{x}_3 \\ 1+\bar{x}_4 \\ 1+\bar{x}_5 \end{pmatrix}, \qquad s_r \begin{pmatrix} 0_{l-1} \\ 1+\bar{x}_l \\ 0_{m-l-1} \\ 1+\bar{x}_m \\ 0_{5-m} \end{pmatrix}, \quad \text{and} \quad \begin{pmatrix} 0_{r-1} \\ s_l s_m \\ 0_{5-r} \end{pmatrix}, \tag{F.48}$$

where $r \neq l$, $r \neq m$, $l < m$, and we used

$$s_k := \sum_{i=0}^{L_k-1} x_k^i \quad (k = 1, 2, \ldots). \tag{F.49}$$

The $i$-th row represents edges in the $x_i$-direction. The ground state degeneracy can be obtained by counting the total number of independent symmetry generators:

$$\log_2 \text{GSD} = L_1 L_2 L_3 L_4 L_5 + 2\sum_{i=1}^{5} L_1 \ldots \hat{L}_i \ldots L_5 - 2 \sum_{i,j=1,i<j}^{5} L_1 \ldots \hat{L}_i \ldots \hat{L}_j \ldots L_5$$

$$+ 4 \sum_{i,j=1,i<j}^{5} L_i L_j - 8 \sum_{i=1}^{5} L_i + 14, \tag{F.50}$$

where the $\hat{L}_i$ denote $L_i$ is omitted in the product.

### F.3.4 $d = 5$, $k = 2$ model

The model is specified by the chain complex

$$0 \longrightarrow C_3 \xrightarrow{\delta_Z} C_2 \longrightarrow 0. \tag{F.51}$$

There are ten faces per unit cell and ten Ising type interactions described by ten cubes in a unit cell. The generating map is given by

$$\sigma = \left( \frac{\sigma_Z}{0_{10 \times 10}} \right), \tag{F.52}$$

$$\sigma_Z = \begin{pmatrix} 0 & 0 & 0 & 0 & 0 & 0 & 0 & 1+x_5 & 1+x_4 & 1+x_3 \\ 0 & 0 & 0 & 0 & 0 & 1+x_5 & 1+x_4 & 0 & 0 & 1+x_2 \\ 0 & 0 & 0 & 0 & 1+x_5 & 0 & 1+x_3 & 0 & 1+x_2 & 0 \\ 0 & 0 & 0 & 0 & 1+x_4 & 1+x_3 & 0 & 1+x_2 & 0 & 0 \\ 0 & 0 & 1+x_5 & 1+x_4 & 0 & 0 & 0 & 0 & 0 & 1+x_1 \\ 0 & 1+x_5 & 0 & 1+x_3 & 0 & 0 & 0 & 0 & 1+x_1 & 0 \\ 0 & 1+x_4 & 1+x_3 & 0 & 0 & 0 & 0 & 1+x_1 & 0 & 0 \\ 1+x_5 & 0 & 0 & 1+x_2 & 0 & 0 & 1+x_1 & 0 & 0 & 0 \\ 1+x_4 & 0 & 1+x_2 & 0 & 0 & 1+x_1 & 0 & 0 & 0 & 0 \\ 1+x_3 & 1+x_2 & 0 & 0 & 1+x_1 & 0 & 0 & 0 & 0 & 0 \end{pmatrix} \begin{matrix} f_{12} \\ f_{13} \\ f_{14} \\ f_{15} \\ f_{23} \\ f_{24} \\ f_{25} \\ f_{34} \\ f_{35} \\ f_{45} \end{matrix}.$$

In the above matrix, the rows from top to bottom correspond to the face $f_{ij}$ in the $x_i$-$x_j$ direction for $i < j$ in the ascending order. We define $f_{ij} < f_{rl}$ if $i < r$ or if $i = r$ then $j < l$. Similarly the columns from left to right correspond to cubic interactions formed without the coordinates $x_i$-$x_j$ for $i < j$ in the ascending order of deleted pair of coordinates. The submodule $\mathrm{Ker}\,\sigma_Z^\dagger$ representing symmetries is generated by the following three types of vectors. First,

$$\begin{pmatrix} 0_{\phi(p,q)-1} \\ s_l s_m s_n \\ 0_{10-\phi(p,q)} \end{pmatrix}, \tag{F.53}$$

where $\{l, m, n, p, q\} = \{1, 2, 3, 4, 5\}$, and $\phi(p, q)$ is the index of the row that represents $f_{pq}$ (or $f_{qp}$, whichever is appropriate) following the ascending order starting from 1. Second,

$$s_r s_t \begin{pmatrix} 0_{\phi(l,m)-1} \\ (1+\bar{x}_l)(1+\bar{x}_m) \\ 0_{\phi(l,n)-\phi(l,m)-1} \\ (1+\bar{x}_l)(1+\bar{x}_n) \\ 0_{10-\phi(l,n)} \end{pmatrix}, \tag{F.54}$$

where $\{r, t, l, m, n\} = \{1, 2, 3, 4, 5\}$ and $\phi(l, m) < \phi(l, n)$. Finally,

$$\begin{pmatrix} 1+\bar{x}_2 \\ 1+\bar{x}_3 \\ 1+\bar{x}_4 \\ 1+\bar{x}_5 \\ 0_6 \end{pmatrix}, \quad \begin{pmatrix} 1+\bar{x}_1 \\ 0_3 \\ 1+\bar{x}_3 \\ 1+\bar{x}_4 \\ 1+\bar{x}_5 \\ 0_3 \end{pmatrix}, \quad \begin{pmatrix} 0 \\ 1+\bar{x}_1 \\ 0_2 \\ 1+\bar{x}_2 \\ 0_2 \\ 1+\bar{x}_4 \\ 1+\bar{x}_5 \\ 0 \end{pmatrix}, \quad \begin{pmatrix} 0_2 \\ 1+\bar{x}_1 \\ 0_2 \\ 1+\bar{x}_2 \\ 0 \\ 1+\bar{x}_3 \\ 0 \\ 1+\bar{x}_5 \end{pmatrix}, \quad \begin{pmatrix} 0_3 \\ 1+\bar{x}_1 \\ 0_2 \\ 1+\bar{x}_2 \\ 0 \\ 1+\bar{x}_3 \\ 1+\bar{x}_4 \end{pmatrix}. \tag{F.55}$$

The above symmetries are all not independent. For example,

$$
s_3 s_4 s_5 \begin{pmatrix} 1 + \bar{x}_2 \\ 1 + \bar{x}_3 \\ 1 + \bar{x}_4 \\ 1 + \bar{x}_5 \\ 0_6 \end{pmatrix} = (1 + \bar{x}_2) \begin{pmatrix} s_3 s_4 s_5 \\ 0_9 \end{pmatrix},
\tag{F.56}
$$

and there are more constraints. The ground state degeneracy is given by

$$
\log_2 \mathrm{GSD} = 4 L_1 L_2 L_3 L_4 L_5 + 6.
\tag{F.57}
$$

### F.3.5 $\; d = 6$, $k = 1$ model

Let us denote the coordinates in six dimensions by $x_1$, $x_2,\ldots$, and $x_6$. Let $L_i$ be the number of unit cells in the $i^{th}$ direction of the hypercubic lattice. We assume periodic boundary conditions along all the six directions. The model is specified by the chain complex

$$
0 \longrightarrow C_5 \xrightarrow{\delta_Z} C_1 \longrightarrow 0.
\tag{F.58}
$$

There are six edges per unit cell and six Ising type interactions which are obtained by choosing five dimensional hypercubes perpendicular to any of the axes. Let us denote

$$
f_1^6(i, j, l, m) = (1 + x_i)(1 + x_j)(1 + x_l)(1 + x_m).
\tag{F.59}
$$

The generating map for this model is given by

$$
\sigma = \begin{pmatrix} \sigma_Z \\ \hline 0_{6 \times 6} \end{pmatrix},
\tag{F.60}
$$

$$
\sigma_Z = \begin{pmatrix}
0 & f_1^6(3,4,5,6) & f_1^6(2,4,5,6) & f_1^6(2,3,5,6) & f_1^6(2,3,4,6) & f_1^6(2,3,4,5) \\
f_1^6(3,4,5,6) & 0 & f_1^6(1,4,5,6) & f_1^6(1,3,5,6) & f_1^6(1,3,4,6) & f_1^6(1,3,4,5) \\
f_1^6(2,4,5,6) & f_1^6(1,4,5,6) & 0 & f_1^6(1,2,5,6) & f_1^6(1,2,4,6) & f_1^6(1,2,4,5) \\
f_1^6(2,3,5,6) & f_1^6(1,3,5,6) & f_1^6(1,2,5,6) & 0 & f_1^6(1,2,3,6) & f_1^6(1,2,3,5) \\
f_1^6(2,3,4,6) & f_1^6(1,3,4,6) & f_1^6(1,2,4,6) & f_1^6(1,2,3,6) & 0 & f_1^6(1,2,3,4) \\
f_1^6(2,3,4,5) & f_1^6(1,3,4,5) & f_1^6(1,2,4,5) & f_1^6(1,2,3,5) & f_1^6(1,2,3,4) & 0
\end{pmatrix},
$$

where each row from top to bottom represents an edge along one of the six directions in ascending order in the subscript of the coordinates. Each column represents a five-dimensional hypercube by deleting one of the six coordinates. They are arranged from left to right in the ascending order in the subscript of deleted coordinates. Symmetries are given by $\mathrm{Ker}\,\epsilon \cong \mathrm{Ker}\,\sigma_Z^\dagger$. $\mathrm{Ker}\,\sigma_Z^\dagger$ is a submodule generated by vectors of the form

$$
s_n \begin{pmatrix} 0_{l-1} \\ 1 + \bar{x}_l \\ 0_{m-l-1} \\ 1 + \bar{x}_m \\ 0_{6-m} \end{pmatrix}, \quad \text{with} \quad n \in \{1, \ldots, 6\} \setminus \{l, m\}, \quad 1 \le l < m \le 6,
\tag{F.61}
$$

and

$$
s_l s_m \begin{pmatrix} 0_{n-1} \\ 1 \\ 0_{6-n} \end{pmatrix}, \quad \text{with} \quad l < m \in \{1, \ldots, \hat{n}, \ldots, 6\}.
\tag{F.62}
$$

The ground state degeneracy is given by

$$
\log_2 \mathrm{GSD} = 4 \sum_{i=1}^{6} L_1 \ldots \hat{L}_i \ldots L_6 - 4 \sum_{i<j} L_1 \ldots \hat{L}_i \ldots \hat{L}_j \ldots L_6 + 6 \sum_{i<j<l} L_i L_j L_l
$$
$$
- 10 \sum_{i<j} L_i L_j + 16 \sum_{i} L_i - 24 , \tag{F.63}
$$

where the $\hat{L}_i$ denote $L_i$ is omitted in the product.

### F.3.6 $\quad d = 6, k = 2$ model

The model is specified by the chain complex

$$
0 \to C_4 \to C_2 \to 0 . \tag{F.64}
$$

There are 15 planes and 15 Ising type interactions obtained by choosing four dimensional hypersurfaces perpendicular to any of the 15 planes. Let us denote

$$
f_2^6(i,j) = (1+x_i)(1+x_j) . \tag{F.65}
$$

The generating map for this model is

$$
\sigma_Z = \begin{pmatrix}
0 & 0 & 0 & 0 & 0 & 0 & 0 & 0 & 0 & f_2^6(5,6) & f_2^6(4,6) & f_2^6(4,5) & f_2^6(3,6) & f_2^6(3,5) & f_2^6(3,4) \\
0 & 0 & 0 & 0 & 0 & 0 & f_2^6(5,6) & f_2^6(4,6) & f_2^6(4,5) & 0 & 0 & 0 & f_2^6(2,6) & f_2^6(2,5) & f_2^6(2,4) \\
0 & 0 & 0 & 0 & 0 & f_2^6(5,6) & 0 & f_2^6(3,6) & f_2^6(3,5) & 0 & f_2^6(2,6) & f_2^6(2,5) & 0 & 0 & f_2^6(2,3) \\
0 & 0 & 0 & 0 & 0 & f_2^6(4,6) & f_2^6(3,6) & 0 & f_2^6(3,4) & f_2^6(2,6) & 0 & f_2^6(2,4) & 0 & f_2^6(2,3) & 0 \\
0 & 0 & 0 & 0 & 0 & f_2^6(4,5) & f_2^6(3,5) & f_2^6(3,4) & 0 & f_2^6(2,5) & f_2^6(2,4) & 0 & f_2^6(2,3) & 0 & 0 \\
0 & 0 & f_2^6(5,6) & f_2^6(4,6) & f_2^6(4,5) & 0 & 0 & 0 & 0 & 0 & 0 & 0 & f_2^6(1,6) & f_2^6(1,5) & f_2^6(1,4) \\
0 & f_2^6(5,6) & 0 & f_2^6(3,6) & f_2^6(3,5) & 0 & 0 & 0 & 0 & 0 & f_2^6(1,6) & f_2^6(1,5) & 0 & 0 & f_2^6(1,3) \\
0 & f_2^6(4,6) & f_2^6(3,6) & 0 & f_2^6(3,4) & 0 & 0 & 0 & 0 & f_2^6(1,6) & 0 & f_2^6(1,4) & 0 & f_2^6(1,3) & 0 \\
0 & f_2^6(4,5) & f_2^6(3,5) & f_2^6(3,4) & 0 & 0 & 0 & 0 & 0 & f_2^6(1,5) & f_2^6(1,4) & 0 & f_2^6(1,3) & 0 & 0 \\
f_2^6(5,6) & 0 & 0 & f_2^6(2,6) & f_2^6(2,5) & 0 & f_2^6(1,6) & f_2^6(1,5) & 0 & 0 & 0 & 0 & 0 & 0 & f_2^6(1,2) \\
f_2^6(4,6) & 0 & f_2^6(2,6) & 0 & f_2^6(2,4) & 0 & f_2^6(1,6) & 0 & f_2^6(1,4) & 0 & 0 & 0 & 0 & f_2^6(1,2) & 0 \\
f_2^6(4,5) & 0 & f_2^6(2,5) & f_2^6(2,4) & 0 & 0 & f_2^6(1,5) & f_2^6(1,4) & 0 & 0 & 0 & 0 & f_2^6(1,2) & 0 & 0 \\
f_2^6(3,6) & f_2^6(2,6) & 0 & 0 & f_2^6(2,3) & f_2^6(1,6) & 0 & 0 & f_2^6(1,3) & 0 & 0 & f_2^6(1,2) & 0 & 0 & 0 \\
f_2^6(3,5) & f_2^6(2,5) & 0 & f_2^6(2,3) & 0 & f_2^6(1,5) & 0 & f_2^6(1,3) & 0 & 0 & f_2^6(1,2) & 0 & 0 & 0 & 0 \\
f_2^6(3,4) & f_2^6(2,4) & f_2^6(2,3) & 0 & 0 & f_2^6(1,4) & f_2^6(1,3) & 0 & 0 & f_2^6(1,2) & 0 & 0 & 0 & 0 & 0
\end{pmatrix} , \tag{F.66}
$$

where the rows correspond to face $f_{ij}$ in ascending order from top to bottom. Ascending order is defined similarly as in the $d = 5, k = 2$ case: $f_{ij} > f_{ml}$ if $i > m$ or if $i = m$ then $j > l$. Similarly the columns correspond to four dimensional hypercubes with $x_i$ and $x_j$ coordinates removed and are placed in ascending order from left to right. Ascending order is with respect to the removed coordinates $x_i$ and $x_j$ as in $d = 5, k = 2$ case. Let us define $g_{ij} := (1+\bar{x}_i)(1+\bar{x}_j)$. The submodule $\mathrm{Ker}\, \sigma_Z^\dagger$ representing symmetries is generated by the vectors of the form

$$
\begin{pmatrix} g_{12} & g_{13} & g_{14} & g_{15} & g_{16} & g_{23} & g_{24} & g_{25} & g_{26} & g_{34} & g_{35} & g_{36} & g_{45} & g_{46} & g_{56} \end{pmatrix}^T ,
$$

$$
s_r s_t \begin{pmatrix} 0_{\phi(l,q)-1} \\ (1+\bar{x}_l) \\ 0_{\phi(p,q)-\phi(l,q)-1} \\ (1+\bar{x}_p) \\ 0_{15-\phi(p,q)} \end{pmatrix} , \text{ with } l < p \text{ and } l,q,p,r,t \in \{1,\ldots,6\} \text{ distinct} ,
$$

$$
s_t \begin{pmatrix} 0_{\phi(l,m)-1} & g_{lm} & 0_{\phi(l,n)-\phi(l,m)-1} & g_{ln} & 0_{\phi(m,r)-\phi(l,n)-1} & g_{mr} & 0_{\phi(r,n)-\phi(m,r)-1} & g_{rn} & 0_{15-\phi(r,n)} \end{pmatrix}^T ,
$$
$$
\text{with } l < m < n \text{ and } l < r \text{ and } l,m,n,r,t \in \{1,\ldots,6\} \text{ distinct} ,
$$

$$
\text{and } s_l s_m s_n \begin{pmatrix} 0_{\phi(r,t)-1} \\ 1 \\ 0_{15-\phi(r,t)} \end{pmatrix} , \text{ with } l < m < n \in \{1,\ldots,\hat{r},\ldots,\hat{t},\ldots,6\} , \tag{F.67}
$$

where $\phi(p,q)$ is the index of the row that represents $f_{pq}$ (or $f_{qp}$, whichever is appropriate). The ground state degeneracy is given by

$$
\log_2 \text{GSD} = \prod_{i=1}^{6} L_i + 4 \sum_{i=1}^{6} L_1 \dots \hat{L}_i \dots L_6 - 4 \sum_{i<j} L_1 \dots \hat{L}_i \dots \hat{L}_j \dots L_6 + 8 \sum_{i<j<l} L_i L_j L_l
$$
$$
- 16 \sum_{i<j} L_i L_j + 34 \sum_{i} L_i - 74 . \tag{F.68}
$$

### F.4 Computation of ground state degeneracy using Gröbner basis method

Here we provide the details on the computation of the ground state degeneracy on the $d$ dimensional torus of various models that we considered above.

#### F.4.1 Classical model $\text{CC}_{(d,k)}^{\text{self-dual}}$

The Hamiltonian for this model is given in (F.32). The ground state degeneracy can be computed by counting the total number of independent symmetry generators. These independent symmetry generators can also be thought of as independent logical $X$ operators when $\text{CC}_{(d,k)}^{\text{self-dual}}$ is viewed as a quantum CSS code without any $X$-type stabilizer terms. Symmetries are given by columns of $\text{Ker}\,\sigma_Z^{\dagger}$. From (F.29), we know that $\text{Ker}\,\sigma_Z^{\dagger} = (\text{Im}\,\sigma_Z)^{\perp}$. Hence, to count the independent symmetry generators we need to simply find the dimension of $(\text{Im}\,\sigma_Z)^{\perp}$ inside the module $\mathbf{R}^{\binom{d}{k}}$ as an $\mathbb{F}_2$-vector space. Here $\mathbf{R} = \frac{\mathbb{F}_2[x_1,\dots,x_d]}{\langle x_1^{L_1}+1,\dots,x_d^{L_d}+1 \rangle}$ and $\binom{d}{k}$ is the number of rows or columns in the matrix $\sigma_Z$. Then the ground state degeneracy is given by

$$
\log_2 \text{GSD} = \dim_{\mathbb{F}_2} \frac{\left( \mathbb{F}_2[x_1,\dots,x_d]/\langle x_1^{L_1}+1,\dots,x_d^{L_d}+1 \rangle \right)^{\binom{d}{k}}}{\text{Im}\,\sigma_Z} . \tag{F.69}
$$

Now let us denote the rows of $(\mathbb{F}_2[x_1,\dots,x_d])^{\binom{d}{k}}$ when written as column vector by $\hat{e}_i$ for $i = 1,\dots,\binom{d}{k}$. Then (F.69) can be equivalently written as

$$
\log_2 \text{GSD} = \dim_{\mathbb{F}_2} \frac{(\mathbb{F}_2[x_1,\dots,x_d])^{\binom{d}{k}}}{\langle \text{Im}\,\sigma_Z, (x_1^{L_1}+1)\hat{e}_1,\dots,(x_d^{L_d}+1)\hat{e}_1,\dots,(x_1^{L_1}+1)\hat{e}_{\binom{d}{k}},\dots,(x_d^{L_d}+1)\hat{e}_{\binom{d}{k}} \rangle} , \tag{F.70}
$$

where the denominator on R.H.S. is a submodule generated by the elements in the angle brackets. The way to compute (F.70) is the following:

- First we find a Gröbner basis [85] for the submodule generated by $\text{Im}\,\sigma_Z$ as well as the $\binom{d}{k}$ boundary conditions $(x_1^{L_1}+1)\hat{e}_i,\dots,(x_d^{L_d}+1)\hat{e}_i$ for $i = 1,\dots,\binom{d}{k}$. This is a submodule of $(\mathbb{F}_2[x_1,\dots,x_d])^{\binom{d}{k}}$ and the Gröbner basis span this submodule.[30]

- We extract the leading monomials in the Gröbner basis for the submodule. The monomial order is defined as follows: $x_1^{\alpha_1} x_2^{\alpha_2} \dots x_d^{\alpha_d} \hat{e}_i > x_1^{\beta_1} x_2^{\beta_2} \dots x_d^{\beta_d} \hat{e}_j$ if (i) the first nonzero entry in the component-wise subtraction $(\alpha_1,\dots,\alpha_d) - (\beta_1,\dots,\beta_d)$ is a positive integer, or (ii) the subtraction vanishes and $i < j$.[31]

- Using the division algorithm for modules[32] all the elements in the quotient submodule has a leading monomial that is not divisible by all the leading monomials of the Gröbner

---

[30]See proposition 2.7 in chapter 5 of [85].

[31]This monomial ordering is defined as TOP order in chapter 5 of [85].

[32]This is stated as Theorem 2.5 in chapter 5 of [85].

basis that we found in the first step. Hence, counting the dimension of the quotient submodule is equivalent to counting all the monomials in $x_1, \ldots, x_d$ for each $\hat{e}_i$ that are not divisible by the leading monomials of the Gröbner basis.

### F.4.2 Quantum CSS code $QC_{(d,k,l)}^{\text{self-dual}}$

Hamiltonian for this model is given in (181). This can be represented in the algebraic notation with the generating map

$$\sigma = \begin{pmatrix} \sigma_Z & 0 \\ 0 & \sigma_X \end{pmatrix} \equiv \begin{pmatrix} \delta_Z & 0 \\ 0 & \delta_X^\dagger \end{pmatrix}, \tag{F.71}$$

where $\sigma_Z$ is a $\binom{d}{k} \times \binom{d}{k}$ matrix and $\sigma_X$ is a $\binom{d}{k} \times \binom{d}{l}$ matrix. The ground state degeneracy can be computed by counting the number of logical $Z$ operators. Given the chain complex (179), the ground state degeneracy is equal to the dimension of the homology

$$\log_2 \text{GSD} = \dim_{\mathbb{F}_2} \frac{\text{Ker}\, \delta_X}{\text{Im}\, \delta_Z} = \dim_{\mathbb{F}_2} \text{Ker}\, \delta_X - \dim_{\mathbb{F}_2} \text{Im}\, \delta_Z. \tag{F.72}$$

Now $\dim_{\mathbb{F}_2} \text{Im}\, \delta_Z$ can be computed using the formula

$$\dim_{\mathbb{F}_2} \text{Im}\, \delta_Z = \dim_{\mathbb{F}_2} \mathbf{R}^{\binom{d}{k}} - \dim_{\mathbb{F}_2} \text{Coker}\, \delta_Z \tag{F.73a}$$

$$= \binom{d}{k} L_1 \ldots L_d - \dim_{\mathbb{F}_2} \text{Coker}\, \delta_Z. \tag{F.73b}$$

Note that $\delta_Z \equiv \sigma_Z$ and we computed $\dim_{\mathbb{F}_2} \text{Coker}\, \sigma_Z$ in the classical code case using the Gröbner basis method. So it remains to compute $\dim_{\mathbb{F}_2} \text{Ker}\, \delta_X$. Using the exact argument we used in (F.28) to prove (F.29), we can prove $\text{Ker}\, \sigma_X^\dagger = (\text{Im}\, \sigma_X)^\perp$. With the identification $\sigma_X \equiv \delta_X^\dagger$, we get

$$\dim_{\mathbb{F}_2} \text{Ker}\, \delta_X = \dim_{\mathbb{F}_2} \left( \text{Im}\, \delta_X^\dagger \right)^\perp = \dim_{\mathbb{F}_2} \text{Coker}\, \delta_X^\dagger. \tag{F.74}$$

Combining (F.73) and (F.74), we conclude

$$\log_2 \text{GSD} = \dim_{\mathbb{F}_2} \text{Coker}\, \delta_X^\dagger + \dim_{\mathbb{F}_2} \text{Coker}\, \delta_Z - \binom{d}{k} L_1 \ldots L_d. \tag{F.75}$$

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
