# Peer review of "Anomaly inflow for CSS and fractonic lattice models and dualities via cluster state measurement"

_SciPost Physics, doi:SciPost Phys. 17, 113 (2024)_

## Round 1 · Referee Report · Anonymous (Referee 1) · 2024-8-6

Report

This paper studies the foliated construction of CSS codes first proposed in Phys. Rev. Lett. 117, 070501 [1]. While [1] mostly focused on the error correction properties of the codes, this paper discusses various physical properties of the constructed codes, including their potential SPT order, anomaly inflow, duality, and BF theory description.

The paper pointed out some potentially interesting features of the constructed code. For example, while the system with periodic boundary has unique ground state (as a cluster state), with open boundary in the foliation direction, some bulk stabilizers reduce to logical operators of the boundary CSS code and may anti-commute with each other. In this sense, the cluster state has 1d SPT order under Z2xZ2 (or copies of it) symmetry. While this is a very simple type of SPT order, it may be of interest to the QI community. Similarly, the paper discussed the idea of anomaly inflow, duality, strange correlator and field theory description for these models. While it is not immediately clear what these features can be useful for, it may help the QI community better learn these physics ideas which are usually discussed in a very different language.

I think this paper is a good addition to the literature and probably above the bar for publication in SciPost Physics. However, I think a major weakness of the paper is that it started the discussion from the very beginning using a sophisticated chain-complex formalism. The chain-complex language is a powerful language for discussing CSS codes, but not every one is familiar with it. To make things worse, notations are not always well-defined. For example, section 2.2 started by talking about "objects assigned to qubits i". It is not clear at all what these ojects are referring to. Basic terminologies associated with chain complex, like Ker, Im, and Hom, are not defined or explained. I think the paper will be much more readable if the authors do a more careful job explaining these ideas and illustrate them immediately using a simple example. It would be very helpful if the authors can use one (or two) example to illustrate the various ideas introduced in the paper (SPT order, anomaly inflow, duality etc). Right now, examples are clustered in later sections and different parts use different examples.

Overall, I think the result is worth publishing, but if the authors want more people to actually read it, it needs to be written in a much better way.

Recommendation

Ask for major revision

---

## Round 1 · Referee Report · Anonymous (Referee 2) · 2024-9-7

Strengths

  1. The paper successfully connects several important concepts in modern condensed matter theory, including fracton models, SPT order, anomaly inflow, and dualities.
  2. The authors provide detailed mathematical formulations, particularly in their use of chain complexes and their tensor products to describe foliated cluster states. This formulation provides versatility in describing a variety of different topological phases.
  3. The paper presents several new and important results, including: A generalized anomaly inflow mechanism for CSS codes; An extension of the Kramers-Wannier-Wegner duality for a broad class of models; A general construction of strange correlators for CSS codes; Identification of non-invertible symmetries in self-dual models.

Weaknesses

  1. While the mathematical rigor is a strength, it may also make the paper challenging for readers not deeply familiar with algebraic topology and homological algebra. Some additional explanatory text or intuitive descriptions could help broaden the paper's audience.
  2. The paper is quite lengthy. Therefore, sometimes it is hard to comprehend the connection between different parts of the paper. The authors have a brief discussion of summary of results in the beginning. But a expansion of this part can better guide the readers.
  3. The definition of strange correlator does not involve the notion of duality. In Section 4, the authors seem to make a connection between these two ideas. Can the authors elaborate the motivation for such connections preferably in the beginning of the section?

Report

This paper represents a solid contribution to the field, providing a unified framework for understanding a wide range of phenomena related to CSS codes and fracton models. The work is mathematically rigorous and presents several novel and important results. The overall quality and importance of the work are high. However, the presentation could be improved to enhance accessibility. Therefore, I would recommend publication of this paper possibly after some revisions to improve the presentation.

Recommendation

Ask for minor revision

---

## Round 2 · Author Response

We thank the two referees for positive assessment and useful suggestions on our manuscript. We have improved it accordingly. A major update is that we have expanded Introduction to illuminate our key results using the toric code and its foliation, namely, the Raussendorf-Bravyi-Harrington model. We also improved explanations of mathematical notations. We hope they find our updated manuscript suitable for publication.
Reply to Report 1: 1. [Referee comment] However, I think a major weakness of the paper is that it started the discussion from the very beginning using a sophisticated chain-complex formalism. The chain-complex language is a powerful language for discussing CSS codes, but not every one is familiar with it. (…) It would be very helpful if the authors can use one (or two) example to illustrate the various ideas introduced in the paper (SPT order, anomaly inflow, duality etc). Right now, examples are clustered in later sections and different parts use different examples. [Our reply] We thank the reviewer for a useful suggestion. We included a new section “1.2 Example: the toric code and the RBH model” to explain and summarize our key results using a prominent example. In this new section, we avoided using chain complex machinery to enhance accessibility.
- [Referee comment] To make things worse, notations are not always well-defined. For example, section 2.2 started by talking about "objects assigned to qubits i". It is not clear at all what these ojects are referring to. Basic terminologies associated with chain complex, like Ker, Im, and Hom, are not defined or explained. [Our reply] We thank the reviewer for useful comments. We added explanations to these concepts, in particular in Section 2.2. Please see the list of changes below.
Reply to Report 2: 1. [Referee comment] While the mathematical rigor is a strength, it may also make the paper challenging for readers not deeply familiar with algebraic topology and homological algebra. Some additional explanatory text or intuitive descriptions could help broaden the paper's audience. [Our reply] We thank the referee for pointing this out. We have added definitions of all the mathematical concepts which were undefined before. We also refer to the familiar toric code example to explain some concepts.
-
[Referee comment] The paper is quite lengthy. Therefore, sometimes it is hard to comprehend the connection between different parts of the paper. The authors have a brief discussion of summary of results in the beginning. But a expansion of this part can better guide the readers. [Our reply] We thank the referee for suggesting to expand the summary of results. We have added a new subsection in the introduction with an illuminating example of the results we obtain in the main text. We illustrate the results using the example toric code and the RBH model.
-
[Referee comment] The definition of strange correlator does not involve the notion of duality. In Section 4, the authors seem to make a connection between these two ideas. Can the authors elaborate the motivation for such connections preferably in the beginning of the section? [Our reply] We thank the referee for suggesting to include a connection between the strange correlator and duality. At the beginning of section 4.2, we have added a connection between the strange correlator and the symmetry topological field theory (SymTFT). As symmetry topological field theory is studied in the context of dualities, the strange correlator can be thought of as a natural tool to study dualities. We illustrate that the dualities obtained in our manuscript using strange correlator construction can be interpreted as changing the boundary condition in SymTFT.

---

## Round 2 · List of Changes

1. We made new subsections 1.1, 1.2, and 1.3 in the Introduction. This is to explain our results by simple examples without using the rather technical chain complex machinery.
2. We modified the beginning of Section 2.2. We replaced ``objects'' by ``abstract symbols''. We also refer to the toric code example as an intuitive explanation.
3. Below (35), we added some words to clarify the concept of a dual group.
4. We added a new footnote (8) to define and explain Im, Ker, and Hom.
5. We edited footnotes (now 10 and 13) and cited the new subsections in the Introduction.
6. We changed the beginning of Section 4.2 to motivate the connection between strange correlators and dualities.
7. We added a sentence below (156).
8. We moved ``see Appendix D for a proof'' to above (155) to accommodate the new sentence and avoid confusion.
9. At the beginning of Section 4.2.1, we added ``which we have already discussed in Section 1.2.2 to refer to a new subsection.
10. At the beginning of Section 4.3, a new sentence starting with ``The special case of the RBH model was explained...'' and ``another'' were added to refer to a new subsection.
11. Near the end of Section 4.3, below (172), two paragraphs were added to clarify the relation between the strange correlator and dualities.
12. Near the end of Appendix D, below (257), two paragraphs were added to clarify the relation between the strange correlator and dualities.
13. After (255), ``and (254)'' was added.
14. A new figure (Figure 9) was added.
15. Two new references ([59] and [60]) were added.

---

## Editorial Decision

published